



# The Norwegian Earth System Model, NorESM2 - Evaluation of the CMIP6 DECK and historical simulations

Øyvind Seland[1], Mats Bentsen[2], Lise Seland Graff[1], Dirk Olivié[1], Thomas Toniazzo[2,6],
Ada Gjermundsen[1], Jens Boldingh Debernard[1], Alok Kumar Gupta[2], Yanchun He[3], Alf Kirkevåg[1],
Jörg Schwinger[2], Jerry Tjiputra[2], Kjetil Schanke Aas[4], Ingo Bethke[5], Yuanchao Fan[2], Jan Griesfeller[1],
Alf Grini[1], Chuncheng Guo[2], Mehmet Ilicak[7,2], Inger Helene Hafsahl Karset[4], Oskar Landgren[1],
Johan Liakka[3], Kine Onsum Moseid[1], Aleksi Nummelin[2], Clemens Spensberger[5], Hui Tang[4],
Zhongshi Zhang[2], Christoph Heinze[5], Trond Iversen[1,4], and Michael Schulz[1,4]

[1]Norwegian Meteorological Institute, Oslo, Norway
[2]Norwegian Research Centre and Bjerknes Centre for Climate Research, Bergen, Norway
[3]Nansen Environmental and Remote Sensing Center, Bergen, Norway
[4]University of Oslo, Department of Geosciences, Oslo, Norway
[5]Geophysical Institute, University in Bergen, and Bjerknes Centre for Climate Research, Bergen, Norway
[6]Stockholm University, Department of Meteorology, Stockholm, Sweden
[7]Eurasia Institute of Earth Sciences, Istanbul Technical University, Istanbul, Turkey

**Correspondence:** Øyvind Seland (oyvindse@met.no)

**Abstract.** The second version of the fully coupled Norwegian Earth System Model (NorESM2) is presented and evaluated. NorESM2 is based on the second version of the Community Earth System Model (CESM2), but has entirely different ocean and ocean biogeochemistry models; a new module for aerosols in the atmosphere model along with aerosol-radiation-cloud interactions and changes related to the moist energy formulation, deep convection scheme and angular momentum conserva-

tion; modified albedo and air-sea turbulent flux calculations; and minor changes to land and sea ice models. We show results from low ($\sim$2°) and medium ($\sim$1°) atmosphere-land resolution versions of NorESM2 that have both been used to carry out simulations for the sixth phase of the Coupled Model Intercomparison Project (CMIP6). The stability of the pre-industrial climate and the sensitivity of the model to abrupt and gradual quadrupling of $CO_2$ is assessed, along with the ability of the model to simulate the historical climate under the CMIP6 forcings. As compared to observations and reanalyses, NorESM2

represents an improvement over previous versions of NorESM in most aspects. NorESM2 is less sensitive to greenhouse gas forcing than its predecessors, with an equilibrium climate sensitivity of 2.5 K in both resolutions on a 150 year frame. We also consider the model response to future scenarios as defined by selected shared socioeconomic pathways (SSPs) from the Scenario Model Intercomparison Project defined under CMIP6. Under the four scenarios SSP1-2.6, SSP2-4.5, SSP3-7.0, and SSP5-8.5, the warming in the period 2090–2099 compared to 1850–1879 reaches 1.3, 2.2, 3.0, and 3.9 K in NorESM2-LM,

and 1.3, 2.1, 3.1, and 3.9 K in NorESM–MM, robustly similar in both resolutions. NorESM2-LM shows a rather satisfactorily evolution of recent sea ice area. In NorESM2-LM an ice free Arctic Ocean is only avoided in the SSP1-2.6 scenario.



## 1 Introduction

The Norwegian Earth System Model version 2 (NorESM2) is the second generation of the coupled Earth System Model developed by the Norwegian Climate Center (NCC), and is the successor of NorESM1 (Bentsen et al., 2013; Iversen et al., 2013;

Kirkevåg et al., 2013; Tjiputra et al., 2013) which has been used in the 5th phase of the Coupled Model Intercomparison Project (CMIP5; Taylor et al., 2012), and for evaluation of the difference between a 1.5 and 2 °C warmer world than pre-industrial (Graff et al., 2019). NorESM2 is based on the Community Earth System Model CESM2.1 (Danabasoglu et al., 2019). Although large parts of NorESM are similar to CESM, there are several important differences: NorESM uses the isopycnic coordinate Bergen Layered Ocean Model (BLOM; Bentsen et al., in prep.), uses a different aerosol module OsloAero6 (Kirkevåg et al.,

2018; Olivié et al., in prep.), contains specific modifications and tunings of the atmosphere component (Toniazzo et al., 2019; Toniazzo et al., in prep.), and contains the iHAMOCC model to describe ocean biogeochemistry (Tjiputra et al., 2019).

Many changes have contributed to the development of NorESM1 into NorESM2. The model has benefited from the evolution of the parent model CCSM4.0 into CESM2.1, comprising the change of the atmosphere component from CAM4 to CAM6, the land component from CLM4 to CLM5, and the sea ice component from CICE4 to CICE5. Also, specific developments have

been implemented in the description of aerosols and their coupling to clouds and radiation (Kirkevåg et al., 2018), in addition to harmonizing the implementation of the aerosol scheme with the standard aerosol schemes in CESM. To extend the capabilities of NorESM as an Earth System Model, a strong focus has been put on the interactive description of natural emissions of aerosols and their precursors, and tightening the coupling between the different Earth System components. Finally, the ocean model (Bentsen et al., in prep.) and the ocean biogeochemistry module (Schwinger et al., 2016; Tjiputra et al., 2019) have been

further developed.

This manuscript gives a description of NorESM2, and a basic evaluation against observations of the simulation of the atmosphere, sea ice, and ocean in a small set of baseline long-duration experiments with the new model. It focuses on such aspects as the simulated climatology, its stability and internal variability, and also on its response under historical and enhanced-greenhouse gas scenario forcings.

Currently, NorESM2 exists in three versions. The two versions presented here are NorESM2-LM and NorESM2-MM: they differ in the horizontal resolution of the atmosphere and land component (approximately 2° for LM and 1° in MM), but share the same horizontal resolution of 1° for the ocean and sea ice components. These versions are otherwise identical, except for a very limited number of parameter settings in the atmosphere component, and the parameterisation used to diagnose the fraction of ice-clouds. A third version of the model is the $CO_2$-emission driven NorESM2-LME (as opposed to concentration driven),

which can be used for interactive carbon-cycle studies, but is identical to NorESM2-LM in all other aspects.

A range of climate models and model versions participates in the sixth phase of the Coupled Model Intercomparison Project (CMIP6; Eyring et al., 2016). Also NorESM2 has been used to contribute to CMIP6, and all the data generated by the participating models, including NorESM2, can be downloaded from the CMIP6 multi-model data archive.

An overview of the model which highlights the differences since previous versions and from CESM2 is given in Sect. 2,

and a short summary of model initialization and tuning is presented in Sect. 3. A short description of the CMIP6 experiments





considered in this paper is provided in Sect. 4 along with results documenting model stability, climate sensitivity, and the time evolution of selected climate variables during the historical period and future scenarios. Section 5 documents the climatological mean state of the model and atmospheric circulation patterns, with emphasis on ocean temperatures, salinity, Sea Level Anomalies (SLA; Sect. 5.1), sea ice (Sect. 5.2), atmospheric temperature and zonal winds (Sect. 5.3), extratropical storm tracks (Sect. 5.4), precipitation and the fresh water cycle (Sect. 5.6), Northern Hemisphere blocking (Sect. 5.7), the Madden-Julian Oscillation (Sect. 5.8), and the El Niño Southern Oscillation (ENSO; Sect. 5.9). A summary and discussion is provided at the end in Sect. 6.

## 2 From CESM2 and NorESM1 to NorESM2: description and updates

As described in the introduction, NorESM2 is built on the structure and many of the components of CESM2.1 (Danabasoglu et al., 2019), but with several modifications. The development work described in this section was based on a slightly older version, CESM2, but updated to CESM2.1 during the tuning phase of the model (Sect. 3). An overview of the model components can be found in Fig. 1.

Compared to the Community Atmosphere Model version 6 (CAM6; Bogenschutz et al., 2018) of CESM2, the atmospheric component of NorESM2, CAM6-Nor, incorporates a number of modifications. These involve the independently developed module for the life-cycle of particulate aerosols, and the representation of aerosol-radiation-cloud interactions (Kirkevåg et al., 2013, 2018); changes in the moist convection scheme and the local moist energy formulation (Toniazzo et al., in prep.); global conservation of rotational momentum (Toniazzo et al., 2019); and an updated parameterisation of the surface flux layer for the computation of air-sea fluxes. The last two of these modifications have recently been included in the CESM CAM6 code repositories and are available as namelist options. A summary of these changes is given in the atmospheric model section (Sect. 2.2).

The ocean model BLOM is an updated version of the elaborated Miami Isopycnic Coordinate Ocean Model (MICOM) used in NorESM1 (Bentsen et al., 2013). BLOM is coupled to the isopycnic coordinate Hamburg Ocean Carbon Cycle Model (iHAMOCC; Tjiputra et al., 2019), an updated version of the carbon-cycle model found in NorESM1 (Tjiputra et al., 2013). Brief descriptions of the ocean and ocean biochemistry models are given in Sect. 2.3 and 2.4.

The sea ice model, version 5.1.2 of the Los Alamos Sea Ice Model (CICE5.1.2; Hunke et al., 2015), and the land-model, the Community Land Model version 5 (CLM5; Lawrence et al., 2019), only differ from the versions used in CESM2.1 by minor changes which are summarised in Sect. 2.5 and 2.6. The river model is the Model for Scale Adaptive River Transport (MOSART; Li et al., 2013) is identical to the version found in CESM2.1 and hence is not described here. The coupler structure is retained as in CESM2.1 but with changes in flux and albedo calculations summarized below.

The land-ice component included in CESM2 (the Community Ice Sheet Model; CISM; Lipscomb et al., 2019) is not activated in NorESM2 at this time.





## 2.1 Model versions and the coupled model system

Due to the generally high computational cost of NorESM2, two different versions have been set up with different CPU time demands per simulated year. In these, the atmospheric and land components have different horizontal resolutions, one with nominal 1° which is considered medium (M) resolution and another one with nominal 2° which is considered low (L) resolution. The ocean and sea ice components are run with medium (1°) resolution in both versions. To facilitate distinguishing between the different resolutions when discussing set-up and results, a two letter suffix is added to the NorESM2, "LM" for low-resolution atmosphere/land and medium resolution ocean/sea ice and "MM" for medium resolution of both atmosphere/land and ocean/sea ice. Both versions use the "low-top" version of CAM6, with 32 layers in the vertical and model top at 3.6 hPa (40 km). NorESM2-LM is used for most of the CMIP6 simulations, while NorESM2-MM is only used for a limited number of experiments.

## 2.2 Atmospheric model, CAM6-Nor

The atmospheric model component of NorESM2 is built on the CAM6 version from CESM2.1, but with particulate aerosols and the aerosol-radiation-cloud interaction parameterisation from NorESM1 and NorESM1.2 as described by Kirkevåg et al. (2013, 2018). NorESM2-specific changes to model physics and dynamics which are not aerosol related, are described by Toniazzo et al. (2019) and Toniazzo et al. (in prep.).

The latest updates in the aerosol modules (that is, the changes between NorESM1.2 and NorESM2) are described by Olivié et al. (in prep.). Very briefly these can be summarized as follows.

The CMIP6 forcing input files now replace the corresponding CMIP5 files in NorESM2. Greenhouse gas concentrations of carbon dioxide ($CO_2$), methane ($CH_4$), nitrous oxide ($N_2O$), equivalent trichlorofluoromethane (CFC-11), and dichlorodifluoromethane (CFC-12) follow Meinshausen et al. (2017), and solar forcing is prescribed according to Matthes et al. (2017). Emissions that are not calculated online by the model have been updated. Anthropogenic emissions of black carbon (BC), organic matter (OM), and sulfur dioxide ($SO_2$) are prescribed according to Hoesly et al. (2018), and biomass burning emission strengths follow van Marle et al. (2017) applying a vertical distribution according to Dentener et al. (2006). As in NorESM1, continuous tropospheric outgassing of $SO_2$ by volcanoes is taken into account, but we have also added the tropospheric contribution of explosive volcanoes (Dentener et al., 2006). As in NorESM1, an OM/OC ratio of 1.4 is taken for fossil fuel emissions and 2.6 for biomass burning emissions, and sulphur emissions are assumed to be 97.5 % $SO_2$ and 2.5 % $SO_4$. For oxidant concentrations used to describe the formation of secondary aerosols, for ozone ($O_3$) concentrations used in the radiative transfer calculations, and for $H_2O$ production rates due to $CH_4$ oxidation in the stratosphere, we use the same fields as used in CESM2(CAM6) (Danabasoglu et al., 2019), which originate from a pre-industrial control and three historical simulations of CESM2(WACCM6) (Gettelman et al., 2019b). The oxidant fields (hydroxyl radical (OH), nitrate radical ($NO_3$), hydroperoxy radical ($HO_2$) and $O_3$) and $H_2O$ emission rates are 3-dimensional monthly varying fields, and the $O_3$ fields used in the radiation are zonally averaged 5-daily varying fields; in the historical simulations these fields are provided at a decadal frequency (Danabasoglu et al., submitted). The impact of stratospheric aerosol in NorESM1 was taken into account by prescribing volcanic



aerosol mass concentrations. In NorESM2, prescribed optical properties from CMIP6 are instead used, and are integrated in the calculation of total optical parameters for use in the radiation module together with other aerosols. The assumed complex refractive index of mineral dust for wavelengths below 15 $\mu$m has furthermore been changed according to more recent research (for details, see Olivié et al. (in prep.) and references therein), compared to the values applied in NorESM1.2.

There have also been some changes in parameterisations and aerosol-specific tunings. Due to exaggerated extinction by
mineral dust in dust-dominated regions in the previous model version, a scaling coefficient in the emission flux for interactive dust emissions has been reset to the original CAM6 value, in effect halving the emissions. Sea-salt emissions have been tuned up by changing the wind speed (at 10 m height, U10) dependency to the recommended value by Salter et al. (2015), now being proportional to U10$^{3.74}$ instead of U10$^{3.41}$. This has been done as a measure to help cool the model sufficiently in the spin-up and control simulation in spite of already exaggerated sea-salt aerosol optical depth (Gliss et al., in prep.) and surface mass
concentrations (Olivié et al., in prep.) compared to in-situ measurements. Since the emission flux of oceanic primary organic aerosols is proportional to that of fine sea-salt aerosols (Kirkevåg et al., 2018), this specific change also has an impact on the natural oceanic organic matter emissions.

The aerosol nucleation formulation described by Kirkevåg et al. (2018) has been updated by allowing all pre-existing particles to act as coagulation sinks for freshly nucleated particles (Sporre et al., 2019) to give a more realistic rate of survival
for these 2 nm nucleation particles into the smallest explicitly modeled mode/mixture of co-nucleated sulfate and secondary organic aerosols. This reduces the number concentrations of fine-mode particles, while increasing their size, which in effect yields increased cloud condensation nuclei and cloud droplet concentrations.

In NorESM2, oceanic dimethyl sulfide (DMS) emission is prognostically simulated by the ocean biogeochemistry component (Sect. 2.4), hence allowing for a direct biogeochemical climate feedback in coupled simulations. The DMS air-sea flux is
simulated as a function of upper-ocean biological production following the formulation of Six and Maier-Reimer (1996) and was first tested in the NorESM model framework by Schwinger et al. (2017).

While hygroscopic swelling of aerosols in earlier versions always used the grid averaged relative humidity as input to lookup tables which take into account the effects of hygroscopic growth on water uptake and optical properties, in CAM6-Nor we instead use the mean cloud-free relative humidity, as in the host model CAM6 and a number of other atmospheric models
(Textor et al., 2006; Kirkevåg et al., 2018; Gliss et al., in prep.).

The other differences of CAM6-Nor relative to CAM6 are summarised as follows. A correction to the zonal wind increments due to the Lin and Rood (1997) dynamical core is introduced in order to achieve global conservation of atmospheric angular momentum along the Earth's axis of rotation, as described and discussed in (Toniazzo et al., 2019). The local energy update of the model is also modified by including a missing term (the hydrostatic pressure work) related with changes in atmospheric
water vapour and thus achieve better local energy conservation. Finally, a set of modifications to the deep convection scheme is introduced which eliminate most of the resolution dependence of the scheme, and mitigate the cold tropospheric bias of CAM6. The energy and convection changes (which are not available in the CAM6 code repository) are described in Toniazzo et al. (in prep.).





## 2.3 Ocean model

The ocean component BLOM is based on the version of MICOM used in NorESM1 and shares the use of near-isopycnic interior layers and variable density layers in the surface well-mixed boundary layer. The dynamical core is also very similar but with notable differences in physical parameterisations and coupling. For vertical shear-induced mixing a second-order turbulence closure (Umlauf and Burchard, 2005; Ilicak et al., 2008) using a one equation closure within the family of $k - \varepsilon$ models has replaced a parameterisation using the local gradient Richardson number according to Large et al. (1994). Parameterised eddy-

induced transport is modified to more closely follow the Gent and McWilliams (1990) parameterisation with the main impact of increased upper ocean stratification and reduced mixed layer depths. As for NorESM1-MICOM, the estimation of diffusivity for eddy-induced transport and isopycnic eddy diffusion of tracers is based on the Eden et al. (2009) implementation of Eden and Greatbatch (2008) with their diagnostic equation for the eddy length scale, but modified to give a spatially smoother and generally reduced diffusivity. Hourly exchange of state and flux variables with other components is now used compared to

daily ocean coupling in NorESM1. The sub-diurnal coupling allows for the parameterisation of additional upper ocean mixing processes. Representation of mixed layer processes is modified to work well with the higher frequency coupling and in general to mitigate a deep mixed layer bias found in NorESM1 simulations. The penetration profile of shortwave radiation is modified, leading to a shallower absorption in NorESM2 compared to NorESM1. With respect to coupling to the sea ice model, BLOM and CICE now use a consistent salinity dependent seawater freezing temperature (Assur, 1958). Selective damping of external

inertia–gravity waves in shallow regions is enabled to mitigate an issue with unphysical oceanic variability in high latitude shelf regions, causing excessive sea ice formation due to breakup and ridging in CMIP5 versions of NorESM1.

For the CMIP6 contribution, BLOM uses identical parameters and configuration in coupled ocean-sea ice OMIP (Ocean Model Intercomparison Project; Griffies et al., 2016) experiments and fully coupled NorESM2-LM and NorESM2-MM experiments, except for sea surface salinity restoring in OMIP experiments. As for NorESM1, 53 model layers are used with

two non-isopycnic surface layers and the same layer reference potential densities for the layers below. A tripolar grid is used instead of the bipolar grid in CMIP5 versions of NorESM1, allowing for approximately a doubling of the model time step. At the equator the grid resolution is $1°$ zonally and $1/4°$ meridionally, gradually approaching more isotropic grid cells at higher latitudes. The model bathymetry is found by averaging the S2004 (Marks and Smith, 2006) data points contained in each model grid cell with additional editing of sills and passages to their actual depths. The metric scale factors are edited to the realistic

width of the Strait of Gibraltar so that strong velocity shears can be formed, enabling realistic mixing of Mediterranean water entering the Atlantic Ocean.

OMIP provides protocols for two different forcing datasets, OMIP1 (Large and Yeager, 2009) and OMIP2 (Tsujino et al., 2018). Tsujino et al. (2019) is a model intercomparison evaluating OMIP1 and OMIP2 experiments, including BLOM/CICE of NorESM2. Further details on the BLOM model and its performance in OMIP coupled ocean-sea ice simulations can be found

in Bentsen et al. (in prep.).





## 2.4 Ocean biogeochemistry

The ocean biogeochemistry component iHAMOCC (isopycnic coordinate HAMburg Ocean Carbon Cycle model) is an updated version of the ocean biogeochemistry module used in NorESM1. The model includes prognostic inorganic carbon chemistry following Dickson et al. (2007). A Nutrient Phytoplankton Zooplankton Detritus (NPZD) type ecosystem model (Six and Maier-Reimer, 1996) represents the lower trophic biological productivity in the upper ocean. The updated version includes riverine inputs of biogeochemical constituents to the coastal ocean. Atmospheric nitrogen deposition is prescribed according to the data provided by CMIP6. The parameterisations of the particulate organic carbon sinking scheme, dissolved iron sources and sinks, nitrogen fixation, and other nutrient cycling have been updated as well. NorESM2 also simulates preformed and natural inorganic carbon tracers, which can be used to facilitate a more detailed diagnostic of interior ocean biogeochemical dynamics. Details on the updates and improvements of the ocean biogeochemical component of NorESM2 are provided in Tjiputra et al. (2019).

## 2.5 Sea ice

The sea ice model component is based upon version 5.1.2 of the CICE sea ice model of Hunke et al. (2015). A NorESM2-specific change is including the effect of wind drift of snow into ocean following Lecomte et al. (2013), as described in Bentsen et al. (in prep).

CICE model uses a prognostic ice thickness distribution (ITD) with five thickness categories. The standard CICE elastic–viscous–plastic (EVP) rheology is used for ice dynamics (Hunke et al., 2015). The model uses mushy-layer thermodynamics with prognostic sea ice salinity from Turner and Hunke (2015). Radiation is calculated using the Delta-Eddington scheme of Briegleb and Light (2007), with melt ponds modeled on level, undeformed ice, as in Hunke et al. (2013).

CICE uses the same horizontal grid as the ocean model (Sect. 2.3), and is configured with 8 layers of ice and 3 of snow.

## 2.6 Land

The NorESM2 land model is CLM5 (Lawrence et al., 2019) with one minor modification described below. A general description of the model will therefore not be presented here. It should however be noted that CLM5 has a new treatment of nitrogen-carbon limitation, which is very important for the carbon cycle in NorESM2 and has increased the land carbon uptake substantially relative to NorESM1 (Arora et al., 2019).

In NorESM2, one specific modification was made to the surface water treatment in CLM. The surface water pool is a new feature replacing the wetland land unit in earlier versions of CLM (introduced in CLM4.5). This water pool does not have a frozen state, but is added to the snow-pack when frozen. To avoid water being looped between surface water and snow during alternating cold and warm periods, we remove infiltration excess water as runoff if the temperature of the surface water pool is below freezing. This was done to mitigate a positive snow bias and an artificial snow depth increase found in some Arctic locations during melting conditions.





## 2.7  Coupler

The state and flux exchanges between model components and software infrastructure for configuring, building and execution of model experiments is handled by the CESM2 coupler Common Infrastructure for Modeling the Earth (CIME; Danabasoglu et al., 2019). The coupler computes the turbulent air-sea fluxes of heat and momentum and in NorESM2 this is implemented as a version of the COARE-3 (Fairall et al., 2003) scheme, replacing the calculation based on Large and Yeager (2004) in CESM2. State and flux exchanges via the coupler between atmosphere, land and sea ice components occur half-hourly, aligned with the atmosphere time step, while the ocean exchanges with the coupler every hour. CIME also provides common utility functions and among these are estimation of solar zenith angle. In NorESM2, this utility function is modified with associated changes in atmosphere, land and sea ice components, ensuring that all albedo calculations use zenith angle averaged over the components time-step instead of instantaneous angles.

## 3  NorESM2 initialisation and tuning

Most of the general development of the model as described in Sect. 2 was tested in stand-alone versions of the different model components, CAM6-Nor in present-day AMIP-mode under year 2000 conditions and BLOM and iHAMOCC forced by a data-atmosphere. The main targets of these separate experiments were to test improved representations of the physical processes in the simulations, to mitigate model systematic biases when compared to the observed climate, and to reduce the residual radiative imbalance at the top of the model atmosphere (hereafter RESTOM) given prescribed SSTs from observations.

The first coupled version of NorESM2 included all changes described in Sect. 2. This version was heavily tested in a pre-industrial setting (as defined in Sect. 4).

This initial version of the coupled model was initialized using a hybrid of observational estimates and earlier model simulations. The ocean model was initialised with zero velocities and temperature and salinity fields from the Polar science center Hydrographic Climatology (PHC) 3.0 (updated from Steele et al., 2001). Following the OMIP protocol (Orr et al., 2017), the nutrients (phosphate, nitrate, and silicate) and oxygen fields in NorESM2 were initialized with the gridded climatological fields of the World Ocean Atlas database (Garcia et al., 2014a, b). For dissolved inorganic carbon and total alkalinity, we used the pre-industrial and climatological values from the Global Ocean Data Analysis Project (GLODAPv2) database (Lauvset et al., 2016). Other biogeochemical tracers are initialized using values close to zero. CAM and CLM were initialized using the files included in the CESM2 release. Aerosols and aerosol precursors were initialised to near zero values. As there were no low-resolution pre-industrial initial files for the land model available this was replaced by an interpolation of the 1° initial file from CESM2. At a later stage in the coupled spin-up, the land surface fields were re-initialised from a long (approximately 1400 years) stand-alone CLM spin-up simulation driven by a repeat 50-yrs climatology fields of the earlier coupled run.

While preparing the coupled model for the spin-up, it was found that the sensitivities of important climatological variables, including RESTOM, to changes in parameterisations were often different in the coupled configuration compared to stand-alone simulations with the individual components using prescribed boundary conditions. The coupled response could be both amplified or damped with respect to single-component simulations. As a result, tuning test simulations had to be performed





in coupled mode and the model had to be restarted from the initial state several times. Similar to CESM, NorESM2 adjusted towards its climatology with an initial phase of strong cooling in the high latitudes of the northern hemisphere, after which an intensification of ocean heat advection stabilised the simulation. After that point, the climatology tended to settle to a steady drift. In order to save computer resources, minor tuning, especially toward balanced RESTOM, was performed during this second stage of the spin-up phase of the model. Alongside the final tuning, the CESM components were updated to the

versions found in CESM2.1.

The main goal of the tuning process was to create a reasonably stable pre-industrial control simulation. The simulation can produce a steady climatology only if the time-average radiative imbalance on the top of the model (RESTOM) vanishes. In practice, a commonly used target is for RESTOM to be within $\pm 0.1\,\mathrm{W\,m^{-2}}$. Secondary tuning targets are to obtain and maintain values of mean atmospheric and ocean temperatures close to observations. As the ocean heat again reflects the top of

the atmosphere imbalance, the two requirements are strongly connected. One additional constraint was that the tuning should not significantly degrade other important climatological variables such as temperature, precipitation, cloud, and the main mode of coupled variability, i.e. the El-Niño Southern Oscillation (ENSO). Each tuning step was performed in isolation, and an effort was made to ensure the greatest possible similarities in the two model configurations LM and MM. No tuning was performed that attempted to target other modes of variability beside ENSO, or a particular climate response to external forcings, e.g. from

changes in greenhouse gas concentration, anthropogenic aerosol emissions, or volcanic or solar forcing.

As found in CESM2 (Danabasoglu et al., 2019), also NorESM2 had development of excessive sea ice cover in the Labrador Sea (LS) region, although the temporal development in NorESM2 differed from CESM2. For any tested combination of parameter choices, NorESM2 developed excessive LS sea ice cover starting around year 60 after model initialisation. This was however only a temporary model state and in all experiments the sea ice returned close to observed state in the LS region after

additional 60–80 model years of simulation.

One of the most common methods to tune RESTOM is to change the amount and thickness of low clouds. The main parameter used for tuning the low clouds in the CLUBB scheme is the "gamma" parameter, which controls the skewness of the assumed Gaussian PDF for subgrid velocities. A low gamma implies weaker entrainment at the top of the clouds, in particular for marine stratocumulus. This increases the amount of low clouds and results in a higher short-wave cloud forcing.

Given the same gamma values the RESTOM was higher in the low resolution version of the model. In addition the sensitivity to the change of the gamma parameter was different in the two model resolutions, so a different choice of gamma was needed for the two resolutions. The final parameter values are well within the gamma range of 0.1–0.5 tested by Zhang et al. (2018). The resulting bias in short-wave cloud forcing (SWCF) was somewhat off-set by regulating the parameter dcs (autoconversion size threshold for cloud ice to snow) in NorESM2-LM but this had only a small impact on the tropospheric temperature bias.

Changing dcs in NorESM2-MM did not improve the overall skill of this model version compared to the initial value so was not used for this versions

While the amount of change in SWCF could be estimated by running the atmosphere and land model in a stand-alone configuration, the change in RESTOM in coupled set-up was small compared to the change in cloud forcing. Further attempts at reducing positive RESTOM by tuning the boundary layer stability were neutralised by SST adjustment, while worsening





the tropospheric cold bias. A more effective tuning of low cloud radiative effects was achieved by modifying air-sea fluxes of sea salt and DMS *detail here*. As described in Sect. 2.2 the disadvantage of increasing the sea-salt flux, however, is that this resulted in too dominant marine aerosols with respect to optical thickness and surface mass concentrations. RESTOM was decisively reduced by increasing outgoing long-wave radiation. This was achieved in three ways. First, alterations were made to the Zhang and McFarlane (1995) convection scheme, as described in Toniazzo et al. (in prep.), aimed at increasing mid-

and high-altitude latent heating of the atmosphere. Second, higher sea-surface temperatures were achieved by reverting to the NorESM1 level of ocean background vertical mixing after having used up to 50% higher diffusivity for the purpose of reducing upper ocean biases. Third, positive cloud radiative forcing in the terrestrial radiation spectrum was reduced by intervening on the parameterisation of ice cloud fraction.

Several versions of the ice cloud fraction parameterisation are provided (as namelist options) in CESM. Initial tuning of the
parameters of the CESM2 default option appeared promising, but coupled adjustment again tended to neutralise the effect on model radiative imbalance. An effective reduction in the high- and mid-level cloud cover could only be achieved by switching parameterisation in NorESM2-LM, such that there is no direct functional dependence of ice cloud fraction on environmental relative humidity (this is option number 4 in CESM). By contrast, the CESM default scheme (option number 5, with explicit RH dependence) could be tuned sufficiently in NorESM2-MM, by including a minor modification that narrows the range of
cloud sensitivity to environmental RH (and thus provides a continuous switch between the two parameterisations). This purely empirical part of the cloud parametrisation of CESM2 is very poorly constrained by observations, and its future development might be better rooted in physical processes.

Compared to Schwinger et al. (2017), NorESM2 has doubled the diatom-mediated DMS production parameter in order to maintain the observed high DMS concentration at high latitudes. This tuning is necessary due to the lower biological production
simulated in NorESM2 (relative to NorESM1), which is a better representation to the observations, during spring bloom in both hemispheres (Tjiputra et al., 2019).

## 4 Control simulations and model response to forcing

This section presents a basic description of the climatology simulated in CMIP6 experiments with the two versions of the model, NorESM2-LM and NorESM2-MM (Sect. 2.1). We consider the time evolution of temperature in historical and enhanced
greenhouse gas climate scenarios, along with aspects of the ocean circulation and sea ice. We validate the historical coupled simulations against observational estimates and reanalyses, and compare them with results from simulations with previous versions of NorESM (Sect. 5).

We consider three sets of experiments that are important for documentation and application of CMIP6 models: the DECK experiments, the CMIP6 Historical experiment, and the Tier 1 experiments of the ScenarioMIP. A brief description of the set-up
of these experiments is given in Sect. 4.1.

The analysis is divided into three parts. Section 4.2 focuses on the stability of the pre-industrial control simulation. In Sect. 4.3, we consider the simulated climate sensitivity to abrupt and gradual quadrupling of $CO_2$. A brief analysis of the



warming, sea ice, AMOC, and the transport through the Drake Passage in the historical simulations and the scenarios is given in Sect. 4.4.

## 4.1 Experiment set-up

As described by Eyring et al. (2016), a set of common experiments known as DECK (Diagnostic, Evaluation, and Characterization of Klima) has been defined to better coordinate different model intercomparisons and provide continuity for model development and model progress studies. The DECK consists of the following four baseline experiments: (1) the Historical Atmospheric Model Intercomparison Project (AMIP) experiment; (2) the pre-industrial control (piControl) experiment defined by estimated forcings from 1850; (3) the experiment corresponding to the piControl, but where the $CO_2$ concentrations are instantaneously quadrupled at the start of the run (abrupt-4xCO$_2$); (4) the experiment corresponding to the piControl, but where the $CO_2$ concentrations are gradually increased by 1% per year (1pctCO$_2$). Both abrupt-4xCO2 and 1pctCO2 were started from year 1 of the control.

The DECK was run with both versions of the model (NorESM2-LM and NorESM2-MM) and we here consider results from the pre-industrial control and the abrupt-4xCO$_2$ and 1pctCO$_2$ (Sect. 4.2–4.3). As this paper focuses on the coupled aspect of NorESM2, the AMIP runs are not included here, but are described in Olivié et al. (in prep.).

Another experiment required for CMIP6 and important for model evaluation is the historical experiment which is run with forcings from the so-called historical period, defined as 1850–2014. For the low-resolution version of the model (NorESM2-LM), we have carried out a small ensemble consisting of 3 members. The first ensemble member was initialised using initial conditions from the first year of the control experiment, while members number 2 and 3 are initialised from years 32 and 62 respectively. For NorESM2-MM, only a single ensemble member had been carried out when this paper was written. Consistent with historical member 1 from NorESM2-LM, the NorESM2-MM historical experiment was started from identical initial conditions to the NorESM2-MM control simulation.

One of the most important applications for Earth system models is to provide estimates for future climate development. This is typically done using scenarios where critical input for climate models through description and quantification of both land-use change and amount of climate forcing agents in the atmosphere is provided. The latter can be described either as changes in emissions or concentrations. The design of scenarios are based on a combination of socioeconomic and technological development, named the Shared Socioeconomic Pathways (SSPs), with future climate radiative forcing (RF) pathways (RCPs) in a scenario matrix architecture (Gidden et al., 2019).

The simulations included here are the Tier 1 experiments of the ScenarioMIP (O'Neill et al., 2016): SSP1-2.6, SSP2-4.5, SSP3-7.0, and SSP5-8.5. The forcing fields for all the experiments are generally the same as used in CESM2.1. This includes solar forcing, prescribed oxidants used for describing secondary aerosol formation, greenhouse gas concentrations, stratospheric $H_2O$ production from $CH_4$ oxidation, ozone used in the radiative transfer calculations, and land-use. While the experiments in this paper use prescribed greenhouse gas concentrations, NorESM2 can also be run with $CO_2$ emissions as described by Tjiputra et al. (2019).



Files for stratospheric aerosols and emissions of aerosols and aerosol precursors were created based on the input found at the input4mips website: https://esgf-node.llnl.gov/projects/input4mips/. In addition, sulphur from tropospheric volcanoes was included similarly to Kirkevåg et al. (2018), see Sect 2.2.

## 4.2 Stability of the control climate

After the tuning period and the spin-up, both NorESM2-LM and NorESM2-MM were integrated for 500 years with steady pre-industrial forcings to produce the piControl experiments. Below, we present a basic analysis of the general state and drift of important parameters in the system.

During the control integration the forcings as well as the parameter choices were kept constant. Given a sufficiently long spin-up there should be no long-term drift in the state variables or fluxes. In practice any residual trends in the simulated control
climatology should be negligibly small compared with the signal resulting from the response to changes in climate forcings in the historical and enhanced-greenhouse gas experiments. A reasonable target is to maintain RESTOM within $\pm 0.1 \, \mathrm{W \, m^{-2}}$. Any small imbalance in RESTOM is typically reflected in a small trend in ocean temperature. A time-series of the Atlantic Meridional Overturning Circulation (AMOC) can give an indication of the stability of the general ocean circulation.

Figure 2 shows time-series of related global-means in the piControl simulations from NorESM2-LM and NorESM2-MM.
As can be seen in the figure the drift is generally small and comparable for the two model versions. The top-of-the-atmosphere radiative imbalance is -0.057 $\mathrm{W \, m^{-2}}$ for NorESM2-LM and -0.065 $\mathrm{W \, m^{-2}}$ for NorESM2-MM. The ocean volume temperature change of 0.03 K over 500 years is much smaller than the rate of warming observed during the last 50 years. Similarly, there are positive trends in global mean ocean salinity of $2.6 \times 10^{-5} \, \mathrm{g \, kg^{-1}}$ and $4.7 \times 10^{-5} \, \mathrm{g \, kg^{-1}}$ over 500 years for NorESM2-LM and NorESM2-MM, respectively, that we consider small since for NorESM2-MM this is equivalent to an average surface freshwater loss of $2.9 \times 10^{-5} \, \mathrm{mm \, day^{-1}}$. The remaining trends are not significantly different from 0 on a 5% level t-test. We
found however a slight decrease in DMS sea-to-air flux of 2% over the 500 year control period, reflecting a residual drift in ocean bio-geochemistry. AMOC variations are reasonably small and show no significant trend.

## 4.3 Equilibrium climate sensitivity and transient response

The two enhanced greenhouse gas experiments of the CMIP-DECK aim to facilitate a comparison of climate change in response
to a standardized specified forcing across different models. The corresponding NorESM2 simulations were started at the same nominal model year and with the same initial conditions as piControl. They are referred to as abrupt4×$CO_2$ and 1pct$CO_2$.

Figure 3 shows the time evolution of near-surface temperature for abrupt4×$CO_2$, 1pct$CO_2$ and piControl for both model configurations. Three commonly used metrics for the response to $CO_2$ forcing, based on the evolution of the simulated global-mean temperature, are the Equilibrium Climate Sensitivity (ECS), the Transient Climate Response (TCR), and the Transient
Climate Response to cumulative $CO_2$ Emissions (TCRE). Their values are given in table 1 for the NorESM2 experiments, and compared to those for NorESM1. The ECS is defined as the change in global near-surface temperature when a new climate equilibrium is obtained with an atmospheric $CO_2$ concentration that is doubled compared to the pre-industrial amount. In order to reach a new equilibrium, a model simulation of several thousand years is required (Boer and Yu, 2003). There are some





examples in the literature of models for which this has been done, but in general ECS is more commonly estimated from

the relationship between surface temperature and RESTOM from the abrupt4×$CO_2$ experiment using the so-called Gregory method (Gregory et al., 2004). The numbers in table 1 are calculated using years 1–150 from the simulations shown in Fig. 3, and are divided by 2 to get the number for $CO_2$ doubling instead of quadrupling. The ECS is 2.54 K for NorESM2-LM, which is slightly lower than the equivalent value for NorESM1 of 2.8 K. Both are significantly lower than the CMIP5 mean value of 3.2 K but well inside the bounds of the likely range of 1.5–4.5 K (Stocker et al., 2013). On the other hand, the ECS in NorESM2

is markedly smaller than the ECS found in CESM2 of 5.3 K by Gettelman et al. (2019a), despite sharing many of the same component models. The ECS in NorESM2 is discussed in more detail in Gjermundsen et al. (in prep.). There are indications that the different behaviour of the BLOM ocean model (compared to the POP ocean model used in CESM2), contributes to a delayed warming in the first 150 years of abrupt-4xCO2 in NorESM2. Using the Gregory et al. (2004) method on that period leads to an ECS estimate which is considerably lower than for CESM2. However, after the initial slow warming in the abrupt-

4xCO2 experiment, NorESM2 shows a sustained warming similar to CESM2, when the abrupt-4xCO2 experiment is extended to 500 years or longer. This suggests that the actual ECS (the value one finds when the model is run for thousands of years until equilibrium) in NorESM2 and CESM2 is not very different, but that the Gregory et al. (2004) method based on the first 150 years only does not give a good estimate of ECS for models.

The TCR is defined as the global-mean surface temperature change at the time of $CO_2$ doubling, and accordingly it was

calculated from the temperature difference between the 1pctCO2 experiment averaged over years 60–80 after initialisation and piControl. The TCR is 1.48 K and 1.33 K for NorESM2-LM and NorESM2-MM, respectively. As for ECS, these values fall in the lower part of the distribution obtained from the CMIP5 ensemble (Forster et al., 2013), similar to those obtained for NorESM1. A recent observational estimate for the 90 % likelihood range of TCR is 1.2–2.4 K (Schurer et al., 2018).

We also give an estimate of the transient climate response to cumulative carbon emissions (TCRE) calculated from TCR

and the corresponding diagnosed carbon emissions. Following Gillett et al. (2013), TCRE is defined as the ratio of TCR to accumulated $CO_2$ emissions in units of K EgC$^{-1}$. As $CO_2$ fluxes were not calculated in NorESM1-M and NorESM1-Happi, the NorESM1 values are obtained from the carbon cycle version of NorESM1 (NorESM1-ME; Tjiputra et al., 2013). TCRE is reduced from 1.93 K EgC$^{-1}$ in NorESM1-ME to 1.36 K EgC$^{-1}$ and 1.21 K EgC$^{-1}$ in NorESM2-LM and MM, respectively. Since TCR is comparable, the main difference is due to changes in carbon uptake. NorESM1, with CLM4 as the

land component, had a very strong nitrogen limitation on land carbon uptake. This limitation is weaker in CLM5 (Arora et al., 2019) used in NorESM2.

### 4.4 Climate evolution in historical and scenario experiments

In this section we provide a very brief analysis of the response of the model to historical forcings in the three historical members carried out with NorESM2-LM and the one realisation carried out with NorESM2-MM. We also consider the model

response for the Tier 1 experiments from ScenarioMIP (SSP1-2.6, SSP2-4.5, SSP3-7.0, and SSP5-8.5). The focus here will be on the response in global-mean near-surface temperature, the Atlantic Meridional Overturning Circulation (AMOC), the volume transport through the Drake Passage, and on sea ice area.



Figure 4 shows the time evolution of the surface atmospheric temperature in the historical simulations from NorESM2-LM
and NorESM2-MM along with observations. Both versions of NorESM2 follow the observations rather closely for the first
80 years. After 1930 the model displays somewhat weaker warming than the observations until around 1970. After that the
rate of the warming in the models are similar to that seen in the observations. The cooling over the period 1930–1970 in
NorESM2 is probably caused by the combination of a low climate sensitivity (see Sect. 4.3) and a strong negative aerosol
forcing. Atmosphere-only simulations with NorESM2-LM (see Olivié et al., in prep.) show that the aerosol forcing strengthens
from around -0.3 W m$^{-2}$ around 1930 to -1.5 W m$^{-2}$ in the period 1970–1980, becoming slightly weaker again in 2014 with
a value of -1.2 W m$^{-2}$. On a global scale anthropogenic $SO_2$ emissions have risen strongly in the period 1950–1980, and
these are assumed to contribute most to the anthropogenic aerosol forcing. Although no such experiments have been done
with NorESM2-MM yet, it is likely that the aerosol forcing is similar in both model versions. Note also that our choice of
the reference period for temperature anomaly computation (1850-1880) enhances the NorESM2 negative bias with respect to
observations in the last half of the 20th century.

Figure 5 shows again the evolution of the surface air temperature in the historical simulations (only the first ensemble member
for NorESM2-LM), followed by the temperature evolution under the four SSP scenarios for NorESM2-LM and NorESM2-
MM. Compared to the 1850–1879 period, the model shows a warming in 2005–2014 of 0.72 and 0.54 K for NorESM2-LM
and NorESM2-MM, respectively. Under the four scenarios SSP1-2.6, SSP2-4.5, SSP3-7.0, and SSP5-8.5, the warming in the
period 2090–2099 compared to 1850–1879 reaches 1.30, 2.15, 2.95, and 3.94 K in NorESM2-LM, and 1.33, 2.08, 3.06, and
3.89 K in NorESM-MM. Although the historical warming is slightly weaker in NorESM2-MM compared to NorESM2-LM,
the warming at the end of the 21st century is rather similar in both versions of NorESM2. For SSP1-2.6, the temperature
stabilizes in the second half of the 21st century. In NorESM1, under the scenarios RCP2.6, RCP4.5, and RCP8.5, the surface
air temperature in the period 2071–2100 was 0.94, 1.65, and 3.07 K higher than in 1976–2005 (Iversen et al., 2013). For the
same periods and looking at SSP1-2.6, SSP2-4.5, and SSP5-8.5, we find rather similar (but slightly stronger) warmings of 1.06,
1.81, and 3.22 K in NorESM2-LM, and 1.11, 1.83, and 3.26 K in NorESM2-MM.

The simulated Atlantic Meridional Overturning Circulation (AMOC) at 26.5° N shows a multi-centennial variability that is
15% of the mean in the control simulation (Fig. 2). In the historical simulations the AMOC peaks for both MM and LM in
the 1990's at around 24 Sv before starting a rapid decline at around year 2000 (Fig. 6). In both versions the AMOC reaches a
quasi-equilibrium by the end of the century at around 15-10 Sv depending on the scenario. Since we only have a few ensemble
members, it remains unclear how fast the AMOC declines in response to the greenhouse gas forcing and which part of e.g. the
initial decline is due to the multi-decadal variability. In any case, it is noteworthy that the initial AMOC decline begins already
during the historical period in both versions, which is also consistent with the NorESM2 and multimodel mean response to the
OMIP2 forcing (1958-2018, Tsujino et al., 2019).

In addition to the AMOC, also the Antarctic Circumpolar Current (ACC) strength, as measured in the Drake Passage, shows
multi-centennial variability that is about 3% of the mean (Fig. 7). Similar variability in the ACC has been linked to convection
within the Weddell and Ross seas in the CMIP5 ensemble (Behrens et al., 2016). Also in our simulations the Weddel Sea
convection has similar long term variability as the ACC. Unlike the AMOC, there is no clear trend emerging from the scenario



simulations, but rather the multidecadal variability continues throughout the 21st century. Again, a larger number of ensemble members could help identify the forced signal.

The time evolution of Northern Hemisphere sea ice area (March and September) through the historical and scenario periods is shown in Fig. 8. Both model versions are compared with the sea ice area from OSISAF (OSI-V2.0) reprocessed climate data record (Lavergne et al., 2019) for the years 1979–2019. The total sea ice area from NorESM2-LM compares rather well with the observations, while NorESM2-MM has too much ice, especially during summer. The trend in sea ice area found in the observations during summer is also rather well captured by NorESM2-LM, while this trend is too small in NorESM2-MM.
Both models have a reasonable March sea ice area compared to observations. However, the negative trends in winter sea ice area are small compared to observed trends.

During the scenario period both models show a strong reduction in summer sea ice area. The Arctic Ocean is often considered ice free when the total sea ice area drops below 1 million square km. This threshold is denoted by dotted gray lines in Fig. 8. NorESM2-LM loses summer ice shortly after year 2050. This occurs first in the SSP5-8.5 scenario, but also the SSP2-4.5
ensemble shows values close to this threshold even before 2050. SSP3-7.0 scenarios become ice free at around 2070. Any prediction of which year the Arctic Ocean first becomes ice free must therefore be considered rather uncertain due to forcing evolution uncertainty and internal variability. This is consistent with the overall assessment of sea ice evolution in CMIP6 assessed by the SIMIP Community (Notz et al., 2020). In NorESM2-LM an ice free Arctic Ocean is only avoided in the SSP1-2.6 scenario. NorESM2-MM loses ice slower and shows the first ice free summer around 2070. In that model, also the
SSP2-4.5 scenario keeps the ice area above 1 mill. square km all years before 2100. However, the SSP1-2.6 scenario stabilizes at a sea ice area comparable with present day observations, even with SSP1-2.6 warming levels present. Therefore, the sea ice area simulated by NorESM2-MM for the future Arctic seems to be unrealistically high.

## 5   Climatological mean state and circulation patterns compared to observations and NorESM1

### 5.1   Ocean state

In the surface ocean, the large-scale climatological biases are similar in the two NorESM2 versions (Fig. 9), but overall the MM version is closer to the observations (smaller global-mean root-mean-square error; RMSE; $\sqrt{A^2}$ in Fig. 9). In general the Southern Ocean is too warm (Fig. 9b-c), the Atlantic (and the Arctic) are too saline, but the Pacific is too fresh (Fig. 9e-f). The sea level is lower than observed in the Atlantic basin, but higher in the Indo-Pacific basin and thus the gradient between the two basins is larger than in the observations (Fig. 9h-i). If we remove the global-mean biases, the two versions produce even more
similar mean errors, suggesting that some of the regional biases are largely independent of the atmosphere and land resolution.

Indeed, the regional patterns are common to many other models with coarse resolution ocean components (Wang et al., 2016). Both NorESM2 versions are too warm and (relatively) saline over the western boundary currents (the Gulf Stream and the Kuroshio in the Northern Hemisphere and the Brazil current and the Agulhas current in the Southern Hemisphere) and over the major eastern boundary upwelling systems (Canary, Benguela, Humboldt, and California). The biases over the western
boundary currents are due to the errors in the location of the currents, which are linked to the ocean model resolution (Bryan





et al., 2007; Saba et al., 2016; Rackow et al., 2019). The ocean-model resolution also explains two well known biases in the North Atlantic also seen in NorESM2: the southern bias in the Gulf Stream/North Atlantic current path causes the cold (and fresh) bias in the subpolar North Atlantic (Bryan et al., 2007; Saba et al., 2016; Rackow et al., 2019), while the lack of the Labrador Current waters on the east coast of North America causes a large warm and saline bias there (Saba et al., 2016).

While the above mentioned biases are mostly linked to the ocean model, in the Pacific there are biases that are not present in the ocean-only simulations (not shown). Specifically, a fresh bias over the Southern Pacific subtropical gyre and cold biases over the northern Pacific subtropical gyre and the equatorial Pacific.

The fresh bias in the Southern Pacific is linked to the co-located positive net precipitation bias (Fig. 9) and extends throughout the surface mixed layer (Fig. 10). The salinity bias also causes a negative density bias (not shown) as it is not fully compensated

by temperature, supporting an atmospheric origin. A comparison with the OMIP1 and OMIP2 simulations shows that the net precipitation bias in the LM simulation, $250\,\mathrm{mm\,year^{-1}}$ in the mean over the region where the salinity bias is larger than $1\,\mathrm{g\,kg^{-1}}$, would be large enough to cause the simulated salinity bias (assuming mixed layer depth of $100\,\mathrm{m}$ and a residence time of 10 years). Therefore, we suggest that the net precipitation bias leads to accumulation of excess freshwater that is spread throughout the subtropical gyre by the ocean circulation.

Most of the large-scale surface biases are also visible in the subsurface (Figs. 10–11). The upper ocean is too warm and fresh, while the deep ocean is too cold and saline. The biases are again larger in the LM version. The cold deep ocean is due to the cold bias in the Antarctic bottom water, while the warm bias in the mid-depth Atlantic (between $500$–$3500\,\mathrm{m}$) is due to the Antarctic Intermediate Water and the North Atlantic deep water being too warm. There are also subsurface biases without a large surface signature. The Mediterranean outflow and the Red Sea outflow form too warm and saline cores visible at around

$20°\,\mathrm{N}$ and $1000\,\mathrm{m}$ depth in the Atlantic and Indian oceans (respectively, Figs. 10–11). These biases are stronger in the LM version and not visible or much less pronounced in the OMIP simulations (not shown), which suggests that they are due to biases in the surface heat and freshwater budgets in these semi-enclosed basins. In addition, there is a strong cold and fresh (warm and saline) bias in the Pacific (Atlantic) centered around $15°\,\mathrm{S}$ and $200$–$400\,\mathrm{m}$ depth. These anomalies are likely linked to the biases in the tropical upwelling and the resulting thermocline depth that is too shallow (deep) in the Pacific (Atlantic).

Overall, many of such sub-surface ocean biases are similar in the ocean-only simulations and may be linked to coarse ocean resolution and shortcomings in parameterised processes. In some regions, air-sea coupling tends to act to reinforce biases that may be generated in either atmosphere or ocean model components separately. The biases over the upwelling systems for example have generally a complex cause rooted in both local (including mesoscale) and remote (including equatorial) biases in both atmosphere and ocean model components (Toniazzo and Woolnough, 2014; Zuidema et al., 2016; Stammer et al., 2019). For

NorESM2 the biases in the coupled simulations have a similar pattern as, but approximately twice the magnitude of the biases in the OMIP simulations (not shown). The cold bias in the northern subtropical Pacific has a contribution from weak oceanic mixing as there is a large warm bias just below the surface (Fig. 10), but may be amplified by increased atmospheric stability and correspondingly enhanced boundary-layer clouds. Excessively negative short-wave cloud forcing is seen in that region, in contrast to AMIP simulations which show no such regional bias. In the central and eastern equatorial Pacific NorESM2 displays

a characteristic "cold tongue" bias with cold SSTs and easterly wind stress bias. An equatorial easterly bias is present in the




NorESM2 AMIP simulations. Shonk et al. (2017) show that off-equatorial net precipitation biases alone can initiate a feedback leading to an equatorial Pacific cold tongue in coupled simulations, and CAM6-Nor tends to develop such a bias. Finally, the near-surface ocean temperature bias pattern in OMIP1 simulations is cold along the equator, and warm on each side, which may further enhance off-equatorial precipitation. It should be noted that OMIP2 simulations with BLOM/CICE have a warm
bias along the equator (Tsujino et al., 2019). The cold equatorial bias can affect ENSO variability and teleconnections. These are discussed further below.

## 5.2 Sea ice

The geographic distribution of sea ice in March and September, compared with observations are shown in Fig. 12 for NorESM2-LM (12e-h), and NorESM2-MM (12i-l). In common for both models for the Northern Hemisphere (12e,f,i,j) are too large sea
ice extents in the Barents Sea and Greenland Sea and a too small extent in the Labrador Sea, Bering Sea, and Sea of Okhotsk during winter. The total areas are quite close to the observations as shown in Fig. 8. These regional biases are most likely due to persistent biases in the oceanic and atmospheric circulation.

During summer, the distribution of sea ice in NorESM2-LM (Fig. 12f) seems to be more variable. Apart from the persistent, positive bias in the East Greenland Current, the regional biases within the Arctic Ocean are more likely due to inter-annual
variability, and the effect that the observations show a larger downward trend than the model.

NorESM2-MM (Fig. 12j) shows too much sea ice in the central Arctic in September. In general, the model is colder in the Arctic than NorESM2-LM (Fig. 14), and it has thicker sea ice in the Arctic Ocean. The Northern Hemisphere sea ice volume in NorESM2-MM is 19–23% (38–60%) larger in March (September) compared with the NorESM2-LM (not shown). The smaller seasonal cycle in ice area (Fig. 13) and volume is consistent with a thicker sea ice cover in NorESM2-MM, both due
to less winter growth because of increased insulation, and less summer melt due to higher albedo. The situation encountered in NorESM2-MM is similar to the results from NorESM1-M (Bentsen et al., 2013) and NorESM1-Happi (Graff et al., 2019). These models simulate ice cover that is too thick, with the reduction in the Northern Hemisphere summer ice area being too slow.

The winter sea ice area and extent is too low in the Southern Ocean in NorESM2 as seen in Fig. 13 and Fig. 12(g-h,k-l).
Winter area in September is around 4 million square km too small. The largest bias is found in the Atlantic-Indian sector. This bias seems to be associated with the warm bias in the ocean model, and the too warm intermediate Antarctic water (AAIW). The exact reason for this problem is not known, but the warm bias in AAIW is also evident in the OMIP simulations (not shown). However, these uncoupled simulations have a reasonable representation of the upper ocean temperature and the winter sea ice extent that are most likely due to the inherent relaxation towards observed atmospheric temperatures in those experiments.
With the interactive atmosphere these problems increase.

## 5.3 Atmospheric temperature and winds

NorESM2 is a warmer model than its preceding versions. The global-mean near-surface temperature (Fig. 14) in NorESM1-M and NorESM1-Happi is generally too low with global-mean biases of -0.62 K and -0.94 K (see legends above panels in Fig. 14).





NorESM2-MM is closer to the reanalysis with a global-mean bias of -0.05 K. Regionally, cold biases are mostly found in the
polar regions and over the sub-tropical oceans. Warm biases are found over the Southern Ocean, North Atlantic and in central
Eurasia. NorESM2-LM (panel a) is warmer still, and overestimates the near-surface temperatures in the Arctic and in the
global-mean, with a bias of 0.58 K. NorESM2-MM has the best overall performance also in terms of the global-mean RMSE,
with 1.33 K compared to 1.76 K for NorESM2-LM, and 1.71 K for NorESM1-Happi, and 1.79 K for NorESM1-M (cf Fig. 14).

Temperature biases are mitigated in NorESM2 compared to NorESM1, not only near the surface, but also and especially
in the mid and upper troposphere (Fig. 16). In particular NorESM2 has a reduced cold bias compared to NorESM1 partic-
ularly in the tropics and sub-tropics. This is mostly a consequence of the changes made to the cumulus convection scheme
(Toniazzo et al. in prep.). NorESM2-LM being generally warmer in the tropics than NorESM2-MM, its cold biases there are
smaller; however persistent cold mid- and high-latitude biases imply an excessive meridional temperature gradient. By contrast,
NorESM2-MM shows improvements at all latitudes.

All four of NorESM2-MM, NorESM2-LM, NorESM1-Happi and NorESM1-M tend to produce westerly biases in zonal-
mean zonal winds (Fig. 17). At tropical and sub-tropical latitudes, these are more widespread in NorESM2 than NorESM1-M
and NorESM1-Happi, and at the same time the easterly surface biases are mitigated. At higher latitudes, all models tend
to have westerly biases on the poleward side of the sub-polar surface jet (between 50° and 60°) in both hemispheres. The
overestimation on the poleward flank is generally more pronounced in NorESM2 than in NorESM1. Comparing NorESM1-M
to NorESM1-Happi and NorESM2-LM to NorESM2-MM, the biases in the zonal wind tend to be ameliorated with increased
resolution. The differences in the tropics between NorESM2 and its predecessors is in part attributable to the enforcement
of conservation of atmospheric global angular or rotational momentum in NorESM2 (Toniazzo et al., 2019). In all versions,
in common with CAM6/CESM2, there is accumulation of westerly momentum near the model lid, where it is insufficiently
damped.

## 5.4   Extratropical storm tracks

Extratropical storm tracks can be defined as regions of storminess associated with cyclogenesis, cyclone development, and
cyclolysis which take place in the baroclinic zones between the sub-tropics and polar regions. They are important features
at mid- and high latitudes as they are responsible for eddy transport of heat and momentum between low and high latitudes,
and associated with potentially high-impact weather such as heavy precipitation and strong winds. Here, we diagnose storm-
track activity by applying a bandpass filter to retain fluctuations in the geopotential height field at 500 hPa with periodicity
corresponding to that of baroclinic waves, that is, between 2.5 and 6 days (Blackmon, 1976; Blackmon et al., 1977). The
variability of the bandpass-filtered field is dominated by propagating low-pressure and high-pressure systems, and the storm
tracks can be defined as geographically localized maxima in bandpass-filtered variability (Blackmon, 1976; Blackmon et al.,
1977; Chang et al., 2002; Graff and LaCasce, 2012).

The climatological winter storm tracks are shown as the solid black contours in Fig. 18. There are two maxima in the
Northern Hemisphere, one over the North Atlantic and one over the North Pacific. The colors show the bias with respect
to ERA-Interim (Dee et al., 2011). In NorESM1-M, storm-track activity is underestimated in both storm-track regions. In





particular, the North-Atlantic storm track is overly zonal with too little activity on the equatorward side of the climatological maximum as well as over the Norwegian and Barents Sea (Iversen et al., 2013; Graff et al., 2019). The magnitude of the bias is reduced in NorESM1-Happi compared to NorESM1-M in both storm-track regions. This is likely associated with the increased resolution in the atmosphere and land components (1° in NorESM1-Happi versus 2° in NorESM1-M).

Similar improvements are seen when comparing NorESM2-LM and NorESM2-MM. Both versions of NorESM2 are, furthermore, better able to simulate the North-Atlantic storm track with the size of the negative bias on its equatorward side being reduced. Overall, NorESM2-MM displays the smallest biases in Northern Hemisphere storm-track activity out of the four models. There remains, however, too little activity over the Norwegian Sea with extension into the Barents Sea.

In the Southern Hemisphere, the climatological winter storm track surrounds Antarctica with the largest variability occurring over the Indian Ocean (Fig. 18). Storm-track activity is generally too weak on the equatorward side, with the largest biases being located over the Indian Ocean, close to the storm-track maximum. As in the Northern Hemisphere, the largest biases are found in NorESM1-M and the smallest biases in NorESM2-MM.

While the bandpass-filter approach yields a measure of storm-track activity, it cannot be used to isolate the individual cyclone centers. To further assess the robustness of the improvements between NorESM2-LM and NorESM2-MM, we therefore also consider results from the cyclone detection algorithm described in Wernli and Schwierz (2006). The method detects cyclones as minima in the sea-level pressure fields and sets the perimeter as the outermost closed sea-level pressure contour. The storm tracks are then seen as maxima in the local frequency of occurrence of surface cyclones, i.e. the fraction of time when cyclones are present in a given point (Fig. 19a–b).

As for the bandpass-filter approach, the cyclone detection shows a clear reduction in the bias between NorESM2-LM and NorESM2-MM, which is likely to be associated with the higher horizontal resolution in the atmosphere and land components. The cyclone occurrence is underestimated on the equator-ward side of the North Pacific and Southern Hemisphere storm tracks and overestimated on the poleward side. Over the North Atlantic, the cyclone occurrence is underestimated on the equator-ward side of the storm track and over the Norwegian Sea extending into the Barents Sea, and overestimated between The British Isles and Greenland. The magnitude of the bias is clearly reduced in all regions in NorESM2-MM, with the improvement being particularly evident in the regions where the cyclone occurrence is overestimated.

Note that both the climatology and the biases should be expected to differ somewhat between the two approaches considered here because they capture different aspects of the storm tracks. The bandpass-filter approach does not distinguish between cyclones and anti-cyclones, and is dominated by growing and propagating baroclinic waves (Blackmon et al., 1977). The cyclone occurrence reflects the regions where cyclone centers are identified most frequently, and is for instance more sensitive to systems that are slow moving or too long lived.

## 5.5 Clouds and forcing

Table 2 gives an overview of major forcing fluxes in NorESM2 compared to NorESM1 and observational estimates. Despite the large differences in physics and tuning, the overall numbers for top of the atmosphere fluxes and forcings are very similar to the numbers found in NorESM1-Happi and are generally within the observational range. There is however a slightly stronger





negative bias in clear-sky LW flux and long wave cloud forcing. The latter is an unfortunate consequence of the tuning of high clouds in the model implemented in order to increase the outgoing long wave radiation. As seen from the upward LW flux estimate the outgoing long wave radiation is still within the estimate from satellite retrievals. SWCF values are very

similar to the values of NorESM1-Happi and within the observational range. This number hides, however, a major weakness in NorESM1 stratiform cloud parameterisation which underestimated the cloud cover and compensated this by overestimating the cloud liquid water.

The major updates in cloud physics from CAM4 to CAM6 (Bogenschutz et al., 2018) improved the cloud cover, and the cloud liquid water path is now quite close to the observational estimate. The global cloud cover is still slightly lower than

observed. This is partly connected to the tuning in NorESM2. Prior to the tuning the modelled cloud cover was higher than 70 %. As seen from Fig. 15, the cloud cover underestimate is most pronounced in the tropics and subtropics in both hemispheres, while there is good agreement around the extra-tropical stormtrack regions and an overestimate in the high Arctic. Before the tuning (not shown) there was no bias at the low latitudes.

The modelled liquid water path seems to have a systematic bias towards low values at low latitudes and high values in the

extra-tropics. Possible connections between cloud cover biases and the hydrological cycle are discussed in the next section.

## 5.6 Precipitation and hydrological cycle

The bias in the annual-mean total precipitation rate is shown in Fig. 20 for the two versions of NorESM1 and NorESM2 relative to the ERA-Interim re-analysis. along with climatology from the Global Precipitation Climatology Project (GPCP; Adler et al., 2003). While the bias of the global-mean average is not systematically reduced between NorESM1 and NorESM2, there is a

reduced RMSE, indicating that there is less cancellation between positive and negative biases in the global mean.

The reduction of the RMSE is also seen when considering the four seasons separately in Fig. 21 along with climatology from the GPCP. The evaluation of the mean bias, RMSE, and correlation included in the bottom left corner of each panel shows that RMSE and correlation have improved in NorESM2 compared to NorESM1 for all seasons. While the overall wet bias has increased slightly, mostly due to strong biases over the Pacific ocean, there are regions with a large reduction in mean bias.

This is especially pronounced over Africa and equatorial Atlantic ocean. The largest improvement compared to NorESM1-M is seen for NorESM2-MM during northern hemisphere winter, when all three metrics (bias, RMSE, and correlation) consistently indicate higher skill.

As a measure of interannual variability, the standard deviation of monthly means for each season was calculated. The differences compared to GPCP are presented in Fig. 22. While NorESM1 slightly underestimates the precipitation variability,

it is somewhat too high in NorESM2, with the magnitude of the bias being larger in all seasons except DJF and SON in NorESM2-MM. As seen for the mean climatology in Fig. 21, the correlation has improved for all seasons in both NorESM2-LM and MM.

The hydrological cycle (or cycling of fresh water) is of major importance for the climate system. Global means of precipitation and evaporation can serve as integrated measures of the properties of many processes in an earth system model. Results

presented in Table 3 indicate that the intensity of hydrological cycle, as measured by evaporation, in NorESM2 is about 1.1%





larger globally (4.1% over oceans) than in NorESM1-M. This is also manifested in the positive precipitation biases in Figs. 20 and 21. While the values in Table 3 for NorESM2 are higher than for GPCP, they are closer to results from ERA-Interim calculated by Trenberth et al. (2011a). Although NorESM1-Happi has the highest precipitation globally, NorESM2 has the highest precipitation over ocean, suggesting a larger re-cycling of oceanic water vapor and a lower fraction transported from oceans to

continents (measured by E-P over oceans). The overestimated evaporation over oceans is likely linked to the underestimated cloudiness in the tropics and subtropics (see discussion above about Fig. 15 ). Solar radiation over subtropical ocean regions is an important driver of evaporation. The net moisture transport from oceans to continents is nevertheless smaller in NorESM2 than in NorESM1, consistent with more clouds in the extra-tropics and more marine precipitation in NorESM2. This analysis is only preliminary, however, and needs more in-depth studies which is out of scope of the present paper.

In the NorESM2 earth system model a closed hydrological cycle is present, with the difference between evaporation and precipitation being close to zero in the long-term average at equilibrium. In NorESM2-MM the discrepancy is only 0.001 $km^3$/year, whereas it is 0.027 $km^3$/year in NorESM1-M and 0.016 $km^3$/year in NorESM2-LM (means from members 1–3).

### 5.7   Northern Hemisphere blocking

While storm tracks are closely tied to precipitation, atmospheric blocking is associated with persistent anti-cyclones that inhibit

precipitation for time scales up to several weeks. To diagnose blocking, we apply the variational Tibaldi and Molteni (vTM) index (Tibaldi and Molteni, 1990; Pelly and Hoskins, 2003; Iversen et al., 2013; Graff et al., 2019). Blocks are identified when there is persistent reversal of the 500 hPa geopotential height field around a central latitude that last for at least five days and cover at least 7.5 consecutive longitudes. The central longitude varies with the position of the maximum in the Northern Hemisphere climatological storm track.

The seasonal blocking frequency is mostly underestimated over the North Atlantic and in Europe in the four versions of NorESM (Fig. 23), particularly during winter (DJF). During spring (MAM), NorESM2-MM is closest to the reanalysis, while during summer (JJA) and autumn (SON), NorESM1-Happi performs best in these regions. While NorESM1 tends to overestimate the blocking frequency over the Pacific, NorESM2 generally lies closer to the reanalysis in that sector. Consider, for instance, the region between $120°$ E and $180°$ E during summer, or the region between $130°$ W and $90°$ W during winter. In

summary, although the use of 30 years from ERA-Interim for verification may not be fully representative for blocking climatology, the representation of NH blocking continues to be a challenge in NorESM, and in particular over the Atlantic-European sector in winter.

### 5.8   Madden-Julian Oscillation

In the tropical atmosphere, the Madden-Julian Oscillation (MJO) is the dominant mode of variability on timescales between

30 and 90 days (Madden and Julian, 1971; Zhang, 2005). The MJO is characterized by large-scale regions of enhanced and suppressed convection that slowly propagate eastwards along the equator, and interacts with a number of other circulation features such as El Niño events (Hendon et al., 2007), the Indian summer monsoon (Annamalai and Slingo, 2001), tropical cyclones (Liebmann et al., 1994), and even the North Atlantic Oscillation and extratropical variability (Cassou, 2009).





We diagnose the MJO in two ways. One is in terms of temporal correlations between subseasonally filtered anomalies of precipitation and winds along the equatorial Indian ocean. The second is in terms of wavenumber-frequency spectrum for 850 hPa zonal wind (U850) and for outgoing longwave radiation (Fig. 24). These diagnostics have been proposed and described in detail in Waliser et al. (2009).

Positive wavenumbers and frequencies indicate eastward propagation, while negative frequencies (or wavenumbers) indicate westward propagation. The energy in the spectrum for U850 from ERA-Interim (Fig. 24a) shows that the energy in the reanalysis is associated with wavenumbers 1–3 with a maximum for wavenumber 1, and with the energy being more or less contained within timescales of 30 to 80 days. NorESM2-LM and NorESM2-MM also show maximum energy for the same wavenumbers, with the maximum occurring for wavenumber 1, as in ERA-Interim. The maximum is, however, somewhat too strong in both models and the energy is spread out over a wider range of timescales. Both NorESM2-MM and NorESM2-LM peak at longer timescales (lower frequency) than ERA-Interim, and NorESM2-LM has an additional peak at shorter timescales (higher friquency). Similar results are found when comparing the wave-number frequency spectra for outgoing longwave radiation from NorESM2-LM and NorESM2-MM with that from NOAA, however here the peak energy is underestimated.

Lead-lag correlations with respect to precipitation anomalies during extended winter in the equatorial Indian Ocean around 90E are characterised in observations (Figure 25(a)) by a marked, slow eastward propagation and some poleward propagation. There is a strong relationship with zonal winds in quadrature such that westerly wind anomaly maxima precede the precipitation maxima. These characteristics are simulated fairly clearly in NorESM-LM (Figure 25(b)), although there is little propagation into the Indian Ocean from the West, or poleward propagation. The MJO in NorESM-MM (Figure 25(c)) has in general similar properties, but it is too weak in amplitude. Composite plots of MJO events (not shown) indicate a tendency in both model versions to generate westward-propagating convective anomalies, which may weaken activity in the MJO region of the spectrum.

## 5.9 El Niño Southern Oscillation (ENSO)

The coupled model internally generates a self-sustained ENSO mode with spatial and temporal characteristics similar to observations. The timeseries of NINO3.4 SST anomalies are shown in Figure 26, alongside the observed one. The ENSO in LM and MM model versions are very similar in magnitude (Figures 26, 28, 29), spatial pattern (Figure 29), and spectral power distribution in frequency space (Figure 27). ENSO SST anomalies are very large compared to observations (with a NINO3.4 anomaly greater than 2.5° C in the average El-Niño event, compared with 1.5° C in observations), and they tend to peak early in the season, i.e. between November and December instead of between December and January as observed (Figure 28a). The early peak and termination may be partly attributable to weak zonal wind-stress anomalies over the equatorial region, which also peak early, notwithstanding a robust response in equatorial precipitation (Figure 28b,c). Such weak surface wind response may be caused by the general displacement, with respect to observations, of the maximum of climatological precipitation north of the Equator along the Pacific ITCZ. Especially in MM, precipitation anomalies also have their maximum north of the equator (Figure 29b), which tends to result in weaker equatorial anomalous westerlies. A second origin of the early simulated El-Niño SST peak may however also be found in the early rapid demise of positive thermocline depth anomalies in the NINO3 region





during El-Niño events, which is seen also in OMIP1 and OMIP2 simulations forced with prescribed wind-stress (Figure 28d). Given the weak coupled wind-stress and thermocline activity, the large SST anomalies may be partly the result of insufficient surface damping by the action of anomalous surface heat fluxes.

Correlation analysis shows that indeed over the eastern equatorial Pacific the model tends to generate positive downward net short-wave radiative flux anomalies when SST anomalies are positive, in contrast to observations. This might also explain the growth of positive SST anomalies in the NINO3.4 region early during El-Niño events even before positive $20°$ C isotherm depth anomalies have fully reached the area; and the long persistence of both SST and precipitation anomalies in the later stages of El-Niño events. The model climatological bias of a pronounced double ITCZ, with strong ITCZ precipitation away from the equator and a dry, cold equatorial region dominated by marine stratocumulus, rather than trade-cumulus cloud in the eastern Pacific probably contributes to this behaviour. Toniazzo et al. (in prep.) shows that changes in the convection scheme that were made in order to mitigate the tropospheric cold bias and the positive TOA net residual have contributed to this error. Off-equatorial precipitation tends to couple with westward-propagating equatorial modes and can lead to a tendency for westward propagation of convective activity (cf. Figure 24). Westward propagation is also evident in the model's ENSO during the phase change from El-Niño to La-Niña. In spite of such shortcomings, ENSO-related variability in NorESM2 is generally similar to the observed one. In particular, NINO3.4 spectra in the two model versions and in observations are formally indistinguishable (Figure 27). The simulated composite El-Niño SST, precipitation, and geopotential height anomalies have good global pattern correlations with the observed composite El-Niño anomalies (Figure 30), indicating that the simulated ENSO adds important and useful features to the climatology simulated by the model. Particularly prominent and fairly realistic are teleconnections into both hemispheres during and after ENSO peaks, with a PNA pattern that extends into the storm-track entry region of the western Atlantic, as observed. In this respect NorESM2-MM validates better than NorESM2-LM, in spite of its equivalent of slightly worse equatorial ENSO biases, probably due to a better overall sub-tropical and high-latitude atmospheric circulation.

## 6  Summary and discussion

This paper presents and evaluates NorESM2 (the second version of the Norwegian Earth System Model) used for conducting experiments for CMIP6. NorESM2 is based on CESM2 (the second version of the Community Earth System Model), but with several important differences. While the land and sea ice components are largely the same as in CESM2, NorESM2 has entirely different models for the ocean and ocean biogeochemistry, namely BLOM and iHAMOCC. There are also several differences in the atmosphere model (CAM6-Nor), including a different module for aerosol life-cycle, aerosol-radiation-cloud interactions and with changes related to the moist energy formulation, deep convection scheme and angular momentum conservation. Finally, the turbulent air-sea flux calculations are modified and proper time-averaging of solar zenith angle in albedo estimation implemented.

We report results from the CMIP6 DECK experiments, including the pre-industrial control, the abrupt quadrupling of $CO_2$ concentration levels, and 1% increase per year of $CO_2$ concentrations until quadrupling, along with the CMIP6 historical experiments, and the ScenarioMIP Tier 1 experiments (SSP1-2.6, SSP2-4.5, SSP3-7.0, and SSP5-8.5). The experiments were





all carried out with both a medium-resolution version of the model (NorESM2-MM) and a low-resolution version (NorESM2-LM).

The drift over the 500 year long pre-industrial control experiment is generally very small for both versions of the model. NorESM2 in both model resolutions is slightly less sensitive than its predecessors and at the lower end of the CMIP5 and CMIP6 multi-model mean (preliminary calculations, Gjermunden et al, in prep), with the equilibrium climate sensitivity of 2.5 K estimated using the Gregory method (Gregory et al., 2004).

The historical reconstruction of surface temperatures is similar in both model versions. A significant temperature increase due to enhanced climate forcing is setting in late in the historical period. Both model versions reach present day warming levels to within $0.2°$ C of observed temperatures in 2015. Aerosol forcing may be responsible for the delayed warming in the late 20th centuray. Aerosol effective radiative forcing reaches levels of -1.5 W m$^{-2}$ in the period 1970–1980, becoming slightly weaker again in 2014 with a value of -1.2 W m$^{-2}$.

Under the four scenarios SSP126, SSP245, SSP370, and SSP585, the warming in the period 2090–2099 compared to 1850–1879 reaches 1.3, 2.2, 3.0, and 3.9 K in NorESM2-LM, and 1.3, 2.1, 3.1, and 3.9 K in NorESM-MM, robustly similar in both resolutions.

In particular NorESM2-LM shows a satisfactorily evolution of recent sea ice area. In NorESM2-LM an ice free Arctic Ocean is only avoided in the SSP1-26 scenario. NorESM2-MM simulates higher sea ice area both at present and in future scenarios.

The pattern of some biases seen in the fully coupled simulations considered here are similar in coupled ocean-sea ice simulations carried out for OMIP and can thus be linked to the ocean model having too coarse resolution and shortcomings in parameterised processes. NorESM2-LM and MM largely share the same biases in the surface ocean, although the MM version is somewhat closer to the observations. Most of the large-scale biases in the surface ocean are also seen in the subsurface.

Like CESM2, NorESM2 is generally a "cold" model, with an initial deficit in atmospheric long-wave cooling that causes a positive RESTOM and leads to heat gain by the ocean and positive SST biases particularly in the tropics. NorESM2 represents an improvement in this respect compared to NorESM1. This is particularly evident in the tropical and sub-tropical troposphere (Fig. 16). In addition, the medium-resolution version of the model has more realistic upper tropospheric meridional temperature gradients, and reduced near-surface temperature biases.

The extratropical storm tracks are generally better simulated in NorESM2 than in NorESM1, particularly over the North Atlantic. The storm tracks additionally improve with higher resolution, both in the Northern and Southern Hemisphere.

Several aspects of the modeled cycling of fresh water are improved in NorESM2 compared to NorESM1, including the RMSE and spatial correlation of the bias in the total precipitation rate. The intensity of the hydrological cycle as compared to the observationally based findings of Trenberth et al. (2011a) is slightly exaggerated in NorESM2 as it was in NorESM1, consistent with the underestimated cloudiness and thus overestimated solar radiation in the tropics and sub-tropics. The transport of oceanic water vapor over the continents is smaller in NorESM2 than NorESM1, indicating a slightly too efficient re-cycling of oceanic water vapor associated with over-estimated oceanic precipitation and higher cloudiness in the extratropics.

The seasonal blocking frequency in the Northern Hemisphere is in particular underestimated over the Atlantic - European sector during winter (DJF) by NorESM2. During spring (MAM), NorESM2-MM is closest to the reanalysis, while during

summer (JJA) and autumn (SON), NorESM1-Happi performs best in these regions. While NorESM1 tends to overestimate the blocking frequency over the Pacific, NorESM2 generally lies closer to the reanalysis in that sector. Although the use of 30 years from ERA-Interim for verification may not be fully representative for blocking climatology, the simulation of NH blocking continues to be a challenge for NorESM.

The coupled model internally generates a self-sustained ENSO mode with spatial and temporal characteristics similar to observations. ENSO SST anomalies are very large compared to observations (with a NINO3.4 anomaly greater than 2.5° C in the average El-Niño event, compared with 1.5° C in observations), and they tend to peak early in the season, i.e. between November and December instead of between December and January as observed. Nevertheless many proprties of the ENSO are similar to those observed, and El-Niño teleconnections are quite realistic both in the tropics and at mid- and high latitudes. Less

satisfactory is the performance of the coupled model in terms of the Madden-Julian oscillation. Here the low resolution version appears to produce more intense and more realistic sub-seasonal tropical variability than the medium-resolution version.

*Code and data availability.*    The CMIP6 and CMIP5 data considered here is available through data archives operated and maintained by the Earth System Grid Federation: https://esgf-node.llnl.gov/search/cmip6/ and https://esgf-node.llnl.gov/projects/cmip5/. The model code is available on github, is available on request from the corresponding author and will be fully publicly available mid 2020.

*Author contributions.*    ØS coordinated the writing of this article. ØS, DO, AK and TT wrote the CAM6-Nor description section. MB, AN, JS, JT wrote the ocean component description section. AN performed the evaluation of the model climatology for the oceanic variables and wrote the corresponding sections. JBD erformed the evaluation and wrote the sea ice section. TT performed the evaluation and wrote the ENSO section. LSG performed the analysis of the climatological mean state of near-surface temperature, zonally averaged temperature and zonal winds, extratropical storm tracks, atmospheric blocking, and the MJO and wrote the corresponding sections with contributions from

TT. CS contributed to the storm-track analysis. OL and TI performed the analysis and wrote the section on precipitation and the hydrological cycle. DO and Ada G conducted the ECS and feedback analysis. TL performed the analysis and wrote the section on future scenarios. The development of the NorESM2 model was co-ordinated by CH and TI during the first years (EVA-project), and is co-ordinated by MB and MS since fall 2017. All co-authors contributed to the model development, the modelling infrastructure, the setting up of the CMIP6 experiments, data processing and/or data distribution and gave feedback to the manuscript.

*Competing interests.*    There are no competing interests.

*Acknowledgements.*    We are particularly grateful to Cecile Hannay, M. Vertenstein, A. Gettelmann, J.F. Lamarque and others for invaluable advice on numerous scientific and technical issues when using CESM code, and the support by the CESM program directors during the NorESM2 development period. We acknowledge support from the Research Council of Norway funded projects EVA (229771), HappiEVA



(261821), INES (270061), and KeyClim (295046), Horizon 2020 projects CRESCENDO (Coordinated Research in Earth Systems and Climate: Experiments, Knowledge, Dissemination and Outreach, no. 641816,), APPLICATE no. 727862 and IS-ENES3, no. 824084, Nordic project eSTICC (57001) High performance computing and storage resources were provided by the Norwegian infrastructure for computational science (through projects NN2345K, NN9560K, NN9252K, NS2345K, NS9082K, NS9560K, NS9252K, and NS9034K) and the Norwegian Meteorological Institute. The observational SLA dataset used in this work was obtained from the obs4MIPs project hosted on the Earth System Grid Federation and the original altimeter products were produced by Ssalto/Duacs and originally distributed by Aviso+, with support from CNES (https://www.aviso.altimetry.fr)





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





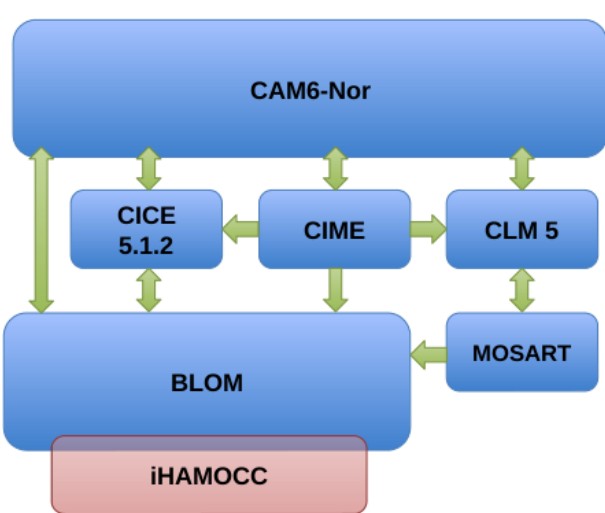

**Figure 1.** Overview of the different components in NorESM2 and their interactions ( CIME: configuration handler; CAM6-Nor: atmosphere and aerosol; CICE5.1.2: sea ice ; CLM5: land and vegetation, MOSART: river transport; BLOM: ocean; iHAMOCC: ocean carbon cycle).



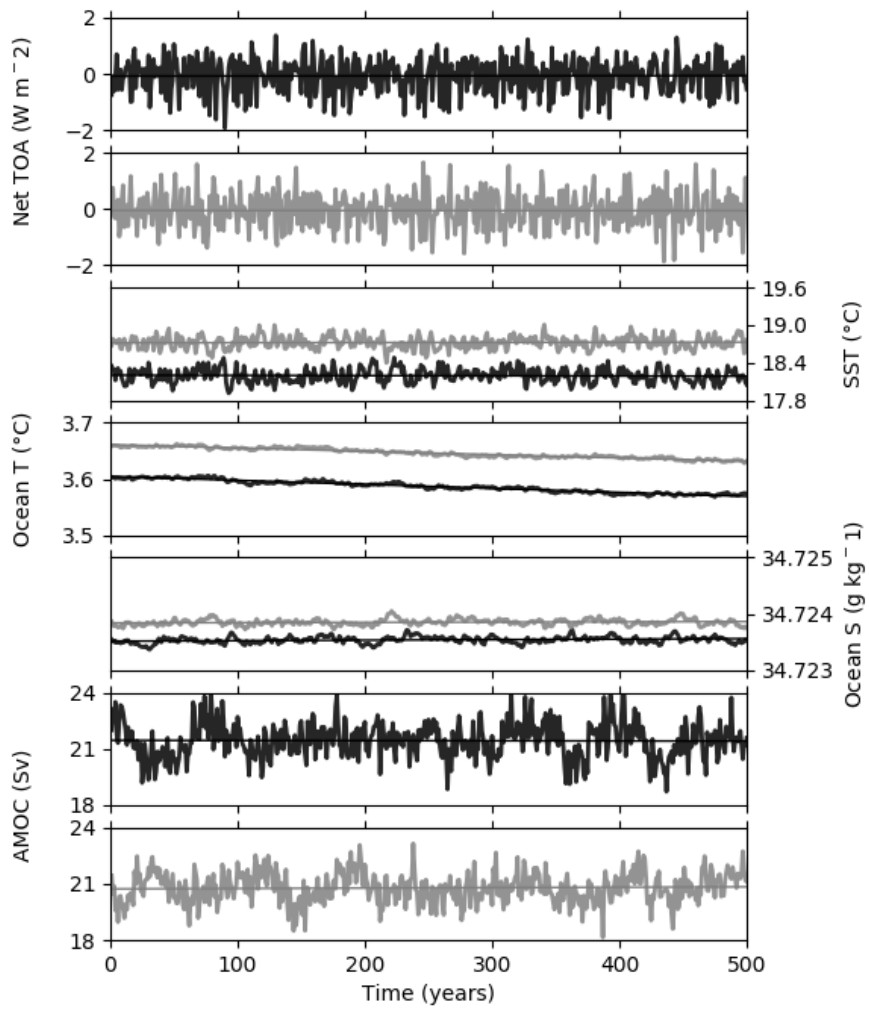

**Figure 2.** Pre-industral control experiment characteristics for NorESM2-LM (black lines) and NorESM2-MM (grey lines). Time evolution of globally averaged top-of-the-atmosphere (TOA) net radiative balance (first and second panel from top), sea surface temperature (SST) (third panel), ocean temperature (fourth panel), ocean salinity (fifth panel), and Atlantic meridional overturning circulation (AMOC) at 26.5° N (bottom two panels) for model years 0–500.



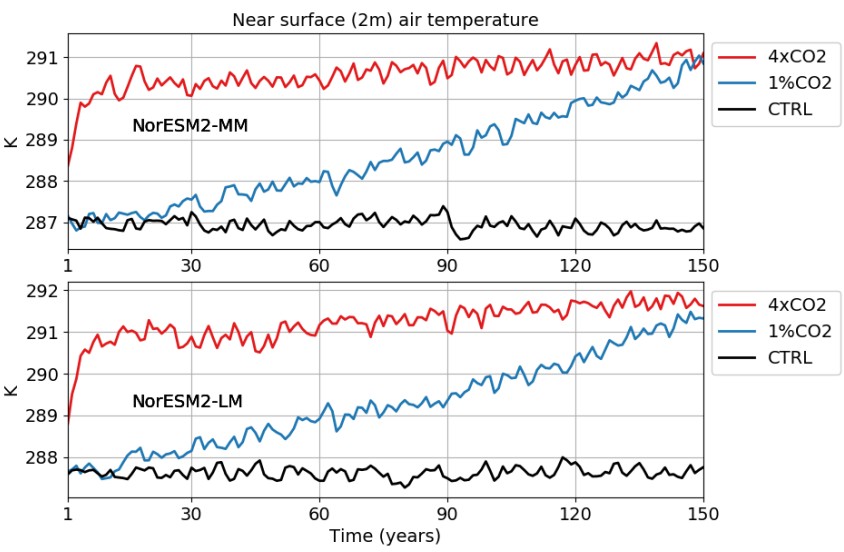

**Figure 3.** Time evolution of globally averaged near-surface temperature in NorESM2-MM (top) and NorESM2-LM (bottom) for the pre-industrial control simulation, the abrupt $4\times CO_2$ experiment, and the gradual increase $1\%CO_2$ experiment for model years 1–150.





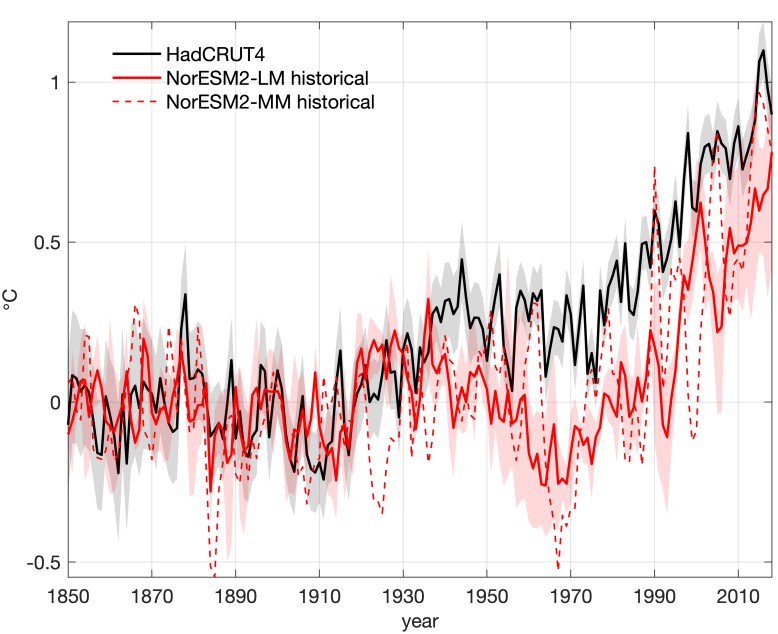

**Figure 4.** Time evolution of globally averaged surface temperature in the historical simulations of NorESM2-LM (three members, solid red line) and NorESM2-MM (one member, dashed red line) shown with the observations (solid black line) from HadCRUT4 (Morice et al., 2012) updated to version HadCRUT.4.6.0.0. Temperatures are computed as anomalies from the time-mean over years 1850–1880. For NorESM2-LM, the solid red line shows the mean and the red shading the spread from three ensemble members. For HadCRUT4, the solid black line shows the median and the grey shading indicates the lower and upper bounds of the 95% confidence interval of the combined effects of all the uncertainties described in the HadCRUT4 error model (measurement and sampling, bias, and coverage uncertainties).



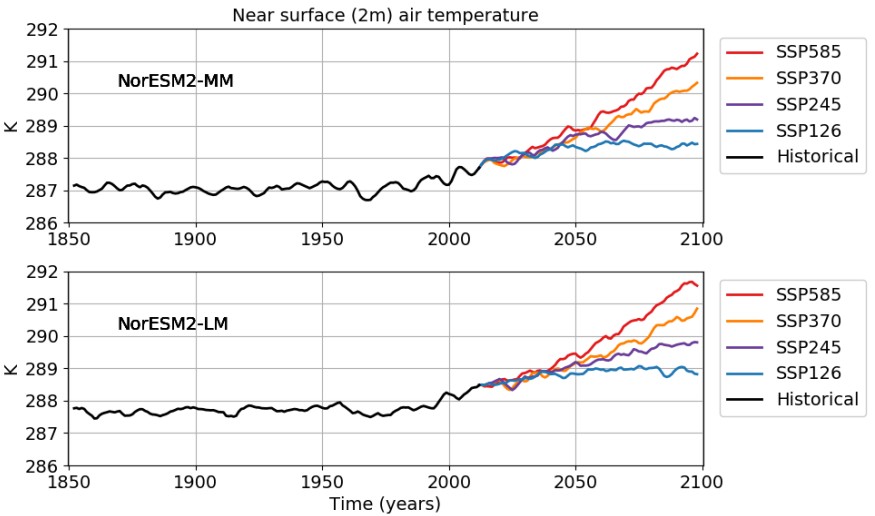

**Figure 5.** Time evolution of globally averaged surface air temperature in NorESM2-MM and NorESM2-LM from the historical simulations (black lines) and CMIP6 scenario experiments SSP1-2.6 , SSP2-4.5, SSP3-7.0 , and SSP5-8.5 (colored lines). A five-year moving average is used.

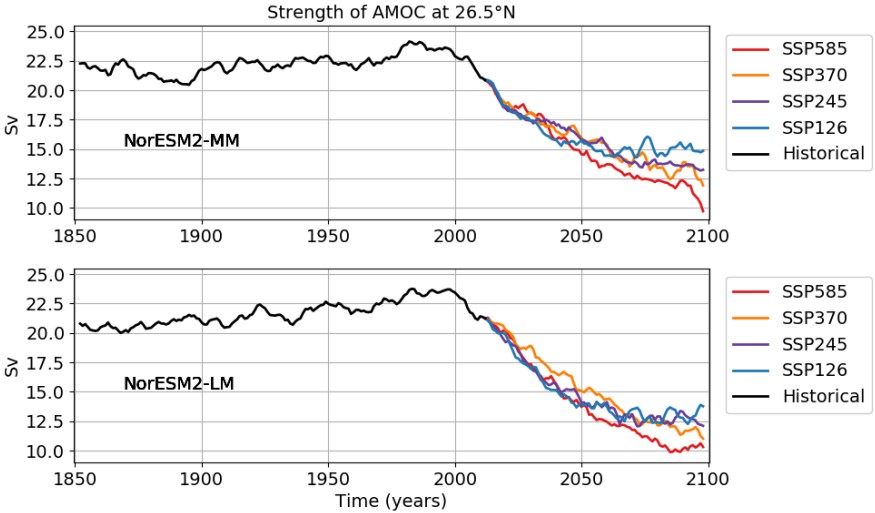

**Figure 6.** Curves as in figure 5, but for the time evolution of Atlantic meridional overturning circulation (AMOC).





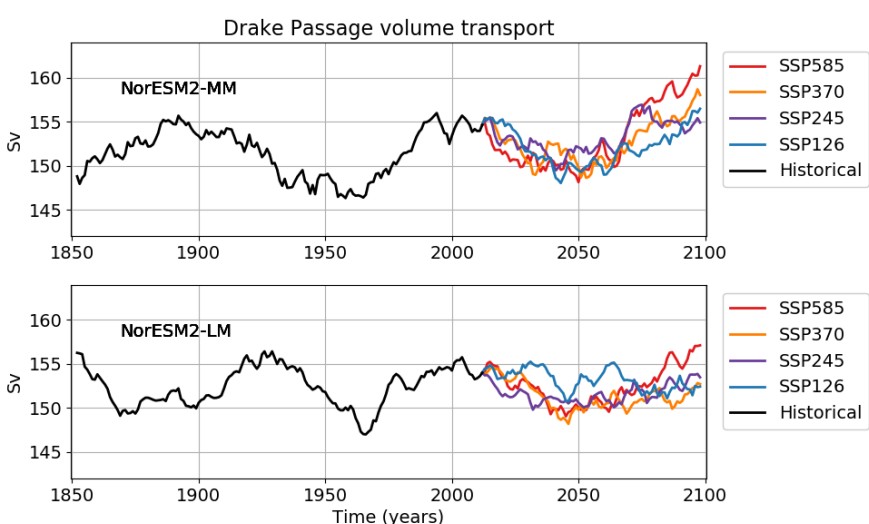

**Figure 7.** As figures 5 and 6, but for the time evolution of transport due to the Antarctic circumpolar current through the Drake Passage.





**Figure 8.** Northern Hemisphere sea ice area for March and September for historical and scenario experiments. Upper panel for NorESM2-MM and lower for NorESM2-LM. Black lines show observations from OSISAF (Lavergne et al., 2019) for the years 1979–2019. For NorESM2-LM the ensemble mean, with shades for the ensemble range are shown for historical (1850–2014), SSP245 (2015–2059), and SSP370 (2015–2100) scenarios. The rest of the lines denotes only one realization.



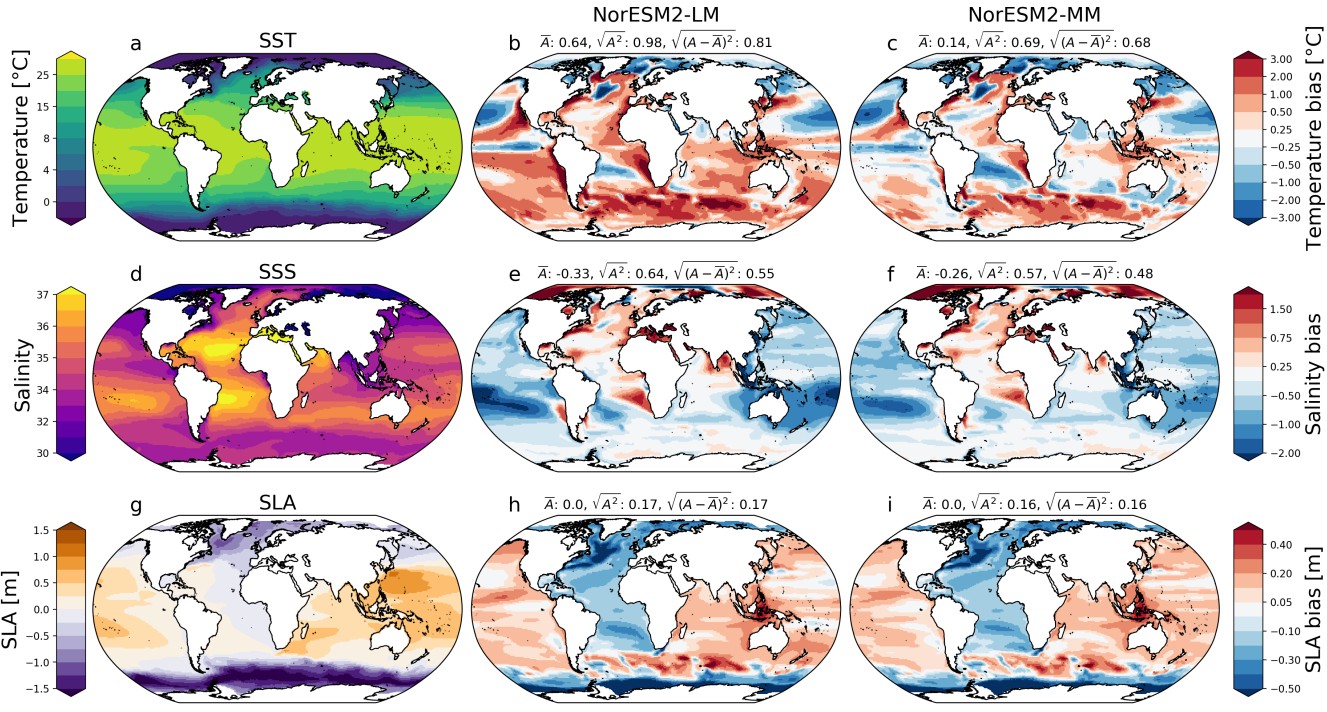

**Figure 9.** Observed climatologies (left column) and biases for NorESM2-LM (middle column) and NorESM2-MM (right column) for sea surface temperature (SST), sea surface salinity (SSS), and sea level anomalies (SLA). SST and SSS are compared to the World Ocean Atlas climatology (Locarnini et al., 2018; Zweng et al., 2018) between 1981–2010, whereas SLA is compared to AVISO altimetry between 1993–2010. For NorESM2-LM, we show the ensemble-mean bias using the three historical members. For NorESM2-MM we only have one ensemble member. Note that $A$ is the anomaly between the model and the observation, and we report the global mean bias ($\overline{A}$), global mean RMSE (root-mean-square error; $\sqrt{\overline{A^2}}$), and global mean RMSE with the mean bias first removed ($\sqrt{\overline{(A-\overline{A})^2}}$). SLA is re-defined to have zero mean over the ice-free region in the observational dataset (thus $\overline{A}$ is 0 by definition).

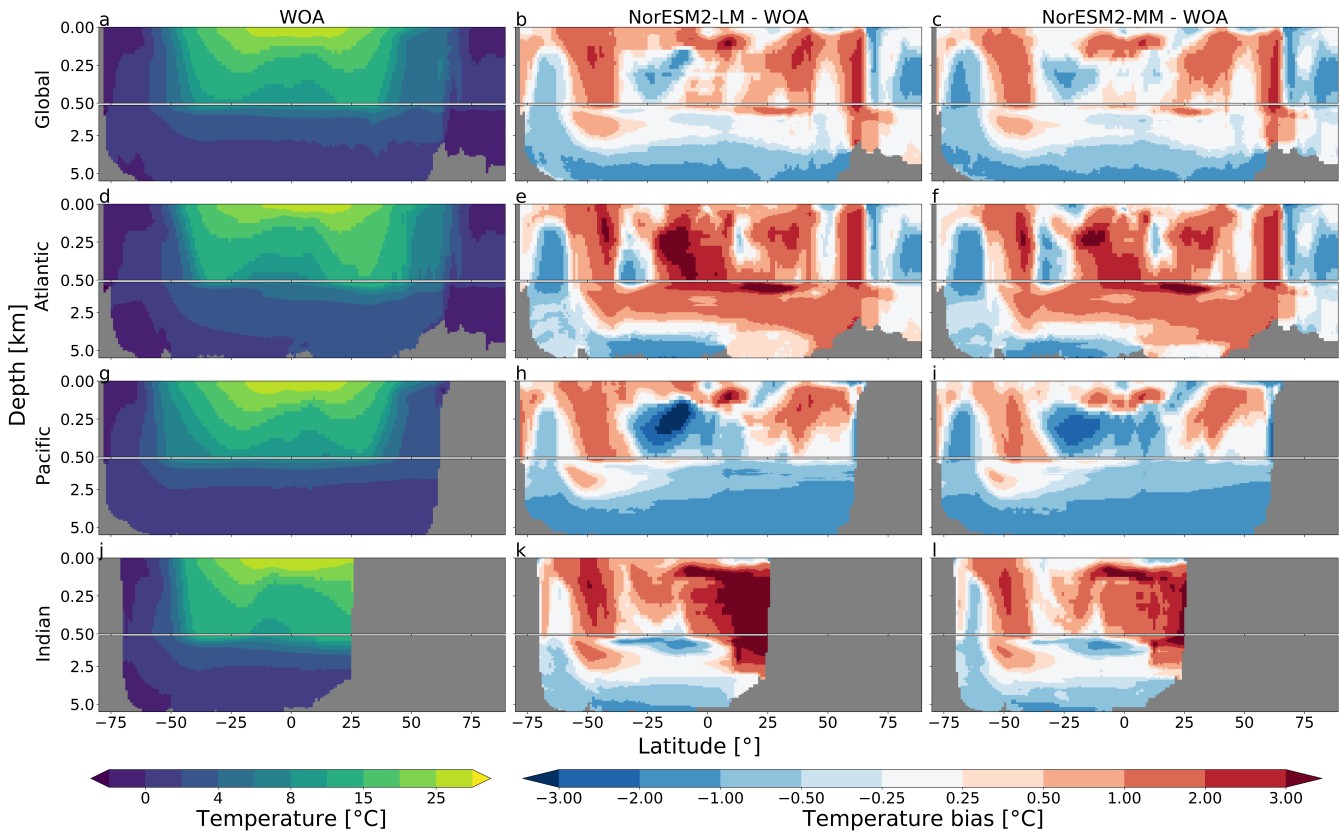

**Figure 10.** Zonal-mean bias in potential temperature for NorESM2-LM and NorESM2-MM. The bias is taken relative to World Ocean Atlas climatology (Locarnini et al., 2018; Zweng et al., 2018) using years 1981–2010 in the main ocean basins. Note the change in the vertical scale between the upper 500 meters and the lower 4500 meters. For NorESM2-LM, we show the ensemble-mean bias using the three historical members. For NorESM2-MM we only have one ensemble member.

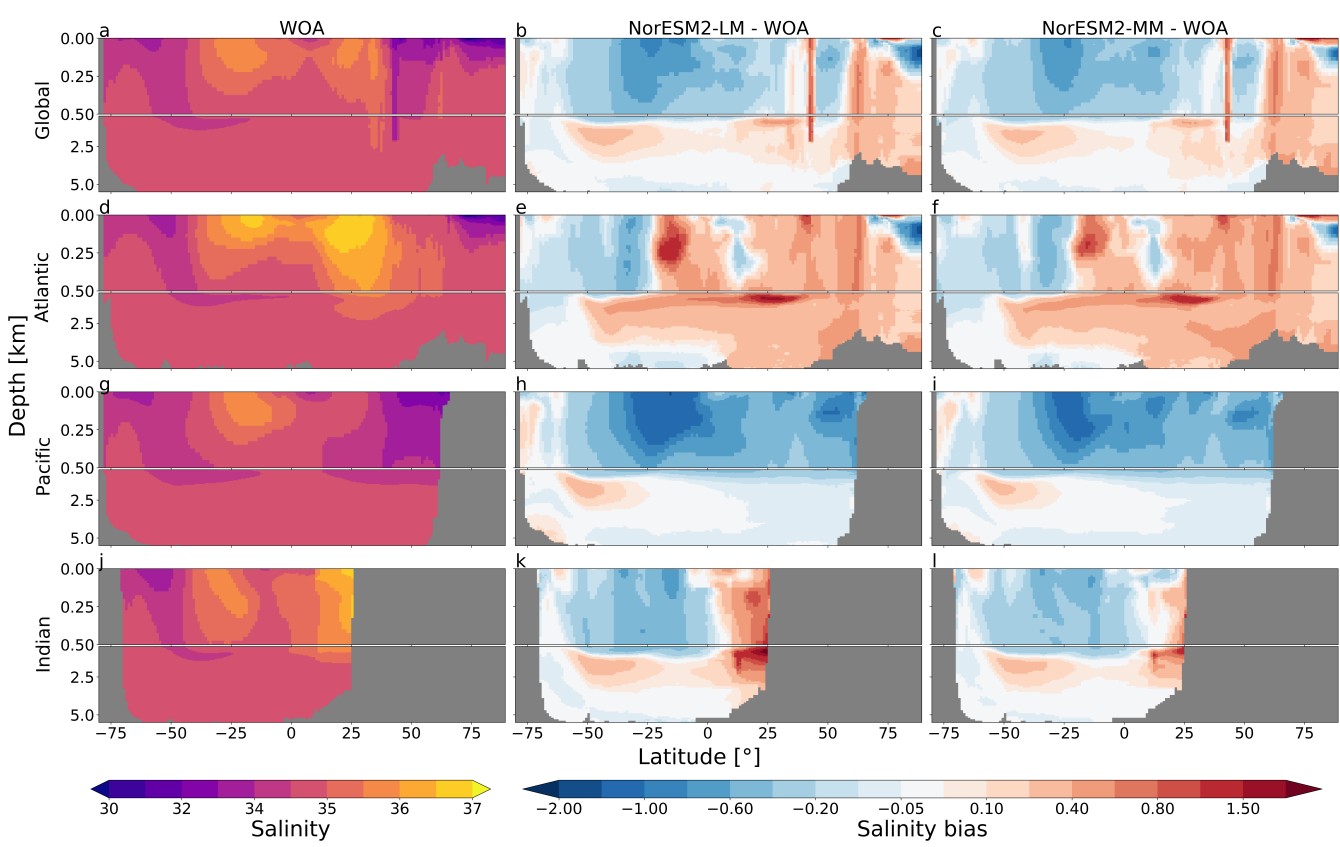

**Figure 11.** As in Fig. 10, but for the zonal-mean bias in salinity.



**Figure 12.** Sea ice concentration from OSISAF observations (OSI-V2.0; Lavergne et al., 2019) in March (a,c) and September (b,d) for Northern (a,b) and Southern (c,d) hemispheres. Differences between model and observations for the respective hemisphere and months are shown for NorESM2-LM (e-h), and NorESM2-MM (i-l). Model and observations are monthly means for the period 1980–2009. Unit is [%].




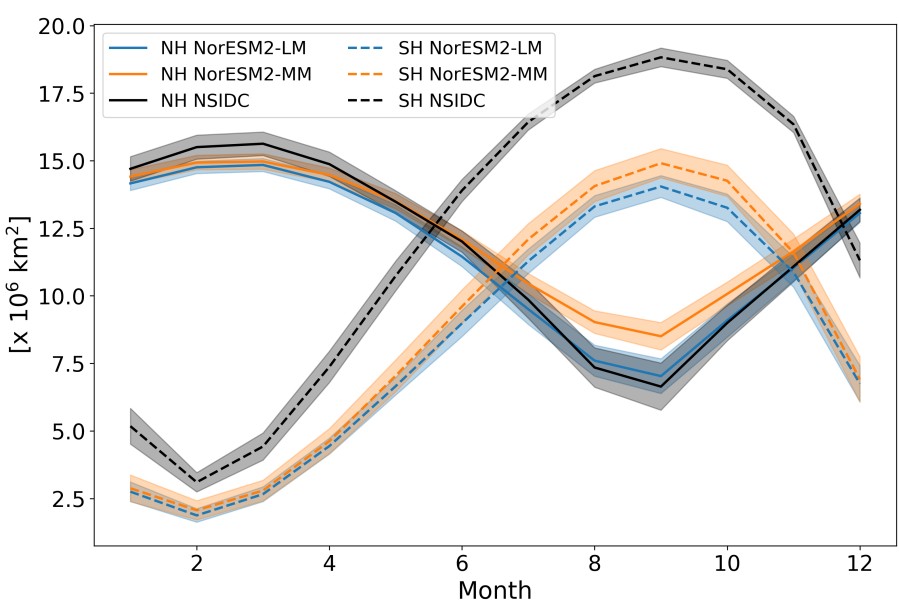

**Figure 13.** Northern hemisphere (NH) and southern hemisphere (SH) seasonal cycles of sea ice extent in the first historical member from NorESM2-LM and from NorESM2-MM averaged over years 1980–2009 and compared to observations from NSIDC. Shaded areas show interannual variation as standard deviation. Units are $10^6$ km$^2$.



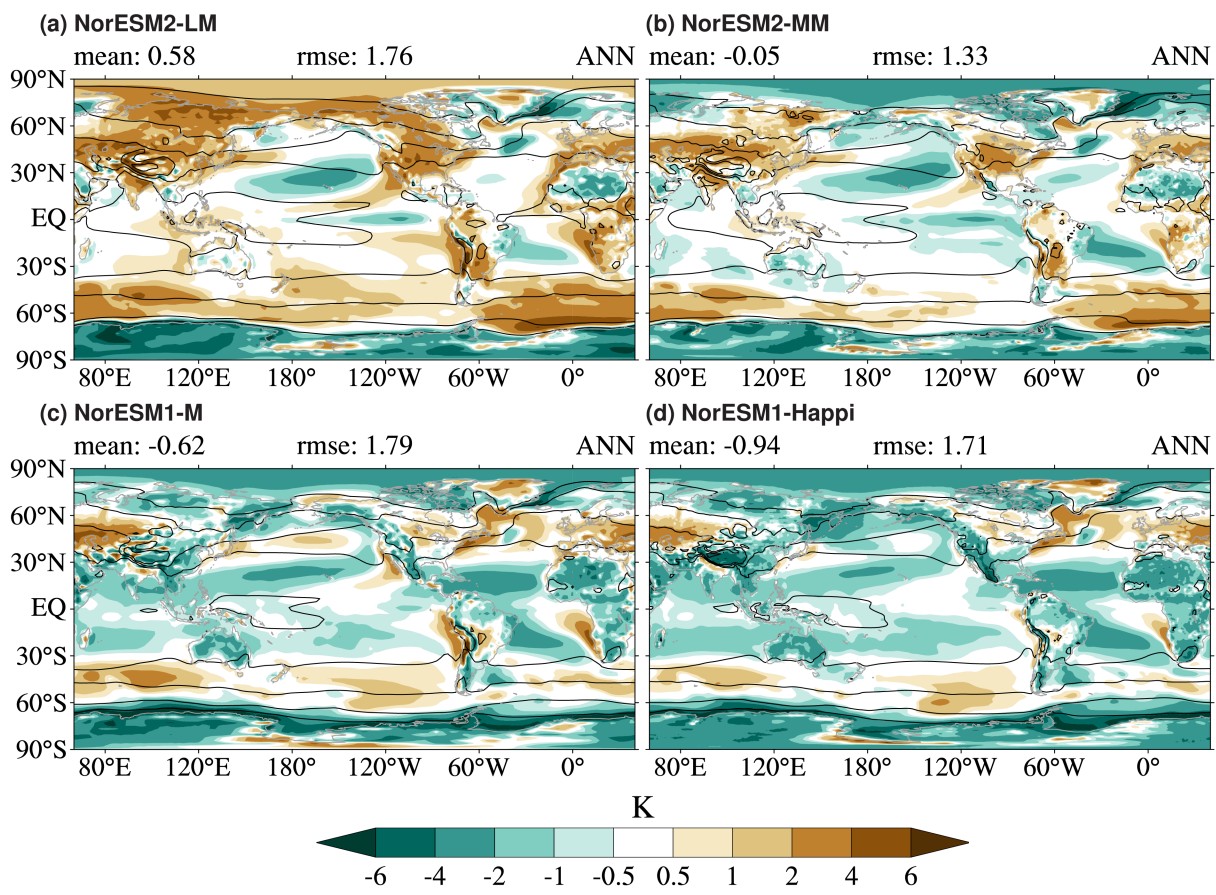

**Figure 14.** Annual-mean ensemble-mean model bias for near-surface temperature (colors) shown with the present-day model climatology (solid black contours) from NorESM2-LM (a; years 1980–2009), NorESM2-MM (b, years 1980–2009), NorESM1-M (c; years 1976–2005), and NorESM1-Happi (d; years 1976–2005). The bias is taken with respect to ERA-Interim (years 1979–2008 for NorESM1-M/Happi and 1980–2009 for NorESM2-LM/MM). Units are K.



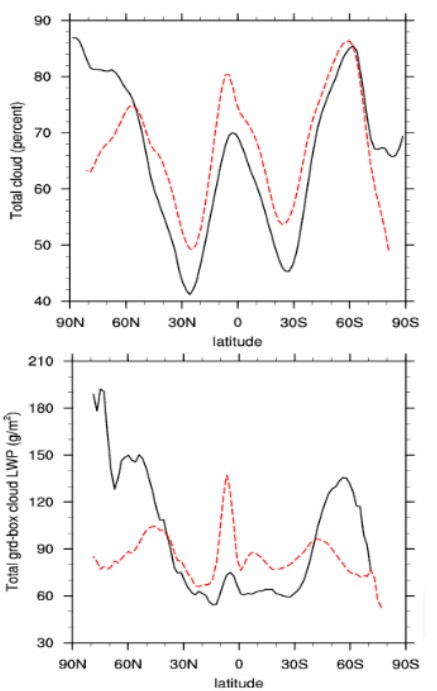

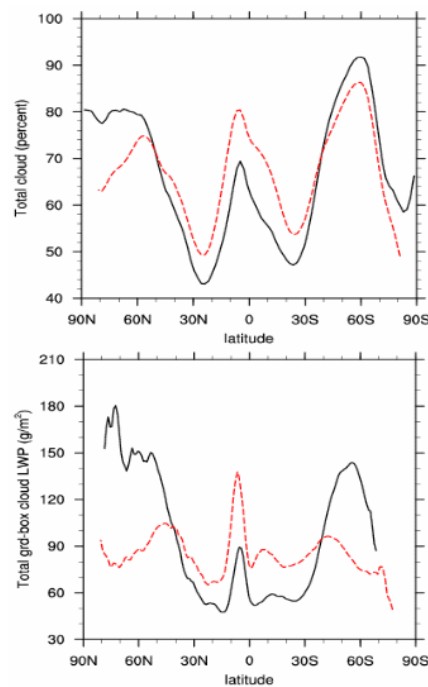

**Figure 15.** Evaluation of cloud cover and liquid water path (LWP) of NorESM2-MM (left column) and NorESM2-LM (right column) (solid black lines, years 1980–2009). Upper panel: Compared to observational estimate (CLOUDSAT) of annual-mean cloud cover (red dashed lines). Lower panel: Observational estimate of annual mean oceanic liquid water path (dashed red lines) (O'Dell et al., 2008).

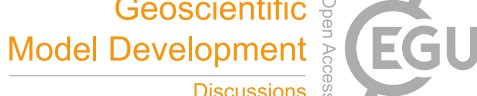

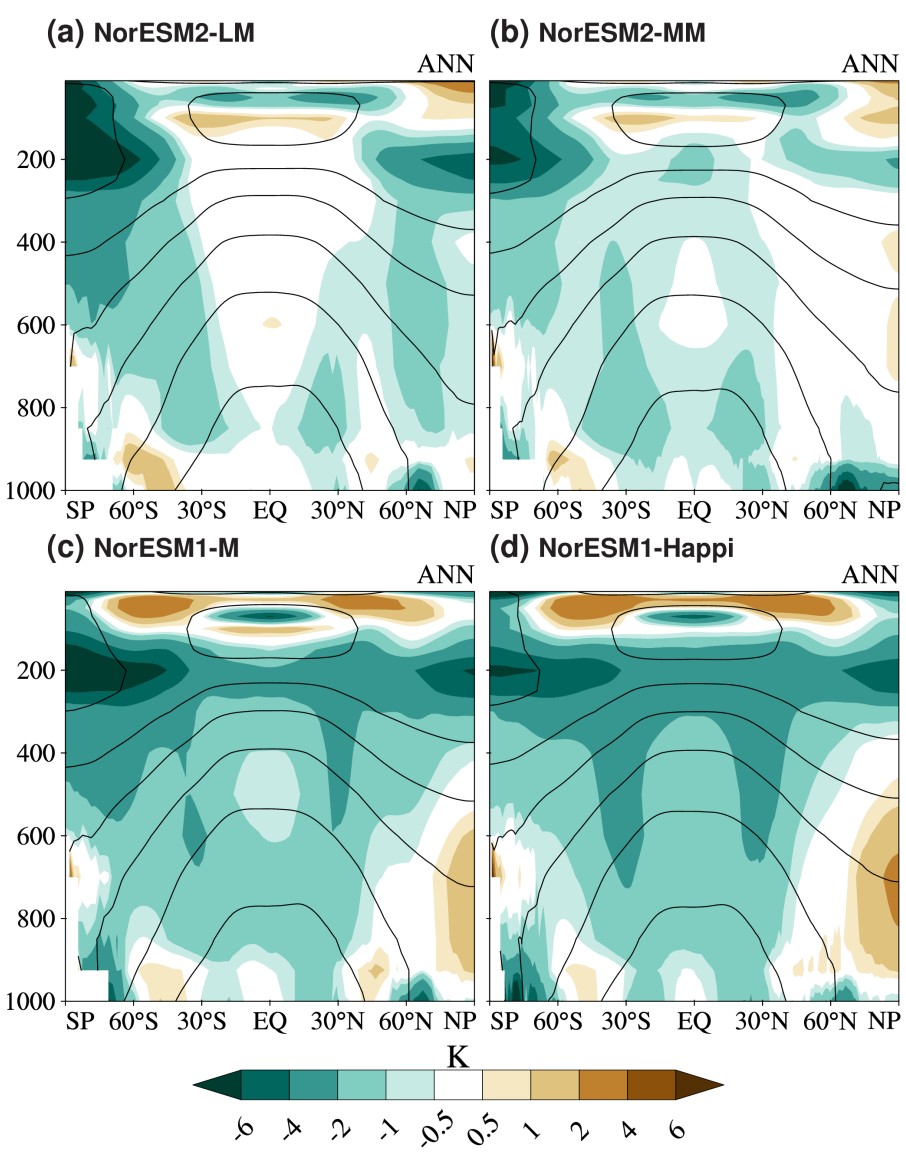

**Figure 16.** Annual-mean ensemble-mean model bias for temperature (colors) shown with the present-day model climatology (solid black contours) from NorESM2-LM (a; years 1980–2009), and NorESM2-MM (b, years 1980–2009, NorESM1-M (c; years 1976–2005), NorESM1-Happi (d; years 1976–2005)). The bias is taken with respect to ERA-Interim (years 1979–2008 for NorESM1-M/Happi and 1980–2009 for NorESM2-LM/MM). Units are K.

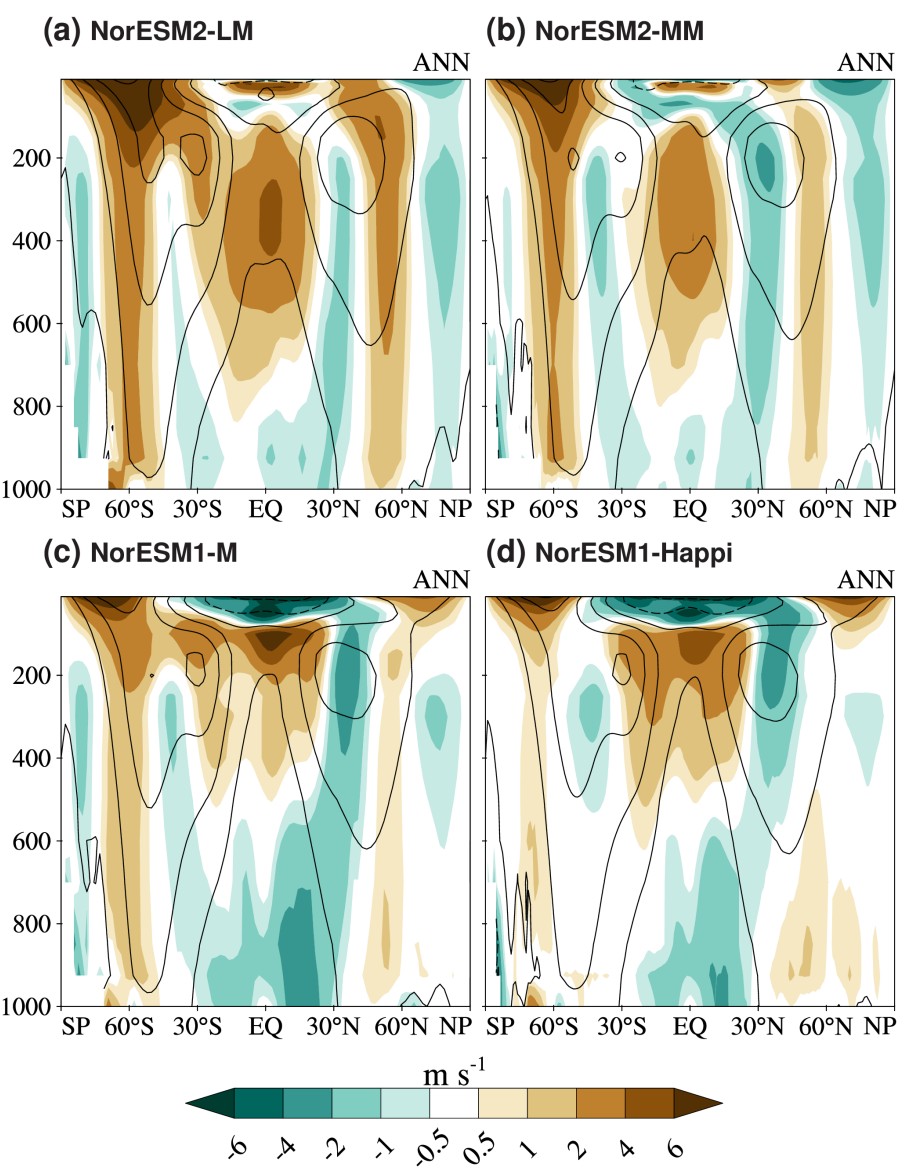

**Figure 17.** Annual-mean ensemble-mean model bias for the zonal wind (colors) shown with the present-day model climatology (solid black contours) from NorESM2-LM (a; years 1980–2009), and NorESM2-MM (b, years 1980–2009), NorESM1-M (c; years 1976–2005), NorESM1-Happi (d; years 1976–2005). The bias is taken with respect to ERA-Interim (years 1979–2008 for NorESM1-M/Happi and 1980–2009 for NorESM2-LM/MM). Units are m s$^{-1}$.



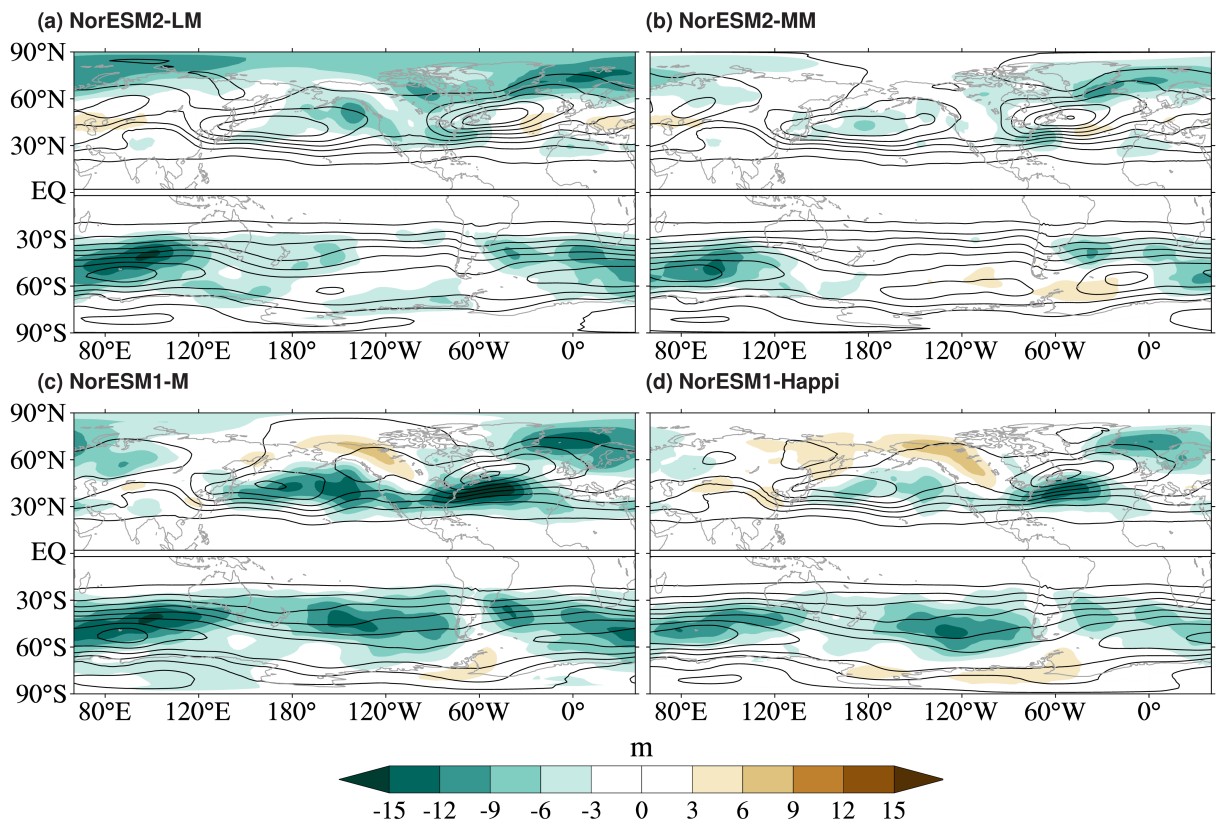

**Figure 18.** Ensemble-mean model bias for the extratropical storm tracks in terms of bandpass-filtered variability (colors) shown with the present-day model climatology (solid black contours; range is from 8 to 70 m with intervals of 8 m) for NorESM2-LM (a), NorESM2-MM (b), NorESM1-M (c), and NorESM1-Happi (d). Panels show the winter season for both hemispheres (DJF for the NH and JJA for the SH). The storm tracks are plotted in terms of the standard deviation of the bandpass-filtered geopotential height field at 500 hPa (Z500). The model data is taken from the CMIP6 historical members for NorESM2-LM (three members) and NorESM2-MM (one member) for years 1980–2009 and from the CMIP5 historical members for NorESM1-M (three members) and NorESM1-Happi (three members) for years 1976–2005. The bias is taken with respect to ERA-Interim (years 1980–2009 for NorESM2-LM/MM and 1979–2008 for NorESM1-M/Happi). Units are m.



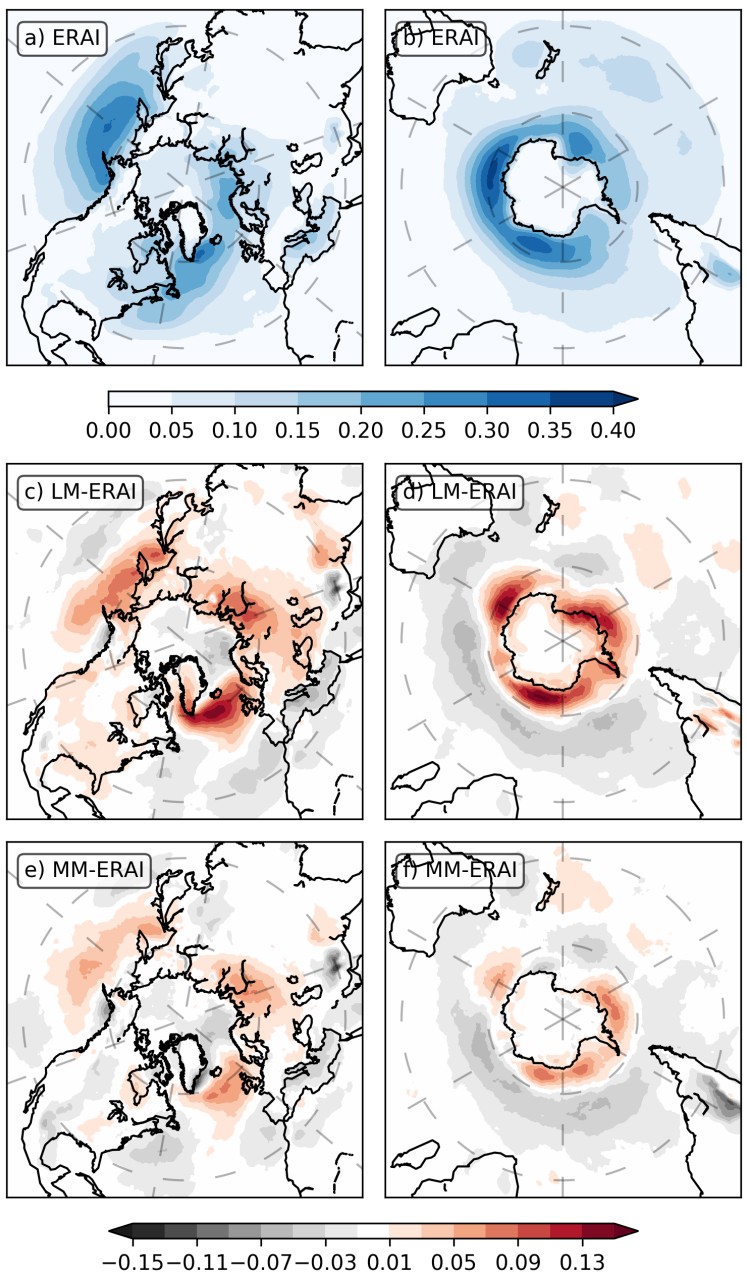

**Figure 19.** Frequency of occurrence (the unitless numbers give the fraction of the time when cyclones are present) of extratropical surface cyclones for the Northern Hemisphere (a, c, e) and Southern Hemisphere (b, d, f) storm tracks during the respective winter seasons (DJF in the Northern Hemisphere and JJA in the Southern Hemisphere). The figure shows the ERA-Interim climatology (a–b) and the bias with respect to ERA-Interim for NorESM2-LM (c–d) and NorESM2-MM (e–f). The data is taken from years 1980–2009. For both NorESM2-LM and NorESM2-MM, only the first historical member is used.



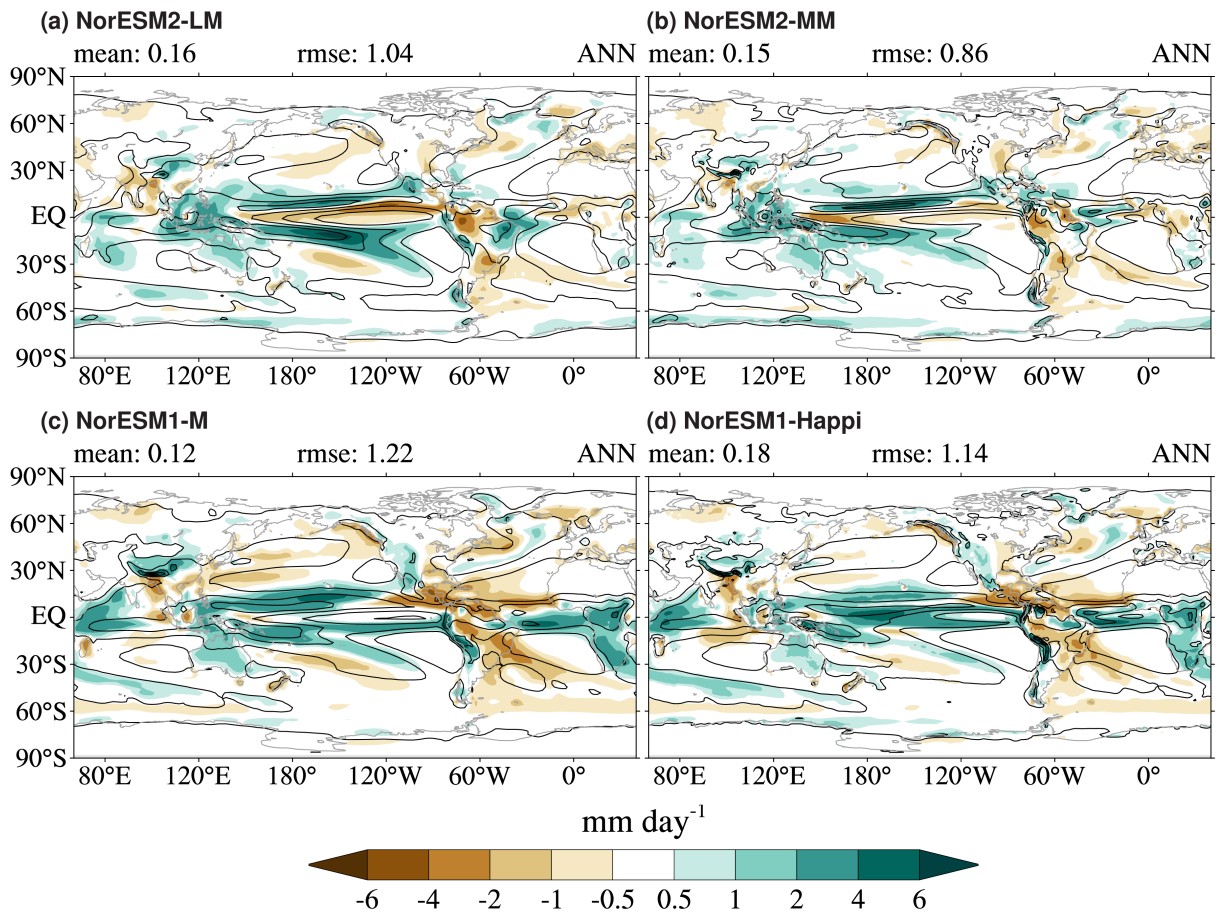

**Figure 20.** Annual-mean ensemble-mean model bias for total precipitation rate (colors) shown with the present-day model climatology (solid black contours) from NorESM1-M (a; years 1976–2005), NorESM1-Happi (b; years 1976–2005), NorESM2-LM (c; years 1980–2009), and NorESM2-MM (d, years 1980–2009). The bias is taken with respect to ERA-Interim (years 1979–2008 for NorESM1-M/Happi and 1980–2009 for NorESM2-LM/MM). Units are mm day$^{-1}$.



**Figure 21.** Annual and seasonal total precipitation rate climatology from GPCP (column 1) and biases with respect to GPCP for NorESM1-M (column 2), NorESM2-LM (column 3), and NorESM2-MM (column 4). The time period is 1980–2009 in all cases and all available members from the historical simulations are used (3 for NorESM1-M and NorESM2-LM, and 1 for NorESM2-MM). For the bias plots, the following global-mean metrics are shown in the bottom left corner of each panel: bias, RMSE (lower is better), and spatial correlation (Cor; higher is better). For NorESM2-LM and NorESM2-MM, green/red dots are shown on the left side of these metrics and indicate whether the NorESM2 results are better (green) or worse (red) than NorESM1-M. Units are mm day$^{-1}$





**Figure 22.** As in Fig. 21, but for the inter-annual variability (IAV) of the total precipitation rate. The IAV is defined as the standard deviation of the timeseries in each grid point.

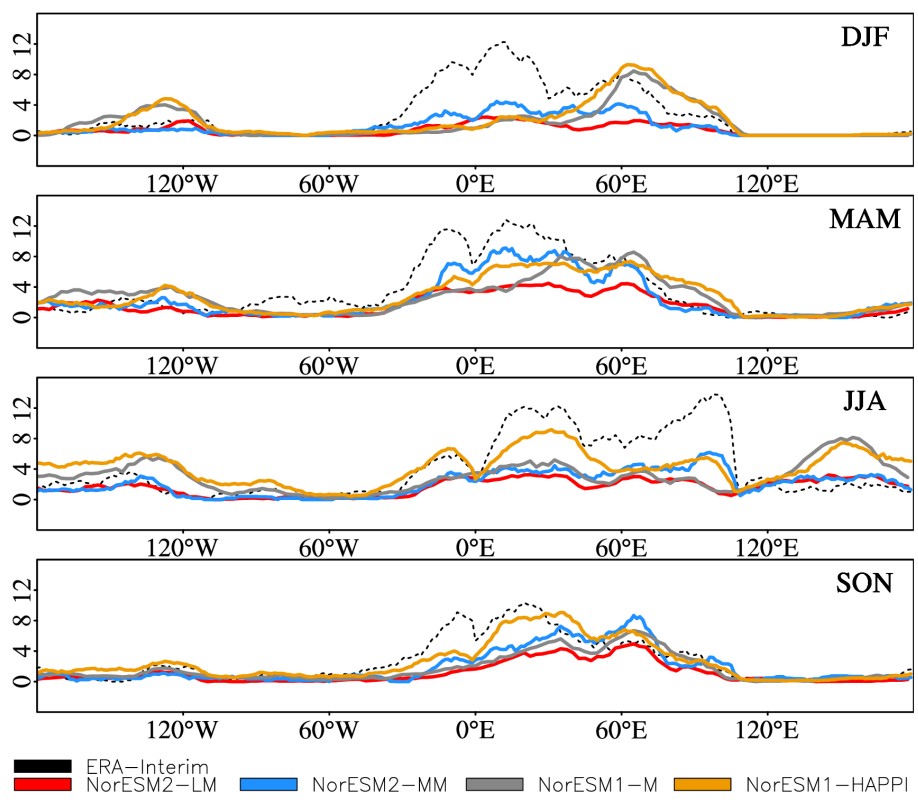

**Figure 23.** Ensemble-mean seasonal blocking frequency in the Northern Hemisphere for ERA-Interim (stippled black line; years 1980–2009), NorESM2-LM (red solid line; years 1980-2009), NorESM2-MM (blue line; years 1980–2009), NorESM1-M (grey line; years 1976–2005), and NorESM1-Happi (yellow line; years 1976–2005). The model data is taken from the CMIP6 historical members for NorESM2-LM (three members) and NorESM2-MM (one member) and from the CMIP5 historical members for NorESM1-M (three members) and NorESM1-Happi (three members). The blocking frequency is computed using the variational Tibaldi and Molenti (vTM) index (Tibaldi and Molteni, 1990; Pelly and Hoskins, 2003; Iversen et al., 2013; Graff et al., 2019). Units are percent.

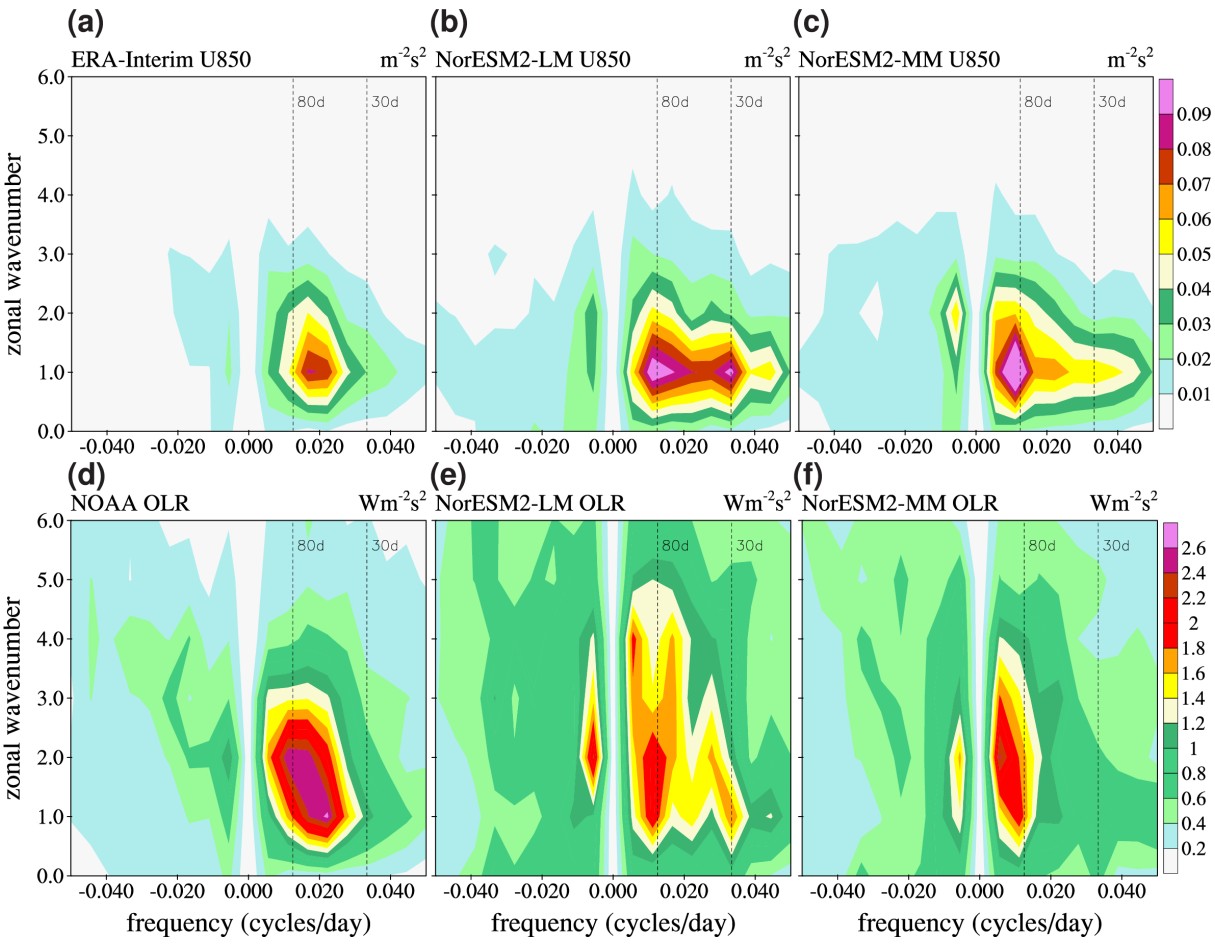

**Figure 24.** Wavenumber frequency spectra for the extended winter season (November - April) for the latitude band 10° S to 10° N for NorESM2-LM and NorESM2-MM. The spectra are computed from daily zonal winds at 850 hPa (U850) using data from (a) ERA-Interim and (b-c) NorESM, and from daily outgoing longwave radiation (OLR) using data from NOAA (d), and NorESM (e-f). The climatological daily cycle was removed prior to computing the spectra. The model data is taken from the first CMIP6 historical member for NorESM2-LM and NorESM2-MM for years 1980–2009, and for years 1979–2008 for ERA-Interim and NOAA.



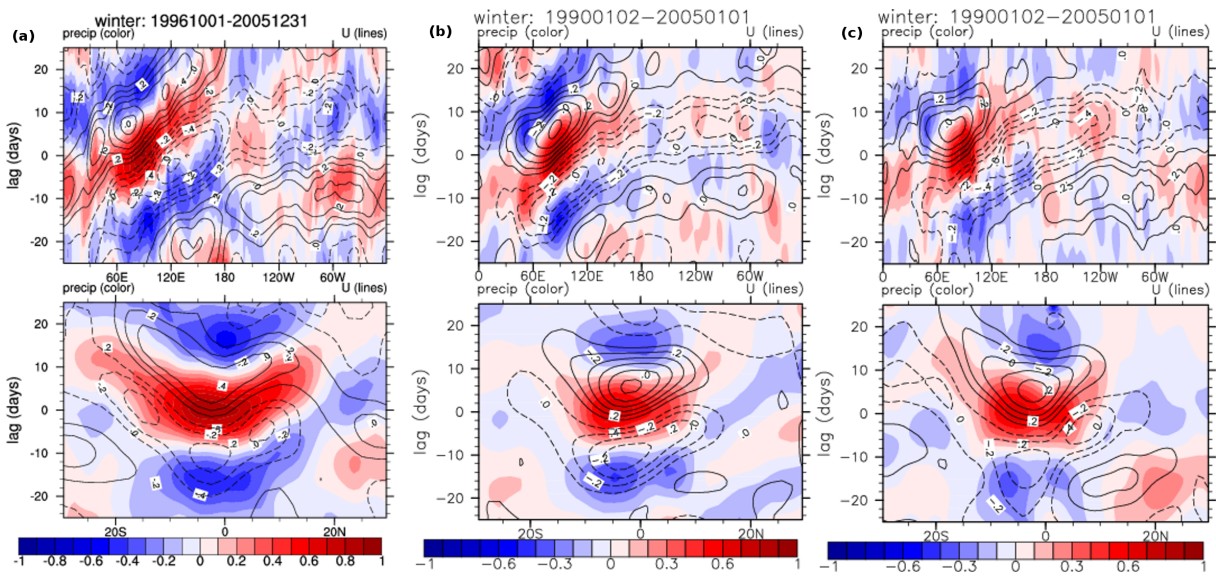

**Figure 25.** Lead-lag correlations during the extended winter season (November - April) for the latitude band $10°$ S to $10°$ N for observations, NorESM2-MM, and NorESM2-LM. The correlations are computed from daily-mean precipitation and zonal winds at 850 hPa (U850) using data from (a) ERA-Interim, (b) NorESM2-MM, and (c) NorESM2-LM. The climatological daily cycle was removed prior to computing the spectra. The ERA-Interim is taken for the years 1996–2005. The model data is taken from the first CMIP6 historical member for NorESM2-MM and NorESM2-LM for years 1990–2005, and .





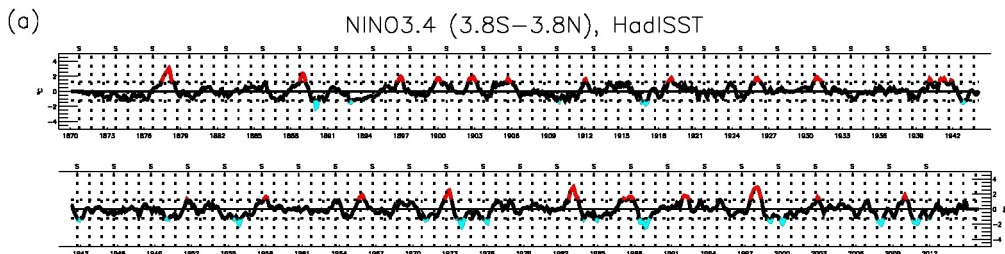

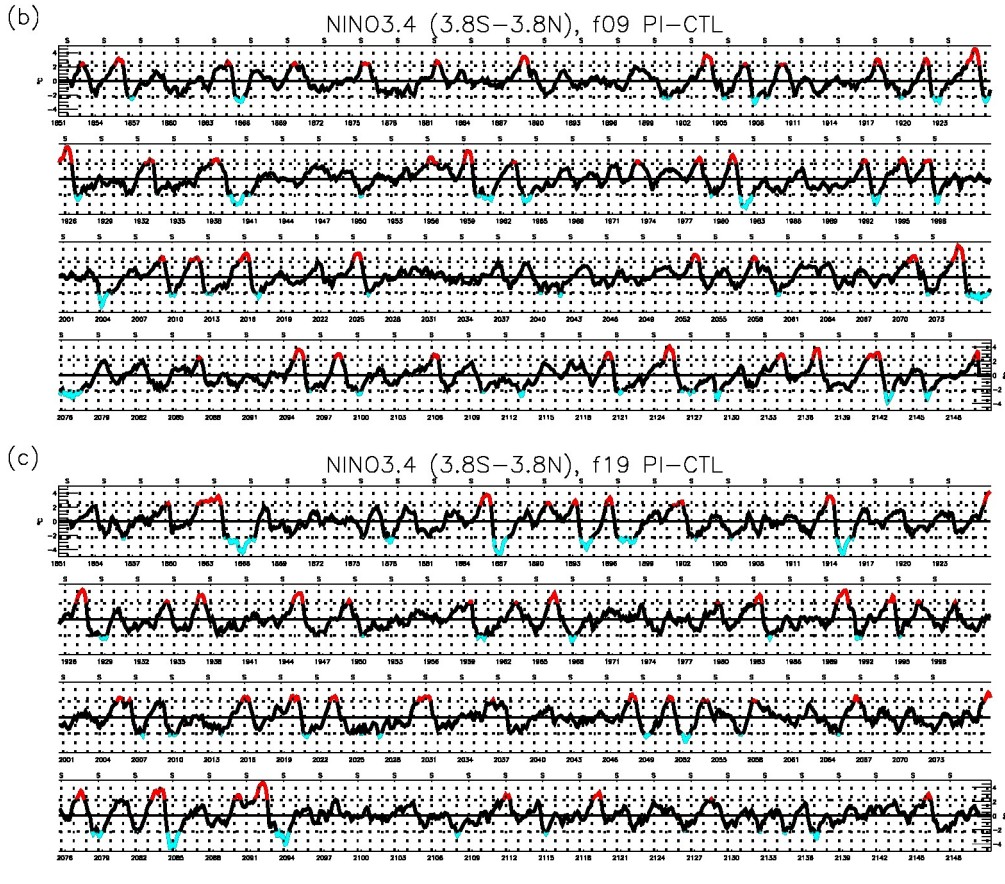

**Figure 26.** Time evolution of NINO3.4 index anomalies, in observations (panel a; HadISST v1.1; Rayner et al., 2003) and pre-industrial control experiments of NorESM2-MM (f09, panel b) and NorESM2-LM (f19, panel c). Anomalies above 1.5 standard deviations of the respective time-series are marked in red colour, anomalies less than -1.5 standard deviations are marked in cyan colour.





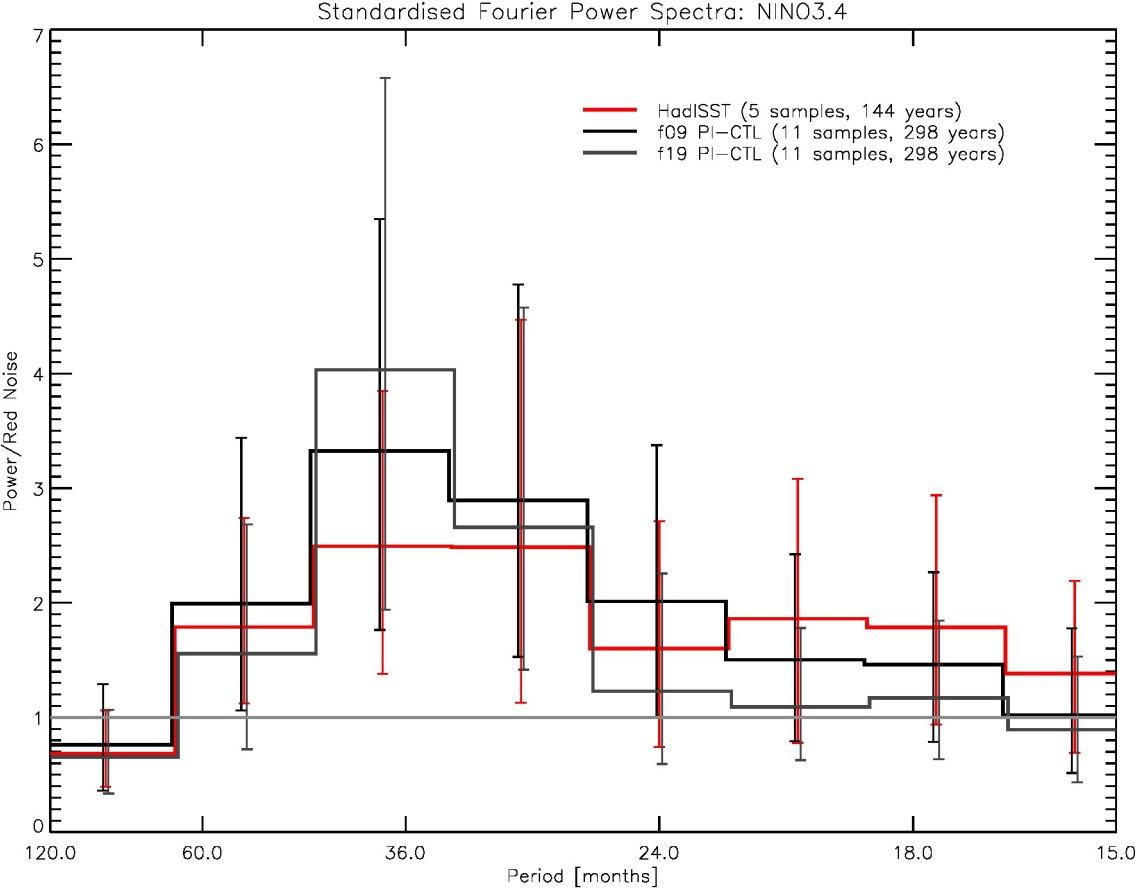

**Figure 27.** Frequency analysis of El Niño events in NorESM2-MM (f09) and NorESM2-LM (f19) and HadISST v1.1 (Rayner et al., 2003) observed data. Fourier power spectra of NINO3.4 anomalies, normalized to AR1 red-noise model for each (using bootstrapping, 1000 samples)



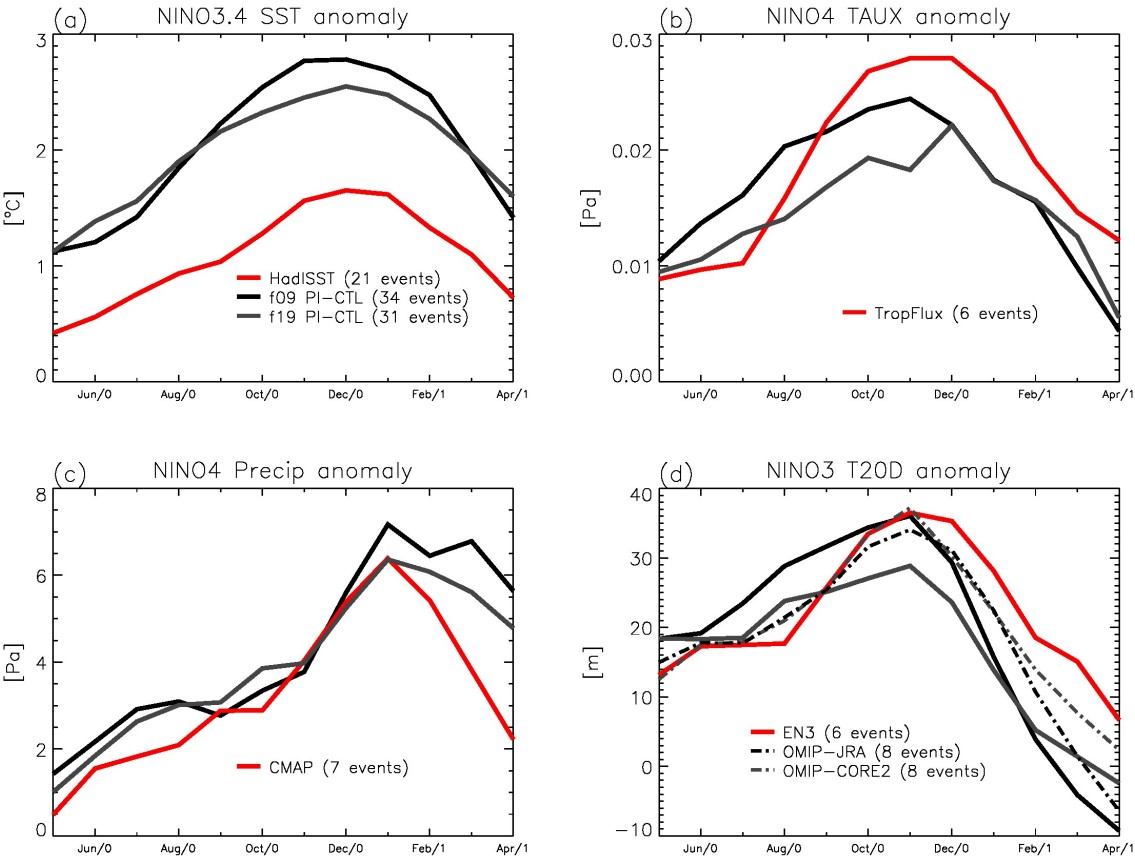

**Figure 28.** El-Nino monthly-mean composite anomalies averaged over NINOx regions (cf legend at the top of each panel) for SST, zonal wind-stress, precipitation, and 20C isotherm depth in the Equatorial Pacific. The composite El Niño event is obtained by averaging a number of years during which the occurrence of El Niño is diagnosed based on NINO3.4 anomalies. The selection criteria are on magnitude (1.5 standard deviations), duration (three months), seasonality (peak between November and January), and phase (no preceding El Niño, and a following La Niña). Year 0 indicates the year during which the event develops. For the observations, SST are HadiSST (Rayner et al., 2003), wind-stress from TropFlux (Praveen Kumar et al., 2012), precipitation from CMAP (Xie and Arkin, 1997), and the 20C isotherm depth is obtained from the EN3 dataseta (Guinehut et al., 2009).





**Figure 29.** El-Nino JFM composite anomalies for SSTs (colours, legend on the right of panel (b)), Precipitation (coloured line contours, red for negative values and blue for positive values; interval 2 mm/day) and geopotential height at 200hPa (black and white contour lines, solid for positive and stipled for negative values; interval 20 metres). The composites are defined as for Figure 28 (cf caption to that Figure).

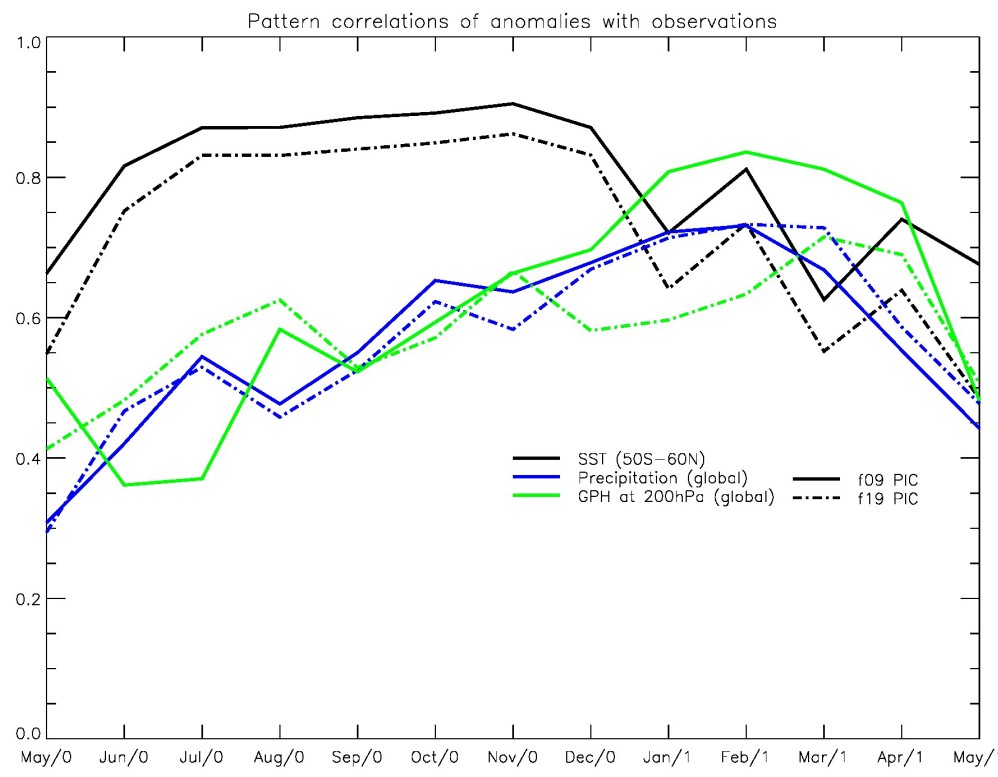

**Figure 30.** Pattern correlations between model simulations and oservations of El-Nino monthly-mean composite anomalies in SST, Precipitation and 200hPa geopotential height The patterns are computer from observations and model simulations as in Figure 29, but month by month over the composite El-Niño event, and the correlations between the resulting simulated and observed patterns are calculated. Observations are interpolated onto the relevant model grid in each case.





**Table 1.** Climate sensitivities of NorESM2-LM and NorESM2-MM compared to NorESM1 model versions (Equilibrium climate sensitivity (ECS), transient climate sensitivity (TCR) and transient climate response to cumulative carbon emissions (TCRE).

| Model version | ECS [K] | TCR [K] | TCRE [K EgC$^{-1}$] |
|---|---|---|---|
| NorESM2-MM | 2.50 | 1.33 | 1.21 |
| NorESM2-LM | 2.54 | 1.48 | 1.36 |
| NorESM1-Happi | 2.82 | 1.52 | N/A |
| NorESM1-ME | 2.99 | 1.56 | 1.93 |
| NorESM1-M | 2.86 | 1.39 | N/A |




**Table 2.** Global and annual means from the historical (years 1980–2009) simulations for NorESM2-MM (one realisation) and NorESM2-LM (3 member ensemble average). Values from re-analysis data or observations (references in the last column). The NorESM values are adjusted to compensate for the slight deviation between the top of the model atmosphere (abbreviated to TOA) and the top of the atmosphere as seen from satellites (Collins et al., 2006; TOA$_{SAT}$ in this table).

| Variable | NorESM1-M | NorESM1-Happi | NorESM2-LM | NorESM2-MM | Observation |
|---|---|---|---|---|---|
| TOA$_{SAT}$ net SW flux (W m$^{-2}$) | 234.9 | 240.2 | 237.3 | 238.2 | 240.6[a] 244.7[b] 234.0[c] |
| TOA$_{SAT}$ net clear-sky SW flux (W m$^{-2}$) | 289.5 | 289.4 | 287.4 | 287.2 | 287.6[a] 294.7[b] 289.3[c] |
| TOA$_{SAT}$ upward LW flux (W m$^{-2}$) | 232.4 | 237.6 | 237.6 | 236.7 | 239.6[a] 239.0[b] 233.9[c] |
| TOA$_{SAT}$ clear-sky upward LW flux (W m$^{-2}$) | 262.3 | 263.5 | 261.5 | 262.0 | 266.1[a] 266.9[b] 264.4[c] |
| TOA$_{SAT}$ LW cloud forcing (W m$^{-2}$) | 29.9 | 25.8 | 24.7 | 24.4 | 26.5[a] 27.2[b] 30.3[c] |
| TOA$_{SAT}$ SW cloud forcing (W m$^{-2}$) | -54.6 | -49.2 | -50.3 | -49.0 | -47.1[a] -48.6[b] -54.2[c] |
| Cloud cover (%) | 53.8 | 46.4 | 61.8 | 62.7 | 66.8[d] 66.8[e] |
| Cloud liquid water path (g m$^{-2}$) | 125.3 | 121.3 | 84.0 | 82.7 | 86.9[f] |
| Surface sensible heat flux (W m$^{-2}$) | 17.8 | 18.0 | 21.5 | 22.0 | 19.4[g] 15.8[h] 13.2[i] |
| Surface latent heat flux (W m$^{-2}$) | 81.7 | 83.7 | 83.2 | 83.1 | 87.9[g] 84.9[j] 82.4[k] 89.1[l] |

[a]CRES-EBAF (Loeb et al., 2005, 2009, 2012), [b]CRES (Loeb et al., 2005, 2009, 2012), [c]ERBE (Harrison et al., 1990; Kiehl and Trenberth, 1997), [d]ISCCP (Rossow and Schiffer, 1999; Rossow and Dueñas, 2004), [e]CLOUDSAT (L'Ecuyer et al., 2008), [f](O'Dell et al., 2008), [g]JRA25 (Onogi et al., 2007), [h]NCEP (Kanamitsu et al., 2002), [i]LARYEA (Large and Yeager, 2004), [j]ECMWF (Trenberth et al., 2011b), [k]ERA40 (Uppala et al., 2005), [l]WHOI (Yu and Weller, 2007; Yu et al., 2008)





**Table 3.** Evaporation (E) and precipitation (P) for different historical NorESM simulations and reference datasets. Units are thousand $km^3$ water per year. The years used are 1980–2009, except for NorESM1-Happi (1976–2005) which is from Graff et al. (2019), and the last two lines (2002–2008) which are from Trenberth et al. (2011a).

| Simulation | $E_{global}$ | $P_{global}$ | $(E-P)_{global}$ | $E_{ocean}$ | $P_{ocean}$ | $(E-P)_{ocean}$ |
|---|---|---|---|---|---|---|
| NorESM1-M r1 | 522.8 | 522.8 | 0.035 | 438.3 | 398.7 | 39.6 |
| r2 | 522.7 | 522.7 | 0.020 | 437.9 | 397.8 | 40.2 |
| r3 | 521.9 | 521.8 | 0.026 | 437.6 | 398.1 | 39.5 |
| NorESM1-Happi | | 533.5 | | 451.7 | 406.5 | 45.2 |
| NorESM2-LM r1 | 530.3 | 530.3 | 0.007 | 459.8 | 416.2 | 43.6 |
| r2 | 529.8 | 529.8 | 0.019 | 459.6 | 415.5 | 44.0 |
| r3 | 530.1 | 530.1 | 0.022 | 459.7 | 415.4 | 44.3 |
| NorESM2-MM r1 | 528.9 | 528.9 | 0.001 | 459.0 | 412.2 | 46.8 |
| Observation synthesis | 500 | 500 | 0 | 426 | 386 | 40 |
| ERA-Interim | 538 | 531 | 7 | 456 | 412 | 44 |