# Peer review of "Overview of the Norwegian Earth System Model, NorESM2, and Key Climate Response of CMIP6 DECK, Historical, and Scenario Simulations"

_Geoscientific Model Development, 2019_

## Referee Comment (RC1) · Anonymous Referee #1 · 16 Apr 2020

Review of "The Norwegian Earth System Model, NorESM2 – Evaluation of the CMIP6 DECK and historical simulations" by Seland et al.

General Comments: This manuscript documents the NorESM2 model, which is being used in CMIP6. The model is based on the CESM2 model with some notable differences in for instance, the representation of aerosols and their climate interactions; convection parameterizations as well as using different physical ocean and ocean biogeochemistry models. A description of the model and its differences from CESM and CAM6 models are detailed and an overview of the development and tuning of the fully coupled configuration in a pre-industrial climate is given. An assessment of some key

climate responses is provided, including the equilibrium climate sensitivity and transient climate response as well as future climate projections. An overview of the present-day performance of the historical simulations against observations is also provided.

This is a useful description and overview paper of the NorESM2 model and its climate characteristics and will be a very useful reference for the on-going and future CMIP6 analysis work. It is suitable for publication in GMD although I find it in its current form very long with 30 figures. The authors have included a lot in this one manuscript, covering a wide range of the model assessment with the result that the analysis of the individual components feels quite "light touch while I feel it is still missing evaluations of important parts of the fully coupled Earth system. I would recommend publication in GMD after my recommended revisions and additions to the analysis are made which I outline below.

Specific Comments:

There is no evaluation of aerosols (even though this is a key difference from the CESM2 model) or ocean or terrestrial biogeochemistry provided. While the land model is essentially CLM5 and documented elsewhere the authors note the important implications of the updated nitrogen-carbon limitation on the carbon cycle in the model but no assessment of the carbon cycle is provided. The ocean biogeochemistry is a totally new component compared to CESM but again no evaluation of this important component of the ES model is given. Overall given this is an overview documentation paper of NorESM2 I feel an assessment of the global carbon cycle at the very least is warranted.

Similarly for aerosols, I note the authors cite other papers that are in preparation, however again as a top-level documentation paper and the importance of aerosol-climate interactions for the climate response of the model some overview of the performance of aerosols in the model is needed. In particular, both sea salt and DMS emissions have been used to tune the final coupled model but no detail of this tuning nor impact

on the aerosol simulation is provided.

Tuning: More detail and clarity is needed in some aspects of the tuning description. It is evident from Section 3 that a number of variables have been used to tune the low and higher resolution models (LM and MM) but the tunings differ in a number of places between the two resolutions. A table summarizing the primary and secondary tuning parameters as well as the untuned/tuned values chosen for each configuration would be very beneficial. It should include the impact of the tuned values on an appropriate metric which ideally would be constrained by observations eg: RESTROM or the SW cloud forcing in the case of the tuning of gamma parameter (paragraph beginning 266).

The tuning was carried out in a pre-industrial climate , yet the authors set their tuning targets in order "to maintain values of mean atmospheric and ocean temperatures close to observations" (L254) , given that the observations are predominantly in the present-day the authors should comment on the limitations of any such comparison. Were any present-day simulations done in parallel to validate this tuning and evaluation?

You state (Line 224) that present-day year 2000 AMIP timeslices were used for the general development of CAM6-Nor. Given that a continuous year 2000 forcing is not a realistic representation of present-day climate or of observations over recent decades, can the authors comment on the decision to use year 2000 forcing instead of a timeseries forcing and implications this may have on the model development and evaluation.

Overall the different choice of tuning parameters for the two resolutions will impact the models evolution and therefore limits the assessment of the role of resolution on the model performance and any potential benefits of the higher resolution. Can the authors comment on this ?

Line 250: What were the main changes from CESM2 –> CESM2.1? These should be documented, perhaps in Section 2 and here in the Tuning section document the impact of these developments on the tuning and development of NorESM2.

Some sort of schematic or diagram (even if included in an appendix) would be useful to aid understanding of the spin-up process, detailing offline spin-up and fully coupled spin-up. How long was the total spin-up period of the final tuned model before the official piControl for the DECK began? Later in Section 4.1 you state the abrupt 4xCO2 and 1pctCO2 runs were started at year 1 of the control -presumably you mean here the piControl but there's no indication of the full length of spin-up and how you determined that the model was fully spun-up.

Ensemble size: It is very hard to assess the robustness of the NorESM2-MM model given there is only a single historical member used in this analysis. The authors need to acknowledge such limits in the text. Indeed, the historical evolution of global mean surface temperature is outside of the LM model range but its impossible to say if this is meaningful. Furthermore, it's not clear if only single ensemble members were run for the future projections in both models, it looks like this is the case but again it needs to be clearly stated and limitations on conclusions drawn need to be discussed.

It is very interesting how the ECS is so remarkably different in NorESM2 compared to CESM2. Can the authors expand on the detail given here? Have additional sensitivity experiments been done to pick apart the role of the differences between the two models on the response eg, ocean model, aerosol-cloud representation? It seems a very relevant investigation to understand potentially more generally the multi-model differences in ECS, in particular given the tendency of CMIP6 models to move towards higher ECS it's all the more interesting that NorESM2 has gone the other way.

Line 402: What is NorESM1-Happi? Given the prevalence of its usage in the analysis of the climate response and present day performance of the NorESM2, an appropriate reference and a brief description of how it's different from NorESM1 is warranted Overall, I don't get the motivation for including NorESM1-Happi in the analysis and find it often confuses the analysis. . Also, Presumably the NorESM1 models are not driven with the updated CMIP6 forcing for instance?

The final "Summary and Discussion" reads really just solely as a Summary. Can the authors draw some overarching Conclusions from their analysis, which might include for instance the overall improved performance of the NorESM2 models compared to its predecessors; the motivation of 2 resolutions for NorESM2, role of resolution on model performance and potential benefits or applications of the different resolutions to different aspects of wider CMIP6 analysis. The would be beneficial to a reader and future potential user of NorESM2 data.

Length of Paper: This paper is very long. It appears to me that many of the figures are surplus to the requirements of the main thread of the analysis and don't get discussed much in the main text I would recommend moving some of them to a supplementary material. The ones that strike me are Figures 15, 21, 22, and a reduction in the number of ENSO plots (currently Figures 26-30).

Technical Comments:

The Title should reflect that use is made use of ScenarioMIP simulations also.

Line 46: participates –> participate

Line 80: It would be nice to inform the reader why the decision was made not to include land-ice model.

Line115: Please include an appropriate reference for the prescribed optical properties used for the stratospheric aerosol.

line 121: Can the authors quantify the impact of tuning both the sea salt (and later the DMS) on the total emissions of these natural sources and perhaps AOD and cloud droplet numbers? Were the aerosol tunings applied in the same way in both LM and MM?

Sect 2.4: does the dust aerosol impact the iron fertilisation of the ocean?

Line 224: I don't understand what is meant by a "data-atmosphere" for the offline forcing of the ocean components – presumably the atmospheric forcing came from the CAM6-Nor simulations?

Line 246: "towards its climatology" –> towards its own climatology

Line 252: steady –> steady-state?

Line 266: Define the acronym CLUBB

Line 281: "detail here" appears to be inserted here in error. Some quantification of the change in seasalt and DMS emissions should be given.

Line 300: The lower biological production of DMS is said to agree better with observations but leads to an underestimation in DMS emissions. What observations are used here? How do the authors explain this apparent discrepancy?

Line 305: greenhouse gas climate scenarios → greenhouse gas future climate scenarios.

Lines 308—310: Please make sure you include the appropriate references here for the CMIP6 DECK and ScenarioMIP experimental protocols.

Line 330: were there any particular criterion used for the choice of initialisation years ?

Line 331: it seems a great shame that there is only one ensemble member of NorESM2-MM included in this analysis and severely limits any conclusions drawn about the performance of this model compared to the LM equivalent. If additional members can be added this would significantly add to the value of the analysis. If not, the limitations of having only 1 ensemble member should be highlighted in the text.

Line 334: future climate development –> future climate change?

Line 338: RCPs – define acronym

Line 354: please define what you mean by a "sufficiently long spin-up"

Line 371: abrupt4xCO2 –> abrupt-4xCO2

Line 420: Does the model have a representation of nitrate aerosol? "it is likely that the aerosol forcing is similar in both model versions" – can you actually make this statement given the different cloud tuning in the two configurations with potential implications for the marine stratocumulus for instance?

Line 473 "sea level is lower" –> sea level anomaly is lower.

Line 488: Fig 9 here I think should refer to Fig 20

Line 489 Fig 10 –> Fig 11

Line 550: North Atlantic –> North American continent (also biases prevalent in south America).

Line 554: "are mitigated"–> are reduced

Figure 14: Why is the LM model so much warmer (at the surface) than MM? Is this a consequence of the different tunings? Its hard to tell from Figure 2 if the net TOA in the piControl is overall warmer in the LM model.

Line 625 "modelled cloud cover" – presumable the 70% here is referring to a global mean value. It would be helpful to refer the reader back to Table 2 here.

Line 626: The reference to Fig 15 seems to be oddly placed in between Fig 19 and Fig 20. I suggest moving the location of the figure.

Line 633: "reanalysis. along"? This whole sentence needs to be corrected as Figure 20 does not use GPCP data.

Figure 20: Either the figure caption of the figure labelling is incorrect here in terms of what model is plotted in what panel, please correct.

Line 695: friquency –> frequency

Line 700: "NorESM-LM (Figure 25(b)) –> this is inconsistent with the Figure caption of Figure 25 which states panel b = NorESM-MM. Please double check all figure captions

to make sure they are correct and consistent with the main text.

Figures 26 – 29 : It feels like there are a disproportionately a lot of figures for Section 5.9. Are all these needed in the main text, can some be moved to supplementary section?

Line 751: "medium-resolution version of the model" – you should clearly state here that the resolution differences here relate to the atmosphere.

Figure 8: March and September lines should me clearly marked on the plots.

Figure 25: In the figure caption the final sentence is incomplete.
* * *

---

## Referee Comment (RC2) · Anonymous Referee #2 · 15 May 2020

The paper is a high-level description of development, tuning, and key CMIP6 simulations of NorESM2. That includes a discussion of climate sensitivity and several aspects of the climatological state of the model.

The paper is half-way between an overview paper and an evaluation paper. It works well as an overview, covering the main development activities, simulations and results. I like the openness of the description of tuning strategies. Sharing components with CESM2 brings the interesting aspect of the impact of ocean/etc. on different on key metrics like sensitivity, which may provide interesting opportunities for new insights.

The paper does not work as an evaluation paper. The evaluation mostly looks at phys-

ical aspects (radiation, clouds, ocean state, sea ice, ENSO) and the more "Earth system" components are not evaluated at all. My suggestion is to focus on the overview, delegating the evaluation of specific components to companion papers. The present paper should then be re-titled and reframed, with minimal effort, as an overview paper only. This reframing would be a good opportunity to make section 5 more balanced in terms of text-to-figure ratio: many figures are only briefly mentioned in the text, so could go.

**1  Specific comments**

- Caption of Figure 1: The information given in parentheses could more efficiently be put in the boxes directly.

- Line 64: I'm curious to know how those modifications were chosen. In response to perceived deficiencies in CESM2? Different scientific priorities? Ad-hoc developments that happened to be ready?

- Line 99: That paragraph would be a good place to say what the time step of the different models is.

- Line 99: That paragraph could be organised more efficiently. Related statements should be grouped together, for example all statements related to emissions; then chemistry; then volcanic forcing; then optical properties. Bullet points would work well here.

- Lines 123-124: What do you mean? The model should not cool in such simulations... Do you mean improve the radiative balance of the model?

- Line 128: Kirkevag 2018 is unclear as to what particles acted as coagulation sink in the previous version. It should be clarified here.

- Line 129: "a more realistic rate" What was the previous value? How big is the change?

- Line 139: How is the mean cloud-free relative humidity calculated? Assuming 100% RH in the cloudy part?

- Line 254: Need to clarify your secondary tuning target. Was it absolute temperature of the preindustrial state, the present-day state, or present-day temperature anomalies? The latter two imply a tuning of the response.

- Paragraphs starting lines 270 and 275: Those two paragraphs are confusing. Which changes made it and which didn't?

- Line 272: "the final parameter values" – might as well give those values here.

- Line 281: That statement looks incomplete.

- Lines 366-367: Is that drift related to the ocean temperature drift?

- Lines 379: Should cite the examples of long equilibrium studies.

- Lines 389-391: It would be useful to show that 500-year simulation on Fig 3. Is there a change in warming rate at some point in time, or is it just a question of time to equilibrium?

- Line 396: I suppose that the slower warming in NorESM2 means that its TCR is lower than that of CESM2?

- Line 417: Is that really the explanation? Isn't it normally a good thing to have a low climate sensitivity when having a strong forcing?

- Line 418: Perhaps say that this is the effective radiative forcing

- Line 425: A good way to summarise the numbers in that paragraph is that the absolute temperature simulated by MM is almost 1 degree warmer than LM throughout the 1850-2100 period, but anomalies are similar.

- Line 468: Although I do not have specific comments on section 5, that section needs to focus on main results only, clearly summarising which model/model and model/obs differences are understood, which are not, and which differences affect the model response to forcing.

- Paragraphs starting lines 436 and 444 and Figures 6-7: SSP126 looks like an outlier in a couple of these timeseries. Is that just variability among ensemble, or is there something more than that?

- Figure 15 should be re-numbered, as it is used after Figure 19.

**2   Technical comments**

- Line 15: Satisfactorily -> satisfactory

- Line 47: Delete "Also"

- Line 793: Typo "properties"

---

## Author Comment (AC1) · 26 Jun 2020

General Comments: This manuscript documents the NorESM2 model, which is being used in CMIP6. The model is based on the CESM2 model with some notable differences in for instance, the representation of aerosols and their climate interactions; convection parameterizations as well as using different physical ocean and ocean bio-geochemistry models. A description of the model and its differences from CESM and CAM6 models are detailed and an overview of the development and tuning of the fully coupled configuration in a pre-industrial climate is given. An assessment of some key climate responses is provided, including the equilibrium climate sensitivity and transient climate response as well as future climate projections. An overview of the present-day performance of the historical simulations against observations is also provided.

This is a useful description and overview paper of the NorESM2 model and its climate characteristics and will be a very useful reference for the on-going and future CMIP6 analysis work. It is suitable for publication in GMD although I find it in its current form very long with 30 figures. The authors have included a lot in this one manuscript, covering a wide range of the model assessment with the result that the analysis of the individual components feels quite "light touch while I feel it is still missing evaluations of important parts of the fully coupled Earth system. I would recommend publication in GMD after my recommended revisions and additions to the analysis are made which I outline below.

> **We see the point that there is sometimes a mismatch between a too detailed analysis on some parts of the model and not much information on other parts. We propose to move some of the figures and relevant analysis into a supplement. Details are given in the answers to the specific comments. We have also made an attempt to include more information on the carbon cycle and aerosol parameterisations in the model but this had to be done in a way that does not take away points from the specific papers planned / or published on these topics. Details under specific comments**

Specific Comments:

There is no evaluation of aerosols (even though this is a key difference from the CESM2 model) or ocean or terrestrial biogeochemistry provided. While the land model is essentially CLM5 and documented elsewhere the authors note the important implications of the updated nitrogen-carbon limitation on the carbon cycle in the model but no assessment of the carbon cycle is provided. The ocean biogeochemistry is a totally new component compared to

CESM but again no evaluation of this important component of the ES model is given. Overall given this is an overview documentation paper of NorESM2 I feel an assessment of the global carbon cycle at the very least is warranted.

> **A full description and evaluation of the ocean biogeochemistry in NorESM2 has been documented in Tjiputra et al. (2020). In the revised manuscript, we will include the following paragraph summarizing its tuning approach and key performance.**
>
> **"The ocean biogeochemistry in NorESM2 (iHAMOCC) was tuned based on three objectives: (i) to improve the interior distribution of biogeochemical tracers, (ii) to better represent the upper ocean seasonal cycle, and (iii) to achieve equilibrium air-sea CO2 fluxes under the pre-industrial setting. Due to the identical ocean component between NorESM2-LM and NorESM2-MM, the performance in ocean biogeochemistry is very similar in both model versions. Compared to NorESM1, the climatological interior concentrations of oxygen, nutrients, and dissolved inorganic carbon have improved considerably in NorESM2. This is mainly due to the improvement in the particulate organic carbon sinking scheme, allowing more efficient transport and remineralization of organic materials in the deep ocean. The seasonal cycle of air-sea gas exchange and biological production at extratropical regions was improved through tuning of the ecosystem parameterizations. The simulated long-term mean of sea-air $CO_2$ fluxes under the pre-industrial condition in NorESM2-LM is -0.126 ± 0.067 Pg C $yr^{-1}$. Under the transient historical simulation, the ocean carbon sink increases to 1.80 and 2.04 Pg C $yr^{-1}$ in the 1980s and 1990s, which is well within the present day estimates. A more detailed evaluation of iHAMOCC performance in NorESM2 is available in Tjiputra et al. (2020)."**
>
> **In the land model description the following information will be added**
>
> **"An overview of gross primary productivity (GPP) and soil and vegetation carbon pools are provided in a table in the supplementary material, showing a substantially better agreement with observations for both resolutions of NorESM2 than NorESM1. Highest agreement with observations and smallest differences between resolutions are found for GPP and vegetation carbon, whereas a considerable negative bias is found for soil carbon in both NorESM2 versions. These results broadly agree with results from offline (land only) simulations with CLM described by Lawrence et al. (2019), who also describe the individual model updates from CLM4 (used in NorESM1) to CLM5."**

Similarly for aerosols, I note the authors cite other papers that are in preparation, however again as a top-level documentation paper and the importance of aerosol-climate interactions for the climate response of the model some overview of the performance of aerosols in the model is needed. In particular, both sea salt and DMS emissions have been used to tune the final coupled model but no detail of this tuning nor impact on the aerosol simulation is provided.

> **We will add two figures in the Supplementary material to document the performance of aerosols. The first figure shows a comparison of aerosol optical depth (AOD) at 550 nm between NorESM and observations for the period 2005--2014. The second figure shows global mean time series over the historical period of AOD and ERF.**
>
> **The following will be added to the manuscript:**
>
> **"Figure Sxx shows a comparison of observed and modelled aerosol optical depth at 550 nm for NorESM2-LM at AERONET sites (Holben et al., 1998), based on the years 2005--2014 of the historical experiment (member 1), indicating a reasonable agreement between the model simulations and observations."**

**"Figure Syy (a) shows the evolution of the global mean aerosol optical depth (stratospheric aerosol excluded) in the historical simulation for NorESM2-LM (member 1). It indicates that AOD has steadily increased over the historical period from a value of around 0.14 in 1850 to 0.17 at the end of the simulation. Figure Syy (b) shows the historical evolution of the aerosol effective radiative forcing for NorESM2-LM. This forcing is derived from comparing a three-member ensemble of AMIP simulations with only aerosol emissions evolving over the historical period (piClim-histaer) with a reference simulation under pre-industrial conditions (piClim-control). The aerosol forcing strengthens from around -0.3 W/m2 around 1930 to -1.5 W/m2 in the period 1970--2000, becoming slightly weaker again in 2014. In addition, 30 year-long AMIP simulations have been run with only aerosol emissions representative for the year 2014, both with NorESM2-LM and NorESM2-MM (piClim-aer). Averaging over the last 25 years of these simulations and comparing it with piClim-control, we obtained an aerosol ERF for 2014 of -1.36+/-0.05 W/m2 in NorESM2-LM and -1.26+/-0.05 W/m2 in NorESM2-MM."**

**A more detailed description of the tuning / modifications of sea-salt and DMS will be provided in the relevant sections. Suggestion below under the specific comments.**

Tuning: More detail and clarity is needed in some aspects of the tuning description. It is evident from Section 3 that a number of variables have been used to tune the low and higher resolution models (LM and MM) but the tunings differ in a number of places between the two resolutions. A table summarizing the primary and secondary tuning parameters as well as the untuned/tuned values chosen for each configuration would be very beneficial.

**A table of all the tuning parameters for both resolution versions will be included in the supplementary material. The table will also include the corresponding values from CESM2 if applicable.**

It should include the impact of the tuned values on an appropriate metric which ideally would be constrained by observations eg: RESTROM or the SW cloud forcing in the case of the tuning of gamma parameter (paragraph beginning 266).

**Time series of RESTOM, SW and LW cloud forcing, temperatures, AMOC, sea salt and DMS emissions are now included in the supplement for both model versions.**

The tuning was carried out in a pre-industrial climate , yet the authors set their tuning targets in order "to maintain values of mean atmospheric and ocean temperatures close to observations" (L254) , given that the observations are predominantly in the present- day the authors should comment on the limitations of any such comparison. Were any present-day simulations done in parallel to validate this tuning and evaluation?

You state (Line 224) that present-day year 2000 AMIP timeslices were used for the general development of CAM6-Nor. Given that a continuous year 2000 forcing is not a realistic representation of present-day climate or of observations over recent decades, can the authors comment on the decision to use year 2000 forcing instead of a timeseries forcing and implications this may have on the model development and evaluation.

**The entire section will be rewritten in order to give a better description of the development with respect to present and pre-industrial conditions. This will be included in the manuscript.**

**"Similar to CESM, NorESM2 adjusted towards its coupled climatology with an initial phase of strong cooling in the high latitudes of the northern hemisphere, after which an intensification of ocean heat advection stabilised the simulation. After that point, the climatology tended to settle to a steady drift. During major tuning steps, the coupled model had to be restarted from the initial state several times. In order to save computer resources, minor tuning, especially toward balanced RESTOM, was**

**performed on the best-candidate simulation after this initial, large adjustment. Alongside the final tuning, the CESM components were updated to the versions found in CESM2.1. In this second phase of coupled spin-up, it was found that the sensitivity of some aspects of the simulated coupled climatology, including RESTOM, to small changes in parameters or parameterisations could be different than that found in stand-alone simulations of the individual components with prescribed boundary conditions.The coupled response could be both amplified or damped with respect to single-component simulations. As a result, the fine parameter tuning of the model had to be performed in coupled mode."**

**The main goal of the coupled tuning process was to create an energetically balanced pre-industrial control simulation with a reasonably stable state. The simulation can produce a steady climatology only if the time-average radiative imbalance on the top of the model (RESTOM) vanishes. In practice, a commonly used target is for RESTOM to be within ±0.1 W m$^{-2}$. This is the primary coupled tuning target. Secondary tuning targets are to obtain and maintain values of mean atmospheric and ocean temperatures close to observations. Each change in the coupled model was therefore tested in parallel in atmosphere-only (AMIP) and ocean-only (OMIP) mode. As both the ocean heat gain and tropospheric air temperature, humidity and cloudiness are closely associated with the top of the atmosphere imbalance, the two requirements are strongly connected. The coupled fine tuning should not significantly degrade other important climatological variables such as temperature, precipitation, cloud, or the main mode of coupled variability, i.e. ENSO. This parallel testing procedure ensured that the model simulation maintained a degree of consistency both with the present-day, observed climatology, and with a steady pre-industrial climate. Where available, notably in SST and sea-ice, observational estimates of the state of Earth's pre-industrial climate were also considered against the coupled integrations.**

Overall the different choice of tuning parameters for the two resolutions will impact the models evolution and therefore limits the assessment of the role of resolution on the model performance and any potential benefits of the higher resolution. Can the authors comment on this ?

**We believe that a similar and balanced pre-industrial state is more important than the tuning of a limited number of parameters, but hard to prove otherwise. Our evaluation against observations showed more realism for simulations with a higher resolution, showing its benefit. We will change the text as follows:**

**"Each tuning step was performed in isolation, and an effort was made to ensure the greatest possible similarities wrt to cloud forcing in the two model configurations LM and MM…..We kept the same parameter values with the exception of gamma, ice cloud cover parameterisation and a small adjustment of the DCS parameter as defined in the tuning table in the supplementary material."**

Line 250: What were the main changes from CESM2 –> CESM2.1? These should be documented, perhaps in Section 2 and here in the Tuning section document the impact of these developments on the tuning and development of NorESM2.

**We will include some information on this update. As the change was done quite early in the spin-up we do not think this had any impact on the control period:**

**"The changes from CESM2.0 to CESM2.1 are, according to Danabasoglu et al. (2020), mostly technical although with minor bug fixes and updated forcing fields. The switch was done in year 420 of the NorESM2-LM spin-up and thus 1000 model years were integrated after that and before the**

**start of the control. The impact on the global fields are quite small as can be seen in the spin-up figure in the supplementary material."**

Some sort of schematic or diagram (even if included in an appendix) would be useful to aid understanding of the spin-up process, detailing offline spin-up and fully coupled spin-up. How long was the total spin-up period of the final tuned model before the official piControl for the DECK began? Later in Section 4.1 you state the abrupt 4xCO2 and 1pctCO2 runs were started at year 1 of the control -presumably you mean here the piControl but there's no indication of the full length of spin-up and how you determined that the model was fully spun-up.

**A schematic of the tuning time-line will be included in the supplement.**

Ensemble size: It is very hard to assess the robustness of the NorESM2-MM model given there is only a single historical member used in this analysis. The authors need to acknowledge such limits in the text. Indeed, the historical evolution of global mean surface temperature is outside of the LM model range but its impossible to say if this is meaningful. Furthermore, it's not clear if only single ensemble members were run for the future projections in both models, it looks like this is the case but again it needs to be clearly stated and limitations on conclusions drawn need to be discussed.

**Due to very limited computer resources we were unable to run more ensemble members by the time the paper was first submitted. Two more NorESM2-MM ensemble members have been started and are expected to finish around 1st July. We will update the figures to include these results where relevant.**

It is very interesting how the ECS is so remarkably different in NorESM2 compared to CESM2. Can the authors expand on the detail given here? Have additional sensitivity experiments been done to pick apart the role of the differences between the two models on the response eg, ocean model, aerosol-cloud representation? It seems a very relevant investigation to understand potentially more generally the multi-model differences in ECS, in particular given the tendency of CMIP6 models to move towards higher ECS it's all the more interesting that NorESM2 has gone the other way.

**We agree with the reviewer that this is a very interesting difference. We will change the text and include further details from the paper Gjermundsen et al. (submitted).**

**"An extensive analysis of the low ECS value in NorESM2 is given in Gjermundsen et al. (submitted, 2020). Note that the aerosol forcing is not very different between NorESM2 and CESM2 and can not explain the discrepancy in ECS values. Several sensitivity experiments have been conducted and are reported in Gjermundsen et al. in order to investigate the importance of different ice cloud schemes, CLUBB and interactive DMS. However, these NorESM2 experiments exhibit similar ECS values. The main reason for the low ECS in NorESM2 compared to CESM2 is, how the ocean models respond to GHG forcing. The behaviour of the BLOM ocean model (compared to the POP ocean model used in CESM2), contributes to a slower surface warming in NorESM2 compared to CESM2. Using the Gregory et al. (2004) method on the first 150 years leads to an ECS estimate which is considerably lower than for CESM2. However, if 500 years are included in the analysis, NorESM2 shows a sustained warming similar to CESM2. This suggests that the actual equilibrium temperature response to a large GHG forcing (the value one finds when the model is run for many hundred years) in NorESM2 and CESM2 is not very different, but that the Gregory et al. (2004) method based on the first 150 years does not give a good estimate of ECS for models."**

Line 402: What is NorESM1-Happi? Given the prevalence of its usage in the analysis of the climate response and present day performance of the NorESM2, an appropriate reference and a brief description of how it's different from NorESM1 is warranted Overall, I don't get the motivation for including NorESM1-Happi in the analysis and find it often confuses the analysis. Also, Presumably the NorESM1 models are not driven with the updated CMIP6 forcing for instance?

> **NorESM1-Happi is now defined in the beginning of section 4. We also mention the differences between NorESM1-M and NorESM1-Happi and the motivation for including both versions of NorESM1 in the present study. By including both versions we can compare against NorESM1 versions and simulations having corresponding MM and LM resolutions. We will add the following to the manuscript:**

> **"The motivation for including NorESM1-Happi in the present paper is to present results from a low-resolution (-M) and medium-resolution version (-Happi) of NorESM1 alongside the results from the low-resolution (-LM) and medium-resolution versions (-MM) of NorESM2."**

The final "Summary and Discussion" reads really just solely as a Summary. Can the authors draw some overarching Conclusions from their analysis, which might include for instance the overall improved performance of the NorESM2 models compared to its predecessors; the motivation of 2 resolutions for NorESM2, role of resolution on model performance and potential benefits or applications of the different resolutions to different aspects of wider CMIP6 analysis. The would be beneficial to a reader and future potential user of NorESM2 data.

> **We agree that the conclusions may have been expressed a bit too implicitly to give clear recommendations on the use of the two different model versions. In section 2.1 we will include explicit timing information for the two resolutions and extended the sentence "Due to the generally high computational cost of NorESM2" with "and relatively limited amount of computing power"**

> **We will add recommendations for the use of the two model resolutions in the summary and discussion section:**

> **"In general the higher resolution model is closer to observations than the low resolution and has a better representation of regional climatological patterns e.g. extratropical storm tracks. This makes the higher resolution model version a better choice for studying regional responses and creating input fields for further downscaling of climate change scenarios. A notable exception is the time-evolution of sea-ice change which is closer to the observations in the low resolution version of the model. This is likely connected to the fact that the low resolution version is generally warmer in the pre-industrial state than the high resolution, which is associated with overall thinner sea ice and less summer extent that compares better with observational estimates.**

> **On the other hand we find that large scale forcings and responses to forcing including climate change scenarios are very similar for both resolutions. This indicates that the lower resolution version is a useful tool for the development of new parameterisations and testing the overall sensitivity to model parameters, since the overall impact is expected to be very similar in the higher resolution version of the model. The less computationally expensive LM version allowed us to participate in a significantly larger number of CMIP6 model experiments."**

> **Improvements of NorESM2 compared to NorESM1 are already mentioned in the summary and discussion, but we will complement and clarify this in the revised manuscript.**

Length of Paper: This paper is very long. It appears to me that many of the figures are surplus to the requirements of the main thread of the analysis and don't get discussed much in the main text I would recommend moving some of them to a supplementary material. The ones that strike me are Figures 15, 21, 22, and a reduction in the number of ENSO plots (currently Figures 26-30).

**The suggested figures 15, 21 and 22 will be moved to supplementary material. With respect to ENSO we will move Figures 26, 30 to the supplement, while leaving the discussion of ENSO unchanged in the text. This will be the only documentation of ENSO in this model for a while, so it is important to keep it there.**

Technical Comments:

The Title should reflect that use is made use of ScenarioMIP simulations also.

**We will add this.**

Line 46: participates –> participate

**This will be corrected.**

Line 80: It would be nice to inform the reader why the decision was made not to include land-ice model.

**We will add this information.**

**"Our tests with an interactive ice-sheet model over Greenland indicated that we are not able to maintain a realistic mass balance there, and so further development is needed. CESM performed some specific tuning there to achieve a better mass balance; the warmer regional climate of NorESM would have required a dedicated effort. Due to resource limitations we have postponed this until after CMIP6."**

Line115: Please include an appropriate reference for the prescribed optical properties used for the stratospheric aerosol."

**"Monthly distributions of stratospheric sulfate aerosols follow the CMIP6 recommendations: Concentration, surface area density, and volume density are based on the work of Thomason et al. (2018)."**

line 121: Can the authors quantify the impact of tuning both the sea salt (and later the DMS) on the total emissions of these natural sources and perhaps AOD and cloud droplet numbers? Were the aerosol tunings applied in the same way in both LM and MM?

**As described in the tuning section, the aerosol parameters are the same for both model resolutions.**

**With respect to sea-salt tuning the word tuning in this connection is imprecise since this change was included already prior to the spin-up.**

**The words "tuned up" will be replaced with "increased" and**

**"This has been done as a measure to help cool the model sufficiently in the spin-up and control simulation" will be replaced by "This was done in order to reduce the positive top of the atmosphere radiative imbalance."**

**Additional text.**

**"The sea-salt emission changes were tested in a predecessor model version, NorESM1.2 (Kirkevåg et al., 2018). Annual and globally averaged, this lead to increases from 99.5 to 228.3 ng/m2/s (129%)**

**in sea-salt emissions, from 7.8 to 17.2 mg/m2 (121%) in sea-salt column burdens, with corresponding changes in total clear-sky AOD from 0.086 to 0.119 (38%), and cloud droplet numbers at top of the cloud (using the method of Kirkevåg et al., 2018) changed from 31.3 to 32.7 cm-3 (4.5%).**

**In addition to this change it was considered to increase the emissions even more as shown in the tuning overview but this was not included in the control simulation**

**For a more detailed analysis of the impact of changes in the natural emissions please see Olivié et al. (2020)."**

Sect 2.4: does the dust aerosol impact the iron fertilisation of the ocean?

**No it doesn't. This will be specified in the text**

**"Currently the atmospheric deposition into the ocean is decoupled. The ocean biogeochemistry uses the monthly climatological aerial dust (iron) deposition of Mahowald et al. (2005)."**

**Mahowald, N., Baker, A., Bergametti, G., Brooks, N., Duce, R., Jickells, T., Kubilay, N., Prospero, J., and Tegen, I.: Atmospheric global dust cycle and iron inputs to the ocean, Global Biogeochem. Cycles, 19, 4025, https://doi.org/10.1029/2004GB002402, 2005.**

Line 224: I don't understand what is meant by a "data-atmosphere" for the offline forcing of the ocean components – presumably the atmospheric forcing came from the CAM6-Nor simulations?

**With "forced by a data-atmosphere" was meant "forced with prescribed atmosphere and runoff of the OMIP1 protocol", so not using forcing from CAM6-Nor. This will be made clear in the text.**

Line 246: "towards its climatology" –> towards its own climatology

**Will be corrected**

Line 252: steady –> steady-state?

**Will be replaced and slightly reformulated.**

Line 266: Define the acronym CLUBB

**CLUBB will be defined.**

Line 281: "detail here" appears to be inserted here in error. Some quantification of the change in seasalt and DMS emissions should be given.

**Will be done as described above in the comment related to "line 121".**

Line 300: The lower biological production of DMS is said to agree better with observations but leads to an underestimation in DMS emissions. What observations are used here? How do the authors explain this apparent discrepancy?

**The 'lower biological production here' refers to the primary production (PP) through phytoplankton photosynthesis, and not the DMS production. NorESM1 simulates too strong bias in its spring bloom PP in the high latitude, and is now alleviated in NorESM2. The statement**

**"Compared to Schwinger et al. (2017), NorESM2 has doubled the diatom-mediated DMS production parameter in order to maintain the observed high DMS concentration at high latitudes. This tuning is necessary due to the lower biological production simulated in NorESM2 (relative to NorESM1), which is a better representation to the observations, during spring bloom in both hemispheres (Tjiputra et al., 2019)."**

**Will be rephrased to**

**"Compared to Schwinger et al. (2017), NorESM2 has doubled the diatom-mediated DMS oceanic production parameter in order to maintain the observed high oceanic DMS concentration at high latitudes during spring and summer seasons in both hemispheres (Lana et al., 2011). This tuning is introduced because the primary production simulated in NorESM2 is lower than that in NorESM1 (Tjiputra et al., 2020). "**

**Lana, A., Bell, T. G., Simó, R., Vallina, S. M., Ballabrera-Poy, J., Kettle, A. J., Dachs, J., Bopp, L., Saltzman, E. S., Stefels, J., Johnson, J. E., and Liss, P. S.: An updated climatology of surface dimethylsulfide concentrations and emission fluxes in the global ocean, Global Biogeochem. Cy., 25, GB1004, https://doi.org/10.1029/2010GB003850, 2011.**

Line 305: greenhouse gas climate scenarios → greenhouse gas future climate scenarios.

> **We will fix this.**

Lines 308âAT310: Please make sure you include the appropriate references here for the CMIP6 DECK and ScenarioMIP experimental protocols.

> **We will add these references.**

Line 330: were there any particular criterion used for the choice of initialisation years ?

> **No particular criterion was used but clarification will be added**
>
> **"Using the first year for initialisation of control allows for prolongation of scenarios without increasing the length of piControl. The interval of circa 30 years between the initialisations of the individual members follow the typical climate statistics interval. Both choices are the same as used for the historical set-simulations used in NorESM1 for CMIP5. "**

Line 331: it seems a great shame that there is only one ensemble member of NorESM2-MM included in this analysis and severely limits any conclusions drawn about the performance of this model compared to the LM equivalent. If additional members can be added this would significantly add to the value of the analysis. If not, the limitations of having only 1 ensemble member should be highlighted in the text.

> **One more ensemble member has been run and one further member is running so we expect to be able to update the figures and text in the revised paper with two new members of the highest resolution.**

Line 334: future climate development –> future climate change?

**We will fix this.**

Line 338: RCPs – define acronym

**We will define the acronym.**

Line 354: please define what you mean by a "sufficiently long spin-up"

**The sentence will be deleted as it is superfluous. If the control simulation is stable enough fulfilling the requirements, the length of the spin-up is irrelevant.**

Line 371: abrupt4xCO2 –> abrupt-4xCO2

**This will be corrected.**

Line 420: Does the model have a representation of nitrate aerosol?

**No. A sentence stating "Nitrate aerosols are not included." will be added to the aerosol description"**

"it is likely that the aerosol forcing is similar in both model versions" – can you actually make this statement given the different cloud tuning in the two configurations with potential implications for the marine stratocumulus for instance?

**We will add (see above) the aerosol ERF values in text (obtained from comparing two RFMIP simulations, i.e. piClim-aer and piClim-control) : we find an ERF of -1.36 ±0.05 W/m$^2$ in NorESM2-LM and -1.26 ±0.05 W/m2 in NorESM2-MM in 2014 (compared to 1850).**

Line 473 "sea level is lower" –> sea level anomaly is lower.

**We will correct this.**

Line 488: Fig 9 here I think should refer to Fig 20

**Referred to both (surface salinity and total precipitation)**

Line 489 Fig 10 –> Fig 11

**We will correct this.**

Line 550: North Atlantic –> North American continent (also biases prevalent in south America).

**We will modify this as suggested**

Line 554: "are mitigated"–> are reduced

**We will change this.**

Figure 14: Why is the LM model so much warmer (at the surface) than MM? Is this a consequence of the different tunings? Its hard to tell from Figure 2 if the net TOA in the piControl is overall warmer in the LM model.

**The average 500 yr TOA imbalance in NorESM2-MM and LM is very similar (-0.057 W m⁻² for NorESM2-LM and -0.065 W m⁻² for NorESM2-MM). The small difference is not the cause for LM being warmer. A discussion will be added along the following arguments:**

**The stronger cold tropospheric and warm surface tropical bias of NorESM-LM compared with NorESM-MM is in line with the behaviour of both NorESM1 and CESM2. The systematic difference between the two atmosphere resolutions is also consistent between coupled and AMIP simulations, with CAM6-Nor significantly cooler at two degree resolution than at one degree resolution for the same SSTs and the same physics parameters. At the same time, tropospheric specific humidity (and, a fortiori, relative humidity) is higher in two degree resolution. Cooler tropospheric temperature and higher humidity both lead to higher corresponding RESTOM. The ultimate cause of this systematic dependence of the simulated climatology on the resolution of the atmosphere model is not known. There may be a sensitivity of the convection parameterization to the grid-scale variability of near-surface air parameters and to boundary stability. However, a more likely possibility is a resolution dependence of cloud microphysics and the efficiency of stratiform precipitation.**

Line 625 "modelled cloud cover" – presumable the 70% here is referring to a global mean value. It would be helpful to refer the reader back to Table 2 here.

**We will add this.**

Line 626: The reference to Fig 15 seems to be oddly placed in between Fig 19 and Fig 20. I suggest moving the location of the figure.

**Figure 15 will be moved.**

Line 633: "reanalysis. along"? This whole sentence needs to be corrected as Figure 20 does not use GPCP data.

**Figure 20 does use GPCP data, but there was an error in the caption. Both the caption and the sentence (L633 in the submitted version) will be corrected.**

Figure 20: Either the figure caption of the figure labelling is incorrect here in terms of what model is plotted in what panel, please correct.

**This will be corrected.**

Line 695: friquency –> frequency

**This will be corrected.**

Line 700: "NorESM-LM (Figure 25(b)) –> this is inconsistent with the Figure caption of Figure 25 which states panel b = NorESM-MM. Please double check all figure captions to make sure they are correct and consistent with the main text.

**We will correct this.**

Figures 26 – 29 : It feels like there are a disproportionately a lot of figures for Section 5.9. Are all these needed in the main text, can some be moved to supplementary section?

**Figure 26 and 30 will be moved to the supplement.**

Line 751: "medium-resolution version of the model" – you should clearly state here that the resolution differences here relate to the atmosphere.

**This will be included in the text.**

Figure 8: March and September lines should me clearly marked on the plots.

**Will be corrected before final publication.**

Figure 25: In the figure caption the final sentence is incomplete.

**This will be corrected.**

**Reply to anonymous Referee #2**

The paper is a high-level description of development, tuning, and key CMIP6 simulations of NorESM2. That includes a discussion of climate sensitivity and several aspects of the climatological state of the model.

The paper is half-way between an overview paper and an evaluation paper. It works well as an overview, covering the main development activities, simulations and results. I like the openness of the description of tuning strategies. Sharing components with CESM2 brings the interesting aspect of the impact of ocean/etc. on different on key metrics like sensitivity, which may provide interesting opportunities for new insights.

The paper does not work as an evaluation paper. The evaluation mostly looks at physical aspects (radiation, clouds, ocean state, sea ice, ENSO) and the more "Earth system" components are not evaluated at all. My suggestion is to focus on the overview, delegating the evaluation of specific components to companion papers. The present paper should then be re-titled and reframed, with minimal effort, as an overview paper only. This reframing would be a good opportunity to make section 5 more balanced in terms of text-to-figure ratio: many figures are only briefly mentioned in the text, so could go.

> **The title and introduction have been modified to give more focus to the general overview of the model. Title reads now: "Overview of The Norwegian Earth System Model, NorESM2, and Key Climate Response of CMIP6 DECK, Historical, and Scenario Simulations". Some of the more detailed analysis e.g. seasonal precipitation cycle and some of the ENSO aspects have been moved to a supplement. More details on which and how figures are moved are given in the answers to reviewer 1.**

**1 Specific comments**

Caption of Figure 1: The information given in parentheses could more efficiently be put in the boxes directly.

**We feel that including more text within the figure itself would make it more messy and therefore prefer to keep it as it is.**

Line 64: I'm curious to know how those modifications were chosen. In response to perceived deficiencies in CESM2? Different scientific priorities? Ad-hoc developments that happened to be ready?

**Our main initial thrust with regard to CAM dynamics and formulation was directed at improving the local and global conservation properties of the model, and, in a related way, to remove obvious model resolution dependencies, in the belief that this might also bring advantages to the fidelity of the coupled model to observations at both targeted resolutions. Scientific priorities of our own were to include the CAM Oslo aerosol scheme, allowing for coupling with marine biogeochemistry (DMS) not part of CESM along with associated further adjustments to emissions and fluxes.**

Line 99: That paragraph would be a good place to say what the time step of the different models is.

**We will add the following text:**

**"The 30 minute timestep of the atmospheric model physics is the same for both atmospheric resolutions and the same as in CAM6."**

Line 99: That paragraph could be organised more efficiently. Related statements should be grouped together, for example all statements related to emissions; then chemistry; then volcanic forcing; then optical properties. Bullet points would work well here.

**We will rewrite this paragraph.**

Lines 123-124: What do you mean? The model should not cool in such simulations... Do you mean improve the radiative balance of the model?

**We will rewrite this as follows:**

**"This was done as a first attempt at reducing the large positive top of the model radiative imbalance of the model prior to start of the spin-up integration, "**

Line 128: Kirkevag 2018 is unclear as to what particles acted as coagulation sink in the previous version. It should be clarified here.

**In the previous version only the fine mode of co-nucleated sulfate and SOA (mixture no. 1) acted as a coagulation sink for the 2 nm particles. The text will be changed accordingly.**

Line 129: "a more realistic rate" What was the previous value? How big is the change?

**In NorESM1.2 the survival rates in the lower troposphere changed from typically 20 - 80% to 1 - 20% (zonally and annually averaged). Whether this is closer to observations or not, has to be checked more closely. From the literature we found that Kuang et al. (2009) inferred survival probabilities from size distribution measurements and found that at least 80 % of the nucleated particles measured at Atlanta, GA and Boulder, CO were lost by coagulation before the nucleation mode reached CCN sizes in the cases that they studied (20 % survival probability), even during days with high growth rates."**

Line 139: How is the mean cloud-free relative humidity calculated? Assuming 100% RH in the cloudy part?

**We will add the following:**

**"The cloud-free relative humidity is calculated assuming 100 % relative humidity in the cloudy volume."**

Line 254: Need to clarify your secondary tuning target. Was it absolute temperature of the preindustrial state, the present-day state, or present-day temperature anomalies? The latter two imply a tuning of the response.

**We will give a fuller explanation in the text. Basically, at every coupled tuning step towards reducing TOA imbalance in the pre-industrial climate, we used parallel stand-alone atmosphere and ocean integration to validate against present-day, observed climate. In essence, we tuned TOA in coupled mode, and state in stand-alone mode.**

Paragraphs starting lines 270 and 275: Those two paragraphs are confusing. Which changes made it and which didn't?:

**The paragraph starting at line 270 will be changed to.**

**"Given the same gamma values and otherwise identical parameter values the RESTOM was higher in the low-resolution version of the model"**

**We will add the following sentence prior to paragraph at 265:**

**"One disadvantage of reducing the gamma value in order to reduce RESTOM was that the short-wave cloud forcing (SWCF) becomes too negative."**

Line 272: "the final parameter values" – might as well give those values here.

**This will be added along with a reference to the tuning table in the supplementary material**

Line 281: That statement looks incomplete.

**The whole paragraph will be reformulated as follows:**

**"A more effective tuning of low-cloud radiative effects was achieved by modifying air-sea fluxes of DMS. Compared to Schwinger et al. (2017), NorESM2 has doubled the diatom-mediated DMS production parameter in order to maintain the observed high DMS concentration at high latitudes. This tuning is necessary due to the lower biological production simulated in NorESM2 (relative to NorESM1), which is a better representation to the observations, during spring bloom in both hemispheres (Tjiputra et al., 2019). A similar effect was found by increasing the sea-salt emissions but this was not included in the final version. As described in Section 2.1 the disadvantage of increasing the sea-salt flux was that even before the final tuning process, the existing parameterisation resulted in a too dominant marine aerosols with respect to optical thickness and surface mass concentrations.**

Lines 366-367: Is that drift related to the ocean temperature drift?

**Most of the remaining drift in ocean biogeochemistry variables is likely not very dependent on the ocean temperature drift.**

Lines 379: Should cite the examples of long equilibrium studies.

**Paynter et al. (2018) show results from simulations with GFDL-CM3 and GFDL-ESM2 run for more than 4000 years. We will add this reference in the text.**

Lines 389-391: It would be useful to show that 500-year simulation on Fig 3. Is there a change in warming rate at some point in time, or is it just a question of time to equilibrium?

**An extensive analysis of the low ECS, including time series of temperature, is given in Gjemundsen et al 2020 (submitted). There is no substantial change in the warming rate in NorESM2 (except for the first 20 years compared to the later), but the equilibrium time scale differs substantially from CESM2. The text was unclear on this point. We will change the text but leave further details to the paper Gjermundsen et al. (submitted).**

**We will modify the text as follows:**

**"An extensive analysis of the low ECS value in NorESM2 is given in Gjermundsen et al. (submitted, 2020). Note that the aerosol forcing is not very different between NorESM2 and CESM2 and can not explain the discrepancy in ECS values. Several sensitivity experiments have been conducted and are reported in Gjermundsen et al. in order to investigate the importance of different ice cloud schemes, CLUBB and interactive DMS. However, these NorESM2 experiments exhibit similar ECS values. The main reason for the low ECS in NorESM2 compared to CESM2 is, how the ocean models respond to GHG forcing. The behaviour of the BLOM ocean model (compared to the POP ocean model used in CESM2), contributes to a slower surface warming in NorESM2 compared to CESM2. Using the Gregory et al. (2004) method on the first 150 years leads to an ECS estimate which is considerably lower than for CESM2. However, if 500 years are included in the analysis, NorESM2 shows a sustained warming similar to CESM2. This suggests that the actual equilibrium temperature response to a large GHG forcing (the value one finds when the model is run for many hundred years) in NorESM2 and CESM2 is not very different, but that the Gregory et al. (2004) method based on the first 150 years does not give a good estimate of ECS for models."**

Line 396: I suppose that the slower warming in NorESM2 means that its TCR is lower than that of CESM2?

**Yes, that is correct. We will add the following sentence:**

**"The TCR of both NorESM2-LM and NorESM2-MM are lower than the value of 2.0 K found for CESM2."**

Line 417: Is that really the explanation? Isn't it normally a good thing to have a low climate sensitivity when having a strong forcing?

**We agree that the mentioning of low climate sensitivity is not helpful here, and the reasons for the 50s cooling is currently under discussion: The sentence shall read: "The cooling over the period 1930–1970 in NorESM2 is probably caused by a relatively strong negative aerosol forcing.**

Line 418: Perhaps say that this is the effective radiative forcing

**This will be included.**

Line 425: A good way to summarise the numbers in that paragraph is that the absolute temperature simulated by MM is almost 1 degree warmer than LM throughout the 1850-2100 period, but anomalies are similar.

**The sentence "Although the historical warming is slightly weaker in NorESM2-MM compared to NorESM2-LM, the warming at the end of the 21st century is rather similar in both versions of NorESM2." will be replaced with "The absolute temperature simulated by LM is almost 1 degree warmer than MM throughout the 1850-2100 period, but anomalies are similar."**

Line 468: Although I do not have specific comments on section 5, that section needs to focus on main results only, clearly summarising which model/model and model/obs differences are understood, which are not, and which differences affect the model response to forcing.

**We will strengthen the focus of the manuscript on the main results, and move some of the detailed analysis to the supplementary material.**

Paragraphs starting lines 436 and 444 and Figures 6-7: SSP126 looks like an outlier in a couple of these timeseries. Is that just variability among ensemble, or is there something more than that?

**Only one realisation of each scenario makes it uncertain if it is only internal variability or not.**

Figure 15 should be re-numbered, as it is used after Figure 19.

**Figure 15 will be re-numbered.**

**2 Technical comments**

Line 15: Satisfactorily -> satisfactory

**This will be corrected.**

Line 47: Delete "Also"

**This will be deleted.**

Line 793: Typo "properties"

**This will be corrected.**

---

## Author Response (AR1)

General Comments: This manuscript documents the NorESM2 model, which is being used in CMIP6. The model is based on the CESM2 model with some notable differences in for instance, the representation of aerosols and their climate interactions; convection parameterizations as well as using different physical ocean and ocean bio-geochemistry models. A description of the model and its differences from CESM and CAM6 models are detailed and an overview of the development and tuning of the fully coupled configuration in a pre-industrial climate is given. An assessment of some key climate responses is provided, including the equilibrium climate sensitivity and transient climate response as well as future climate projections. An overview of the present-day performance of the historical simulations against observations is also provided.

This is a useful description and overview paper of the NorESM2 model and its climate characteristics and will be a very useful reference for the on-going and future CMIP6 analysis work. It is suitable for publication in GMD although I find it in its current form very long with 30 figures. The authors have included a lot in this one manuscript, covering a wide range of the model assessment with the result that the analysis of the individual components feels quite "light touch while I feel it is still missing evaluations of important parts of the fully coupled Earth system. I would recommend publication in GMD after my recommended revisions and additions to the analysis are made which I outline below.

> **We see the point that there is sometimes a mismatch between a too detailed analysis on some parts of the model and not much information on other parts. We propose to move some of the figures and relevant analysis into a supplement. Details are given in the answers to the specific comments. We have also made an attempt to include more information on the carbon cycle and aerosol parameterisations in the model but this had to be done in a way that does not take away points from the specific papers planned / or published on these topics.** Specific Comments:

There is no evaluation of aerosols (even though this is a key difference from the CESM2 model) or ocean or terrestrial biogeochemistry provided. While the land model is essentially CLM5 and documented elsewhere the authors note the important implications of the updated nitrogen-carbon limitation on the carbon cycle in the model but no assessment of the carbon cycle is provided. The ocean biogeochemistry is a totally new component compared to CESM but again no evaluation of this important component of the ES model is given. Overall given this is an

overview documentation paper of NorESM2 I feel an assessment of the global carbon cycle at the very least is warranted.

**A full description and evaluation of the ocean biogeochemistry in NorESM2 has been documented in Tjiputra et al. (2020). In the revised manuscript, we have included the following paragraph summarizing the key performance.**

**"Due to the identical ocean component between NorESM2-LM and NorESM2-MM, the performance in ocean biogeochemistry is very similar in both model versions. Compared to NorESM1, the climatological interior concentrations of oxygen, nutrients, and dissolved inorganic carbon have improved considerably in NorESM2. This is mainly due to the improvement in the particulate organic carbon sinking scheme, allowing more efficient transport and remineralization of organic materials in the deep ocean. The seasonal cycle of air-sea gas exchange and biological production at extratropical regions was improved through tuning of the ecosystem parameterizations. The simulated long-term mean of sea-air $CO_2$ fluxes under the pre-industrial condition in NorESM2-LM is -0.126 ± 0.067 Pg C $yr^{-1}$. Under the transient historical simulation, the ocean carbon sink increases to 1.80 and 2.04 Pg C $yr^{-1}$ in the 1980s and 1990s, which is well within the present day estimates. A more detailed evaluation of iHAMOCC performance in NorESM2 is available in Tjiputra et al. (2020)."**

**In the land model description the following information will be added**

**"An overview of gross primary productivity (GPP) and soil and vegetation carbon pools are provided in Table 3, showing a substantially better agreement with observations for both resolutions of NorESM2 than NorESM1. There is consistency between observations and model simulations at different resolutions for GPP and vegetation carbon, whereas both NorESM2 model versions have a negative bias in soil carbon. These results broadly agree with results from offline (land only) simulations with CLM described by Lawrence et al. (2019), who also describe the individual model updates from CLM4 (used in NorESM1) to CLM5."**

Similarly for aerosols, I note the authors cite other papers that are in preparation, however again as a top-level documentation paper and the importance of aerosol-climate interactions for the climate response of the model some overview of the performance of aerosols in the model is needed. In particular, both sea salt and DMS emissions have been used to tune the final coupled model but no detail of this tuning nor impact on the aerosol simulation is provided.

**We will add two figures in the Supplementary material to document the performance of aerosols. The first figure shows a comparison of aerosol optical depth (AOD) at 550 nm between NorESM and observations for the period 2005--2014.  The second figure shows global mean time series over the historical period of AOD and ERF.**

**The following will be added to the manuscript:**

**The cooling over the period 1930–1970 in NorESM2 is probably caused by the combination of a low climate sensitivity (see Sect. 4.3) and a strong negative aerosol forcing. Atmosphere-only simulations with NorESM2-LM (see Olivié et al., in prep.) show that the aerosol effective radiativeforcing (ERF) strengthens from around -0.3 W m$_{-2}$around 1930 to -1.5 W m$_{-2}$in the period 1970–1980, becoming slightly weaker again in 2014 with a value of -1.36 W m$_{-2}$. On a global scale anthropogenic SO$_2$ emissions have risen strongly in the495period 1950–1980, and these are assumed to contribute most to the anthropogenic aerosol forcing. The ERF are quite similar in both model versions. We find an ERF of -1.36±0.05 W m$_{-2}$in NorESM2-LM and -1.26±0.05 W m$_{-2}$in NorESM2-MM for the year 2014**

**(compared to 1850). Figure S3b shows the time evolution of ERF for the first ensemble member ofNorESM2-LM. Given that the ERF is not an observable quantity, we have also included time series of aerosol optical depth which can be related to measurements (Fig. S3a) along with a comparison of aerosol optical depth with observations (Fig. S4).Detailed analysis of the aerosol properties is done in Olivié et al. (in prep.)**

**A more detailed description of the tuning / modifications of sea-salt and DMS is provided in the relevant sections. Details  under the specific comments.**

Tuning: More detail and clarity is needed in some aspects of the tuning description. It is evident from Section 3 that a number of variables have been used to tune the low and higher resolution models (LM and MM) but the tunings differ in a number of places between the two resolutions. A table summarizing the primary and secondary tuning parameters as well as the untuned/tuned values chosen for each configuration would be very beneficial.

**A table of all the tuning parameters for both resolution versions will be included. The table will also include the corresponding values from CESM2 where applicable.**

It should include the impact of the tuned values on an appropriate metric which ideally would be constrained by observations eg: RESTROM or the SW cloud forcing in the case of the tuning of gamma parameter (paragraph beginning 266).

**Time series of RESTOM, SW and LW cloud forcing, temperatures, AMOC, sea salt and DMS emissions are now included in the supplement for both model versions.**

The tuning was carried out in a pre-industrial climate , yet the authors set their tuning targets in order "to maintain values of mean atmospheric and ocean temperatures close to observations" (L254) , given that the observations are predominantly in the present- day the authors should comment on the limitations of any such comparison. Were any present-day simulations done in parallel to validate this tuning and evaluation?

You state (Line 224) that present-day year 2000 AMIP timeslices were used for the general development of CAM6-Nor. Given that a continuous year 2000 forcing is not a realistic representation of present-day climate or of observations over recent decades, can the authors comment on the decision to use year 2000 forcing instead of a timeseries forcing and implications this may have on the model development and evaluation.

**The entire section will be rewritten in order to give a better description of the development with respect to present and pre-industrial conditions. This is included in the manuscript.**

**"Similar to CESM, NorESM2 adjusted towards its coupled climatology with an initial phase of strong cooling in the high latitudes of the northern hemisphere, after which an intensification of ocean heat advection stabilised the simulation. After that point, the climatology tended to settle to a steady drift. During major tuning steps, the coupled model had to be restarted from the initial state several times. In order to save computer resources, minor tuning, especially toward reducing RESTOM, was performed on the best-candidate simulation after this initial, large adjustment. Alongside the final tuning, the CESM components were updated to the versions found in CESM2.1.  In this second phase of coupled spin-up, it was found that the sensitivity of some aspects of the simulated coupled climatology to small changes in parameters or parameterisations could be different than that found in stand-alone simulations of the individual components with prescribed boundary conditions.The**

**coupled response could be both amplified or damped with respect to single-component simulations. As a result, some of the final parameter tuning of the model had to be performed in coupled mode."**

**The main goal of the coupled tuning process was to create an energy balanced pre-industrial control simulation with a reasonably stable, adjusted equilibrium state. The simulation can produce a steady climatology only if the time average of the radiative imbalance on the top of the model vanishes. In practice, a commonly used target is to bring RESTOM within ±0.1 W m$^{-2}$ while maintaining values of mean atmospheric and ocean temperatures close to observations. To achieve this, each change in the coupled model was tested in parallel in atmosphere-only (AMIP) and ocean-only (OMIP) mode. Because ocean heat gain and tropospheric air temperature, humidity and cloudiness are strongly associated with the top of the atmosphere fluxes, improving the state in the coupled simulation, and reducing RESTOM and drift in AMIP and OMIP simulations, are closely connected goals. On the other hand, fine tuning of the coupled state should not significantly degrade important climatological variables such as temperature, precipitation, cloud, or the main mode of coupled variability, i.e. ENSO. Our parallel testing procedure ensured that the model simulation maintained a degree of consistency both with the present-day, observed climatology, and with a steady pre-industrial climate. Where available, notably in SST and sea-ice, observational estimates of the state of Earth's pre-industrial climate were also considered against the coupled integrations.**

Overall the different choice of tuning parameters for the two resolutions will impact the models evolution and therefore limits the assessment of the role of resolution on the model performance and any potential benefits of the higher resolution. Can the authors comment on this ?

**We believe that a similar and balanced pre-industrial state is more important than the tuning of a limited number of parameters, but hard to prove otherwise. Our evaluation against observations showed more realism for simulations with a higher resolution, showing its benefit. We will change the text as follows:**

**"Each tuning step was performed in isolation, and an effort was made to ensure the greatest possible similarities in the two model configurations LM and MM. No tuning was performed that attempted to target other modes of variability beside ENSO, or a particular climate response to external forcings, e.g. from changes in greenhouse gas concentration, anthropogenic aerosol emissions, or volcanic or solar forcing.**

**[...]**

**We give a concise summary of the parameters that were used for tuning NorESM2, with their final value and a comparison with CESM2, in Table 2."**

Line 250: What were the main changes from CESM2 –> CESM2.1? These should be documented, perhaps in Section 2 and here in the Tuning section document the impact of these developments on the tuning and development of NorESM2.

**We will include some information on this update. As the change was done quite early in the spin-up we do not think this had any impact on the control period:**

**"The changes from CESM2.0 to CESM2.1 are, according to Danabasoglu et al. (2020), mostly technical although with minor bug fixes and updated forcing fields. The update was done after an initial adjustment, but early in both spin-ups, approximately 1000 model years before the start of the**

**control, at both resolutions. The impact on the global fields are quite small as can be seen in the spin-up figure in the supplementary material."**

Some sort of schematic or diagram (even if included in an appendix) would be useful to aid understanding of the spin-up process, detailing offline spin-up and fully coupled spin-up. How long was the total spin-up period of the final tuned model before the official piControl for the DECK began? Later in Section 4.1 you state the abrupt 4xCO2 and 1pctCO2 runs were started at year 1 of the control -presumably you mean here the piControl but there's no indication of the full length of spin-up and how you determined that the model was fully spun-up.

**A schematic of the tuning time-line is included in the supplement.**

Ensemble size: It is very hard to assess the robustness of the NorESM2-MM model given there is only a single historical member used in this analysis. The authors need to acknowledge such limits in the text. Indeed, the historical evolution of global mean surface temperature is outside of the LM model range but its impossible to say if this is meaningful. Furthermore, it's not clear if only single ensemble members were run for the future projections in both models, it looks like this is the case but again it needs to be clearly stated and limitations on conclusions drawn need to be discussed.

**We have now updated the evaluation by including all three ensemble members of MM.**

It is very interesting how the ECS is so remarkably different in NorESM2 compared to CESM2. Can the authors expand on the detail given here? Have additional sensitivity experiments been done to pick apart the role of the differences between the two models on the response eg, ocean model, aerosol-cloud representation? It seems a very relevant investigation to understand potentially more generally the multi-model differences in ECS, in particular given the tendency of CMIP6 models to move towards higher ECS it's all the more interesting that NorESM2 has gone the other way.

**We agree with the reviewer that this is a very interesting difference. We have changed the text and included further details from the paper Gjermundsen et al. (submitted).**

**"An extensive analysis of the low ECS value in NorESM2 is given in Gjermundsen et al. (submitted, 2020). Note that the aerosol forcing is not very different between NorESM2 and CESM2 and can not explain the discrepancy in ECS values. Several sensitivity experiments have been conducted and are reported in Gjermundsen et al. in order to investigate the importance of different ice cloud schemes, CLUBB and interactive DMS. However, these NorESM2 experiments exhibit similar ECS values. The main reason for the low ECS in NorESM2 compared to CESM2 is, how the ocean models respond to GHG forcing. The behaviour of the BLOM ocean model (compared to the POP ocean model used in CESM2), contributes to a slower surface warming in NorESM2 compared to CESM2. Using the Gregory et al. (2004) method on the first 150 years leads to an ECS estimate which is considerably lower than for CESM2. However, if 500 years are included in the analysis, NorESM2 shows a sustained warming similar to CESM2. This suggests that the actual equilibrium temperature response to a large GHG forcing (the value one finds when the model is run for many hundred years) in NorESM2 and CESM2 is not very different, but that the Gregory et al. (2004) method based on the first 150 years does not give a good estimate of ECS for models."**

Line 402: What is NorESM1-Happi? Given the prevalence of its usage in the analysis of the climate response and present day performance of the NorESM2, an appropriate reference and a brief description of how it's different from NorESM1 is warranted Overall, I don't get the motivation for including NorESM1-Happi in the

analysis and find it often confuses the analysis. Also, Presumably the NorESM1 models are not driven with the updated CMIP6 forcing for instance?

> **NorESM1-Happi is now defined in the beginning of section 4. We also mention the differences between NorESM1-M and NorESM1-Happi and the motivation for including both versions of NorESM1 in the present study. By including both versions we can compare against NorESM1 versions and simulations having corresponding MM and LM resolutions. We will add the following to the manuscript:**

> **"The motivation for including NorESM1-Happi in the present paper is to present results from a low-resolution (-M) and medium-resolution version (-Happi) of NorESM1 alongside the results from the low-resolution (-LM) and medium-resolution versions (-MM) of NorESM2."**

The final "Summary and Discussion" reads really just solely as a Summary. Can the authors draw some overarching Conclusions from their analysis, which might include for instance the overall improved performance of the NorESM2 models compared to its predecessors; the motivation of 2 resolutions for NorESM2, role of resolution on model performance and potential benefits or applications of the different resolutions to different aspects of wider CMIP6 analysis. The would be beneficial to a reader and future potential user of NorESM2 data.

> **We agree that this should be a short section containing only the essential points of the paper; accordingly, we have renamed it "Summary and Conclusions". We also agree that the conclusions may have been expressed a bit too implicitly to give clear recommendations on the use of the two different model versions. We will partly give this information in section 2.1 we will include explicit timing information for the two resolutions and extended the sentence "Due to the generally high computational cost of NorESM2" with "and relatively limited amount of computing power"**

> **We will add recommendations for the use of the two model resolutions in the summary and discussion section:**

> **Improvements of NorESM2 compared to NorESM1 are already mentioned in the summary and discussion, but we will complement and clarify this in the revised manuscript.**

Length of Paper: This paper is very long. It appears to me that many of the figures are surplus to the requirements of the main thread of the analysis and don't get discussed much in the main text I would recommend moving some of them to a supplementary material. The ones that strike me are Figures 15, 21, 22, and a reduction in the number of ENSO plots (currently Figures 26-30).

> **The suggested figures 15, 21 and 22 are moved to supplementary material. We also moved four more Figures to the SM. The discussion of the MJO was shortened, summarising the main points only (a further evaluation of the MJO in NorESM will be given in a separate paper) and the respective Figures (24,25) moved to the SM. We also followed the reviewer's suggestion with regard to ENSO by moving Figures 26, 30 to the supplement; however we left the discussion of ENSO unchanged in the text. This will be the only documentation of ENSO in this model for a while, so it is important to keep it there.**

Technical Comments:

The Title should reflect that use is made use of ScenarioMIP simulations also.

> **Included Title reads now: "Overview of The Norwegian Earth System Model, NorESM2, and Key Climate Response of CMIP6 DECK, Historical, and Scenario Simulations".**

Line 46: participates –> participate

**Corrected.**

Line 80: It would be nice to inform the reader why the decision was made not to include land-ice model.

**We have included this additional information.**

**"Our tests with an interactive ice-sheet model over Greenland show that the model does not maintain a realistic mass balance, indicating that further development is needed. For CESM, specific tuning was carried out in order to achieve a better Greenland ice-sheet mass balance. Although NorESM2 inherited such tunings, its warmer regional climate would have required additional, dedicated effort. Due to resource limitations we have postponed this until after CMIP6."**

Line115: Please include an appropriate reference for the prescribed optical properties used for the stratospheric aerosol."

**Included: "Monthly distributions of stratospheric sulfate aerosols follow the CMIP6 recommendations: Concentration, surface area density, and volume density are based on the work of Thomason et al. (2018)."**

line 121: Can the authors quantify the impact of tuning both the sea salt (and later the DMS) on the total emissions of these natural sources and perhaps AOD and cloud droplet numbers? Were the aerosol tunings applied in the same way in both LM and MM?

**As described in the tuning section, the aerosol parameters are the same for both model resolutions.**

**With respect to sea-salt tuning the word tuning in this connection is imprecise since this change was done prior to the tuning process and followed the recommendations in published sea-salt articles.**

**Rewritten as:**

**The equation for sea-salt emissions has been modified by changing their dependence on 10-meter wind speed. NorESM2adopts the value recommended by Salter et al. (2015), 3.74, for the exponential factor, instead of 3.41 in NorESM1. This change was partly justified as an early tuning prior to the start of the spin-up simulations, in order to reduce the large positive top of the model radiative imbalance of the model before temperature equilibration. Even with the lower exponential factor however the model already produced excessive sea-salt aerosol optical depth (Gliß et al., 2020) and surface mass concentrations (Olivié etal., in prep.) compared to in-situ observations. Thus the change results in an even larger overestimate. Since the emission flux of oceanic primary organic aerosols is proportional to that of fine sea-salt aerosols (Kirkevåg et al., 2018), this specific change also has an impact on the natural oceanic organic matter emissions.**

**Additional text.**

**"The sea-salt emission changes were tested in a predecessor model version, NorESM1.2 (Kirkevåg et al., 2018). Annual and globally averaged, this lead to increases from 99.5 to 228.3 ng/m2/s (129%) in sea-salt emissions, from 7.8 to 17.2 mg/m2 (121%) in sea-salt column burdens, with**

**corresponding changes in total clear-sky AOD from 0.086 to 0.119 (38%), and cloud droplet numbers at top of the cloud (using the method of Kirkevåg et al., 2018) changed from 31.3 to 32.7 cm-3 (4.5%).**

**For a more detailed analysis of the impact of changes in the natural emissions please see Olivié et al. (2020)."**

Sect 2.4: does the dust aerosol impact the iron fertilisation of the ocean?

**No it doesn't. This is specified in the text**

**"Currently the atmospheric deposition into the ocean is decoupled. The ocean biogeochemistry uses the monthly climatological aerial dust (iron) deposition of Mahowald et al. (2005)."**

**Mahowald, N., Baker, A., Bergametti, G., Brooks, N., Duce, R., Jickells, T., Kubilay, N., Prospero, J., and Tegen, I.: Atmospheric global dust cycle and iron inputs to the ocean, Global Biogeochem. Cycles, 19, 4025, https://doi.org/10.1029/2004GB002402, 2005.**

Line 224: I don't understand what is meant by a "data-atmosphere" for the offline forcing of the ocean components – presumably the atmospheric forcing came from the CAM6-Nor simulations?

**With "forced by a data-atmosphere" we meant "forced with prescribed atmosphere and runoff of the OMIP1 protocol", so not using forcing from CAM6-Nor. This is made clear in the text.**

Line 246: "towards its climatology" –> towards its own climatology

**Corrected**

Line 252: steady –> steady-state?

**Replaced and slightly reformulated.**

Line 266: Define the acronym CLUBB

**Defined**

Line 281: "detail here" appears to be inserted here in error. Some quantification of the change in seasalt and DMS emissions should be given.

**Included as in the comment related to "line 121".**

Line 300: The lower biological production of DMS is said to agree better with observations but leads to an underestimation in DMS emissions. What observations are used here? How do the authors explain this apparent discrepancy?

**The 'lower biological production here refers to the primary production (PP) through phytoplankton photosynthesis, and not the DMS production. NorESM1 simulates too strong bias in its spring bloom PP in the high latitude, and is now alleviated in NorESM2. The statement**

**"Compared to Schwinger et al. (2017), NorESM2 has doubled the diatom-mediated DMS production parameter in order to maintain the observed high DMS concentration at high latitudes. This tuning is necessary due to the lower biological production simulated in NorESM2 (relative to NorESM1), which**

**is a better representation to the observations, during spring bloom in both hemispheres (Tjiputra et al., 2019)."**

**Rephrased to**

**"Compared to Schwinger et al. (2017), in NorESM2 the parameter controlling DMS production by diatoms was doubled, which allowed to maintain high DMS concentration at high latitudes during spring and summer seasons in both hemispheres, as in observations (Lana et al., 2011). This tuning compensates for the reduced primary production simulated in NorESM2 compared to that in NorESM1 (Tjiputra et al., 2020). "**

Line 305: greenhouse gas climate scenarios → greenhouse gas future climate scenarios.

> **done**

Lines 308âAT310: Please make sure you include the appropriate references here for the CMIP6 DECK and ScenarioMIP experimental protocols.

> **Added**

Line 330: were there any particular criterion used for the choice of initialisation years ?

> **No. The section has been made simpler only including the interval between the ensemble members.**
>
> **"Following CMIP6 guidelines, for this experiment we carried out a small ensemble of integrations, consisting of 3 members. This helps isolate the forced signal from internal climate variability. The three model integrations of the ensemble differ only in their initial conditions, which were obtained from model states late in the spin-up at intervals of 30 model years apart. This is analogous to the historical ensemble of NorESM1 produced for CMIP5..**

Line 331: it seems a great shame that there is only one ensemble member of NorESM2-MM included in this analysis and severely limits any conclusions drawn about the performance of this model compared to the LM equivalent. If additional members can be added this would significantly add to the value of the analysis. If not, the limitations of having only 1 ensemble member should be highlighted in the text.

> **The other ensemble members have been run and we have been able to update the figures and text in the revised paper with the two new HIST members of NorESM-MM. Since both resolutions have the same number of ensemble members the sentence about MM is removed here.**

Line 334: future climate development –> future climate change?

> **done**

Line 338: RCPs – define acronym

> **done**

Line 354: please define what you mean by a "sufficiently long spin-up"

**The sentence is deleted as it is superfluous. If the control simulation is stable enough fulfilling the requirements, the length of the spin-up is irrelevant.**

Line 371: abrupt4xCO2 –> abrupt-4xCO2

**Done**

Line 420: Does the model have a representation of nitrate aerosol?

**No. A sentence stating "Nitrate aerosols are not included." is added to the aerosol description"**

"it is likely that the aerosol forcing is similar in both model versions" – can you actually make this statement given the different cloud tuning in the two configurations with potential implications for the marine stratocumulus for instance?

**The ERF has now been calculated so we can give the actual figures. We will add (see above) the aerosol ERF values in text (obtained from comparing two RFMIP simulations, i.e. piClim-aer and piClim-control) : we find an ERF of -1.36 ±0.05 W/m$^2$ in NorESM2-LM and -1.26 ±0.05 W/m2 in NorESM2-MM in 2014 (compared to 1850).**

Line 473 "sea level is lower" –> sea level anomaly is lower.

**Corrected**

Line 488: Fig 9 here I think should refer to Fig 20

**Referred to both (surface salinity and total precipitation)**

Line 489 Fig 10 –> Fig 11

**Corrected**

Line 550: North Atlantic –> North American continent (also biases prevalent in south America).

**We will modify this as suggested**

Line 554: "are mitigated"–> are reduced

**Done**

Figure 14: Why is the LM model so much warmer (at the surface) than MM? Is this a consequence of the different tunings? Its hard to tell from Figure 2 if the net TOA in the piControl is overall warmer in the LM model.

**The average 500 yr TOA imbalance in NorESM2-MM and LM is very similar (-0.057 W m$^{-2}$ for NorESM2-LM and -0.065 W m$^{-2}$ for NorESM2-MM). The small difference is a residual at equilibration (after SSTs have warmed) and is therefore not the *cause* for LM being warmer. A brief discussion is added**

**The stronger cool tropospheric and warm surface tropical bias of NorESM-LM compared with NorESM-MM is in line with the behaviour of both NorESM1 and CESM2. The systematic difference between the two atmosphere resolutions is also consistent between coupled and AMIP simulations, with CAM-Nor significantly cooler at two degree resolution than at one degree resolution for the same SSTs and the same physics parameters. At the same time, tropospheric specific humidity (and, a fortiori, relative humidity) is higher.**

**Both lead to higher corresponding RESTOM. The ultimate cause of this systematic dependence of the simulated climatology on the resolution of the atmosphere model is not known. There may be a sensitivity of the convection parameterization to the grid-scale variability of near-surface air parameters and to boundary-layer stability. Another possibility is a resolution dependence of cloud microphysics and the efficiency of stratiform precipitation. LWP and column precipitable water appear almost uniformly higher in CAM-Nor at two degree resolution than at one degree resolution.**

Line 625 "modelled cloud cover" – presumable the 70% here is referring to a global mean value. It would be helpful to refer the reader back to Table 2 here.

    **Done**

Line 626: The reference to Fig 15 seems to be oddly placed in between Fig 19 and Fig 20. I suggest moving the location of the figure.

    **Renumbered to figure 19. Figure 16-19 → 15-18**

Line 633: "reanalysis. along"? This whole sentence needs to be corrected as Figure 20 does not use GPCP data.

    **Figure 20 does use GPCP data, but there was an error in the caption. Both the caption and the sentence (L633 in the submitted version)  is corrected.**

Figure 20: Either the figure caption of the figure labelling is incorrect here in terms of what model is plotted in what panel, please correct.

    **Corrected.**

Line 695: friquency –> frequency

    **Corrected.**

Line 700: "NorESM-LM (Figure 25(b)) –> this is inconsistent with the Figure caption of Figure 25 which states panel b = NorESM-MM. Please double check all figure captions to make sure they are correct and consistent with the main text.

    **Corrected (The figure is also moved to the supplement. )**

Figures 26 – 29 : It feels like there are a disproportionately a lot of figures for Section 5.9. Are all these needed in the main text, can some be moved to supplementary section?

**Figure 26 and 30 is moved to the supplement. As mentioned above, ENSO is a fundamental aspect of the coupled model climatology, and we do not foresee to publish another evaluation of NorESM2 in this respect. So we believe the text of the ENSO section should stay unchanged in this paper.**

Line 751: "medium-resolution version of the model" – you should clearly state here that the resolution differences here relate to the atmosphere.

**Included in the sentence atmospheric in the sentence. Now reads " the atmospheric medium-resolution version of the model (NorESM2-MM) and a low-resolution version (NorESM2-LM).**

Figure 8: March and September lines should be clearly marked on the plots.

**Will be corrected before final publication.**

Figure 25: In the figure caption the final sentence is incomplete.

**This will be corrected.**

**Reply to anonymous Referee #2**

The paper is a high-level description of development, tuning, and key CMIP6 simulations of NorESM2. That includes a discussion of climate sensitivity and several aspects of the climatological state of the model.

The paper is half-way between an overview paper and an evaluation paper. It works well as an overview, covering the main development activities, simulations and results. I like the openness of the description of tuning strategies. Sharing components with CESM2 brings the interesting aspect of the impact of ocean/etc. on different on key metrics like sensitivity, which may provide interesting opportunities for new insights.

The paper does not work as an evaluation paper. The evaluation mostly looks at physical aspects (radiation, clouds, ocean state, sea ice, ENSO) and the more "Earth system" components are not evaluated at all. My suggestion is to focus on the overview, delegating the evaluation of specific components to companion papers. The present paper should then be re-titled and reframed, with minimal effort, as an overview paper only. This reframing would be a good opportunity to make section 5 more balanced in terms of text-to-figure ratio: many figures are only briefly mentioned in the text, so could go.

**The title and introduction have been modified to give more focus to the general overview of the model. Title reads now: "Overview of The Norwegian Earth System Model, NorESM2, and Key Climate Response of CMIP6 DECK, Historical, and Scenario Simulations". Some of the more detailed analysis e.g. seasonal precipitation cycle and some of the ENSO aspects have been moved to a supplement. More details on which and how figures are moved are given in the answers to reviewer 1.**

**1 Specific comments**

Caption of Figure 1: The information given in parentheses could more efficiently be put in the boxes directly.

**I OVerlaying additional  text to the plot panels results in less legibility; we therefore prefer to keep this Figure unchanged.**

Line 64: I'm curious to know how those modifications were chosen. In response to perceived deficiencies in CESM2? Different scientific priorities? Ad-hoc developments that happened to be ready?

**Our main initial thrust with regard to CAM dynamics and formulation was directed at improving the local and global conservation properties of the model, and, in a related way, to remove obvious model resolution dependencies, in the belief that this might also bring advantages to the fidelity of the coupled model to observations at both targeted resolutions. A posteriori, our results appear to support such belief.  Scientific priorities of our own were to include the CAM Oslo aerosol scheme, allowing for coupling with marine biogeochemistry (DMS) not part of CESM along with associated further adjustments to emissions and fluxes.**

Line 99: That paragraph would be a good place to say what the time step of the different models is.

**We have added the following text:**

**"As in CAM6, a 30 minute physics timestep is used, with four-fold and eight-fold dynamics substepping for LM and MM, respectively."**

Line 99: That paragraph could be organised more efficiently. Related statements should be grouped together, for example all statements related to emissions; then chemistry; then volcanic forcing; then optical properties. Bullet points would work well here.

**The paragraph has been split up into changes in external forcings (given as bullet points) and  other changes. The whole section is included here**

[revised manuscript text omitted]

Lines 123-124: What do you mean? The model should not cool in such simulations... Do you mean improve the radiative balance of the model?

**We will rewrite this as follows:**

**This change was partly justified as an early tuning prior to the start of the spin-up simulations, in order to reduce the large positive top of the model radiative imbalance of the model before temperature equilibration.**

Line 128: Kirkevag 2018 is unclear as to what particles acted as coagulation sink in the previous version. It should be clarified here.

**In the previous version only the fine mode of co-nucleated sulfate and SOA (mixture no. 1) acted as a coagulation sink for the 2 nm particles. The text is changed accordingly.**

Line 129: "a more realistic rate" What was the previous value? How big is the change?

**Added text**

**"In NorESM1.2 the survival rates in the lower troposphere changed from typically 20 - 80% to 1 - 20% (zonally and annually averaged). Kuang et al. (2009) inferred survival probabilities from size distribution measurements and found that at least 80 % of the nucleated particles measured at Atlanta, GA and Boulder, CO were lost by coagulation before the nucleation mode reached CCN sizes, even during days with high growth rates."**

Line 139: How is the mean cloud-free relative humidity calculated? Assuming 100% RH in the cloudy part?

**Added text**

**"The cloud-free relative humidity is calculated assuming 100 % relative humidity in the cloudy volume."**

Line 254: Need to clarify your secondary tuning target. Was it absolute temperature of the preindustrial state, the present-day state, or present-day temperature anomalies? The latter two imply a tuning of the response.

**We have given a fuller explanation in the text. Basically, at every coupled tuning step towards reducing TOA imbalance in the pre-industrial (PI) climate, we used parallel stand-alone atmosphere and ocean integration to validate against present-day, observed climate. In essence, we tuned TOA in coupled mode under PI forcings, and state in stand-alone mode und present-day (PD) forcings. So tuning to observations was performed on PD climate. There was no explicit tuning of the response, since we did not target a detailed PI state, but only the PI fluxes (to equilibrium under PI forcings) while trying to minimise coupled model drift; nor did we tune to PD (satellite-era) observed absolute TOA fluxes -- only, partially, the PD cloud radiative forcings -- when adjusting the AMIP/OMIP states towards observations.**

Paragraphs starting lines 270 and 275: Those two paragraphs are confusing. Which changes made it and which didn't?:

**The paragraph starting at line 270 will be changed to.**

**"Given the same gamma values and otherwise identical parameter values the RESTOM was higher in the low-resolution version of the model"**

**We have also further elucidated the role of tuning gamma nd dcs in the two model version further down in this paragraph.**

Line 272: "the final parameter values" – might as well give those values here.

**This will be added along with a reference to the tuning table in the supplementary material**

Line 281: That statement looks incomplete.

**The whole paragraph is reformulated as follows:**

**"A more effective tuning of low-cloud radiative effects was achieved by modifying air-sea fluxes of DMS. Compared to Schwinger et al. (2017), NorESM2 has doubled the diatom-mediated DMS production parameter in order to maintain the observed high DMS concentration at high latitudes. This tuning is necessary due to the lower biological production simulated in NorESM2 (relative to NorESM1), which is a better representation to the observations, during spring bloom in both hemispheres (Tjiputra et al., 2019).**

Lines 366-367: Is that drift related to the ocean temperature drift?

**Most of the remaining drift in ocean biogeochemistry variables is likely not very dependent on the ocean temperature drift.**

Lines 379: Should cite the examples of long equilibrium studies.

**Added Paynter et al. (2018) show results from simulations with GFDL-CM3 and GFDL-ESM2 run for more than 4000 years.**

Lines 389-391: It would be useful to show that 500-year simulation on Fig 3. Is there a change in warming rate at some point in time, or is it just a question of time to equilibrium?

**An extensive analysis of the low ECS, including time series of temperature, is given in Gjermundsen et al 2020 (submitted). There is no substantial change in the warming rate in NorESM2 (except for the first 20 years compared to the later), but the equilibrium time scale differs substantially from CESM2. The text was unclear on this point. We will change the text but leave further details to the paper Gjermundsen et al. (submitted).**

**The text has been modified**

**"An extensive analysis of the low ECS value in NorESM2 is given in Gjermundsen et al. (submitted, 2020). Note that the aerosol forcing is not very different between NorESM2 and CESM2 and can not explain the discrepancy in ECS values. Several sensitivity experiments have been conducted and are reported in Gjermundsen et al. in order to investigate the importance of different ice cloud schemes, CLUBB and interactive DMS. However, these NorESM2 experiments exhibit similar ECS values. The main reason for the low ECS in NorESM2 compared to CESM2 is, how the ocean models respond to GHG forcing. The behaviour of the BLOM ocean model (compared to the POP ocean model used in CESM2), contributes to a slower surface warming in NorESM2 compared to CESM2. Using the Gregory et al. (2004) method on the first 150 years leads to an ECS estimate which is considerably lower than for CESM2. However, if 500 years are included in the analysis, NorESM2 shows a sustained warming similar to CESM2. This suggests that the actual equilibrium temperature response to a large GHG forcing (the value one finds when the model is run for many hundred years) in NorESM2 and CESM2 is not very different, but that the Gregory et al. (2004) method based on the first 150 years does not give a good estimate of ECS for models."**

Line 396: I suppose that the slower warming in NorESM2 means that its TCR is lower than that of CESM2?

**Yes, that is correct.**

**"The TCR of both NorESM2-LM and NorESM2-MM are lower than the value of 2.0 K found for CESM2."**

Line 417: Is that really the explanation? Isn't it normally a good thing to have a low climate sensitivity when having a strong forcing?

**We agree that the mentioning of low climate sensitivity is not helpful here, and the reasons for the 50s cooling is currently under discussion: The sentence reads now: "The cooling over the period 1930–1970 in NorESM2 is probably caused by a relatively strong negative aerosol forcing.**

Line 418: Perhaps say that this is the effective radiative forcing

**Included**

Line 425: A good way to summarise the numbers in that paragraph is that the absolute temperature simulated by MM is almost 1 degree warmer than LM throughout the 1850-2100 period, but anomalies are similar.

**The sentence "Although the historical warming is slightly weaker in NorESM2-MM compared to NorESM2-LM, the warming at the end of the 21st century is rather similar in both versions of**

> **NorESM2." is replaced with "The absolute temperature simulated by LM is almost 1 degree warmer than MM throughout the 1850-2100 period, but anomalies are similar."**

Line 468: Although I do not have specific comments on section 5, that section needs to focus on main results only, clearly summarising which model/model and model/obs differences are understood, which are not, and which differences affect the model response to forcing.

> **We will strengthen the focus of the manuscript on the main results, and move some of the detailed analysis to the supplementary material.**

Paragraphs starting lines 436 and 444 and Figures 6-7: SSP126 looks like an outlier in a couple of these timeseries. Is that just variability among ensemble, or is there something more than that?

> **Only one realisation of each scenario makes it uncertain if it is only internal variability or not.**

Figure 15 should be re-numbered, as it is used after Figure 19.

> **done**

**2 Technical comments**

Line 15: Satisfactorily -> satisfactory

> **done**

Line 47: Delete "Also"

> **done**

Line 793: Typo "properties"

> **done**

Graff

[revised manuscript text omitted]

---

## Author Response (AR2)

Dear editor

Please find my answers to the points you raised

1. Firstly, "code must be published on a persistent public archive with a unique identifier for the exact model version described in the paper or uploaded to the supplement, unless this is impossible for reasons beyond the control of authors. "

**This has now been included:**
**"Code availability.The NorESM2 code can be downloaded from**
**https://doi.org/doi:10.5281/zenodo.3905091 (Seland et al., 2020)"**

2. Model outputs on ESGF is very good practice. However the reader has no way of actually finding them! ESGF provides the correct data citation for each piece of data on the corresponding catalogue page and precise instructions for citing CMIP6 data, for example, are here: https://docs.google.com/document/d/1SnwBL9MJQNEU1_nJ661SN3-SSxR0j1mAYan6-WXfaSU/edit

**References to the data have now been added**
 **"The NorESM2-LM data (Seland et al., 2019) and NorESM2-MM data(Bentsen et al., 2019) can be accessed through the Earth System Grid Federation (ESGF) decentralized database (https://esgf-node.llnl.gov).The NorESM1-Happi data (Seland, 2020) can be accessed from the NIRD Research Data Archive (https://archive.sigma2.no)"**

3. The model input files required to reproduce the simulations are not referred to. These should also be publicly and persistently archived otherwise the results are not reproducible.

**Information about how to obtain the inputdata have been added**
**"The inputdata for the experiments can be found at https://noresm.org/inputdata/. Local copies of relevant files are automatically downloaded when creating and compiling the experiments. "**

**In addition to the changes given above there has been some minor corrections in the reference lista and an additional project has been added to the acknowledgements.**

[revised manuscript text omitted]